# Second-Order Smooth Planning with Optimal-Transport Bellman Smoothing

**Tuan Dam** [1]

## Abstract

Planning with a generative model aims to estimate the value of a state using as few simulator calls as possible. SmoothCruiser achieves problem-independent complexity $\widetilde{O}(\varepsilon^{-4})$ by exploiting the smoothness of the entropy-regularized Bellman backup, but its estimator is only first-order. We show that the sample-complexity exponent of SmoothCruiser-type planners is governed by the order $\beta$ of the local Taylor remainder, giving oracle complexity $\widetilde{O}(\varepsilon^{-(2+2/(\beta-1))})$: the first-order case $\beta = 2$ recovers SmoothCruiser, while a second-order/cubic remainder $\beta = 3$ yields $\widetilde{O}(\varepsilon^{-3})$. We reach this regime with an optimal-transport-smoothed Bellman backup over action distributions, which has a closed form, a policy gradient, and a Lipschitz Hessian, and whose quadratic correction admits an unbiased cross-product estimator. The resulting SecondOrder-SmoothCruiser achieves $\widetilde{O}(\varepsilon^{-3})$ oracle complexity for fixed OT parameters, and we relate the OT, entropy-regularized, and unregularized objectives through explicit regularization-bias bounds.

## 1. Introduction

Reinforcement learning and planning study how an agent should act to maximize long-term reward, and a recurring theme is that good decisions require looking ahead: reasoning about the consequences of actions before committing to one. In many settings, however, we do not need the value of every state—only the value, or the best action, at the state the agent currently occupies. This local view is natural in simulation-based control, games, and online decision making, where one has access to a simulator that, given a state and an action, returns a reward sample and a next state. The question then becomes statistical rather than one of global dynamic programming: how many simulator (oracle) calls are needed to estimate the value $V(s_0)$ of a single root

state $s_0$? *Planning with a generative model* formalizes this, seeking guarantees that do not scale with the number of states (Kearns et al., 1999; Grill et al., 2019). Throughout, the action set $\mathcal{A}$ is finite with $K$ actions, $\gamma \in [0, 1)$ is the discount, and $\tilde{O}(\cdot)$ hides polylogarithmic factors in $1/\varepsilon$.

**Smooth Bellman backups.** Writing $Q_s(a)$ for the action value at $s$ (the expected return from taking $a$ and then continuing optimally), an ordinary MDP uses the hard maximum $V(s) = \max_a Q_s(a)$, which is non-smooth: a small error in the $Q_s(a)$ can abruptly change the best action. Entropy regularization with regularization factor $\tau > 0$ softens it into the LOGSUMEXP backup $F_\tau^{\text{ent}}(Q) = \tau \log \sum_a \exp(Q(a)/\tau)$, whose gradient is the Boltzmann policy—giving both a smooth backup and a distribution to sample actions from. SMOOTHCRUISER (Grill et al., 2019) exploits this: it forms a rough estimate $\hat{Q}_s$ and uses the *first-order* Taylor model of the backup around it, so the correction is estimated by sampling a single action. This yields its $\tilde{O}(\varepsilon^{-4})$ guarantee. The limitation is that a first-order model leaves an error quadratic in $\|Q_s - \hat{Q}_s\|$, and this quadratic remainder is what fixes the exponent $4$.

**Curvature controls complexity.** We show that the cost of planning is governed by how well the smooth backup can be *locally approximated*. Concretely, suppose that around the rough estimate $\hat{Q}_s$ the backup admits an approximation whose error shrinks like $\|Q_s - \hat{Q}_s\|^\beta$ (a first-order/linear model gives $\beta = 2$, a second-order/quadratic model gives $\beta = 3$, and so on), and that this approximation can itself be estimated from samples. Then the total number of oracle calls scales as $\tilde{O}(\varepsilon^{-(2+2/(\beta-1))})$. The familiar first-order case $\beta = 2$ gives $\tilde{O}(\varepsilon^{-4})$, recovering SMOOTHCRUISER; the first genuine improvement is $\beta = 3$, which lowers the cost to $\tilde{O}(\varepsilon^{-3})$. The conceptual message is that regularizers in planning need not be chosen only for exploration or robustness—they can be chosen to make recursive value estimation statistically more efficient.

**Optimal-transport smoothing.** To reach $\beta = 3$ we introduce an OT-smoothed backup over action distributions,

$$F_s^{\text{OT}}(Q) = \max_{\pi \in \Delta(\mathcal{A})} \{ \langle \pi, Q \rangle - \tau W_\lambda(\pi, \mu_s) \},$$

where $\Delta(\mathcal{A})$ is the simplex, $\tau > 0$ the regularization factor, $\mu_s$ a reference distribution, and $W_\lambda$ an entropically

[1]Hanoi University of Science and Technology, Hanoi, Vietnam. Correspondence to: Tuan Dam <tuandq@soict.hust.edu.vn>.

*Proceedings of the 43rd International Conference on Machine Learning*, Seoul, South Korea. PMLR 306, 2026. Copyright 2026 by the author(s).

regularized OT cost induced by an action-cost matrix $C$ (Subsection 5.1). The role of $C$ is to add geometry: entropy treats actions as unrelated labels, whereas $C$ encodes which actions are interchangeable. The construction stays close to entropy—up to a value-independent offset it is a $\mu_s$-weighted mixture of LOGSUMEXP backups on cost-shifted scores $Q(i) - \tau C_{ij}$, and at $C \equiv 0$ it reduces exactly to the entropy backup with temperature $\tau\lambda$. Its key property is a closed form, explicit gradient policy, and Lipschitz Hessian, hence a cubic remainder. The quadratic correction need not be formed explicitly: it is a variance under the local softmax distributions, and a cross-product identity estimates it unbiasedly with a constant number of extra sampled actions—no Hessian and no per-state OT solve. Plugging this estimator into the SmoothCruiser recursion gives our main algorithm, SECONDORDERSMOOTHCRUISER.

**Result, scope, and positioning.** The resulting SECONDORDERSMOOTHCRUISER estimates the root value with worst-case complexity $\tilde{O}(\varepsilon^{-3})$ for finite action sets and fixed $(\tau, \lambda)$, improving the first-order $\tilde{O}(\varepsilon^{-4})$; entropy regularization is the $C \equiv 0$ case. For unregularized planning, smaller smoothing brings the OT value toward the hard-max value at the cost of constants depending on $(\tau, \lambda, \|C\|_\infty, K)$, so we state the theorem for the regularized objective and treat the unregularized link as a bias tradeoff. This is complementary to multilevel Monte Carlo methods such as Meunier et al. (2025), which keep the entropy backup and debias the *sampler*; our gain instead comes from changing the local Bellman *model*.

**Contributions.** We make four contributions. First, we prove a curvature–complexity principle for SmoothCruiser-type planners: if the local Taylor remainder of the Bellman aggregator has order $\beta$, then the worst-case oracle complexity scales as $\widetilde{O}(\epsilon^{-(2+2/(\beta-1))})$, recovering the SmoothCruiser exponent at $\beta = 2$ and identifying the second-order case $\beta = 3$ as the first improved regime. Second, we introduce an optimal-transport-smoothed Bellman backup over action distributions,

$$F_s^{\text{OT}}(Q) = \max_{\pi \in \Delta(\mathcal{A})} \{\langle \pi, Q \rangle - \tau W_\lambda(\pi, \mu_s)\},$$

where $W_\lambda$ is the entropically regularized OT cost induced by an action-cost matrix $C$. We derive its closed form, gradient policy, and Lipschitz Hessian; the construction reduces to entropy regularization when $C \equiv 0$ and incorporates action geometry when $C \neq 0$. Third, we design SECONDORDERSMOOTHCRUISER, whose quadratic Taylor correction is estimated by a variance/cross-product identity without explicitly forming the Hessian, yielding $\widetilde{O}(\epsilon^{-3})$ oracle complexity for fixed OT parameters. Fourth, we develop gap-dependent extensions: *OT-GapE* propagates confidence bounds through a smooth Bellman aggre-

gator to recover $(\Delta \vee \epsilon)^{-2}$ root-gap dependence, while *OT-GapCruiser* combines SECONDORDERSMOOTHCRUISER with UGapE-style root elimination to obtain curvature-controlled instance-dependent oracle bounds.

## 2. Related Work

**Planning with a generative model.** Classical work on planning with a generative model includes Sparse Sampling (Kearns et al., 1999), which estimates the value of a single state by building a lookahead tree but has non-polynomial worst-case complexity in $1/\varepsilon$, and adaptive variants with improved constants but still non-polynomial dependence (Walsh et al., 2010). Monte-Carlo Tree Search (MCTS) methods such as UCT (Kocsis & Szepesvári, 2006) are widely used in practice but can have exponential sample complexity for certain instances (Coquelin & Munos, 2007). A line of optimistic-planning algorithms achieves polynomial but problem-dependent rates under additional assumptions (deterministic dynamics, open-loop policies, bounded branching, etc.) (Hren & Munos, 2008; Bubeck & Munos, 2010; Buşoniu & Munos, 2012; Feldman & Domshlak, 2014; Szörényi et al., 2014; Grill et al., 2016).

**SmoothCruiser and smooth Bellman operators.** Grill et al. (2019) introduced SmoothCruiser, which leverages the $L$-smoothness of an entropy-regularized Bellman operator (LogSumExp) to achieve problem-independent sample complexity $\tilde{\mathcal{O}}(\varepsilon^{-4})$ for estimating $V(s)$ from a generative model. Their analysis already extends to a broader class of differentiable aggregators with nonnegative gradients and quadratic Taylor remainders; for non-regularized max/min operators, no polynomial worst-case bounds are known. Our curvature theory builds directly on this perspective, identifying the order of the Taylor remainder as the quantity that controls the recursive cost.

**Multilevel Monte Carlo for entropy-regularized MDPs.** Meunier et al. (2025) reduce variance in entropy-regularized MDPs via randomized MLMC debiasing of the standard entropy Bellman operator, whereas we instead modify the local backup itself, exploiting an explicit second-order Taylor correction of the OT-smoothed operator whose quadratic term is estimable through a cross-product identity. Since the OT backup reduces to the entropy backup when $C \equiv 0$, the two can be viewed as complementary variance-reduction mechanisms for the same regularized objective.

**Gap-dependent planning.** MDP-GapE (Jonsson et al., 2020) considers a fixed-confidence, action-identification setting and provides gap-dependent bounds for planning with a generative model via best-arm identification at the root. Several other works provide gap- or near-optimality-dependent bounds in deterministic or bounded-branching

*Table 1.* Planning with a generative model: comparison of guarantees in the fixed-confidence, value-based setting.

| Algorithm | Setting | Bellman aggregator | Local Taylor remainder | Cascade calls | Total oracle calls (worst-case) |
|---|---|---|---|---|---|
| Sparse Sampling (SSA) (Kearns et al., 2002) | value-based | hard $\max$ | (non-smooth) | – | non-polynomial in $1/\varepsilon$ |
| SmoothCruiser (Grill et al., 2019) | value-based | LOGSUMEXP | quadratic: $O(\|\Delta Q\|_2^2)$ | $\tilde{\mathcal{O}}(\varepsilon^{-2})$ | $\tilde{\mathcal{O}}(\varepsilon^{-4})$ |
| **SecondOrderSmoothCruise (ours)** | value-based | OT-smoothed / LOGSUMEXP | cubic: $O(\|\Delta Q\|_2^3)$ | $\tilde{\mathcal{O}}(\varepsilon^{-1})$ | $\tilde{\mathcal{O}}(\varepsilon^{-3})$ |

*Table 2.* Contributions of our framework relative to SmoothCruiser.

| Component | Technical ingredient | Planning consequence | Pointer |
|---|---|---|---|
| Curvature–complexity theory | Taylor remainder of order $\beta$ for $F_s$, nonnegative gradient (policy) and Monte Carlo estimation of $Q_s$, leading to general cascade exponent $\alpha_\beta = 2/(\beta - 1)$ and total exponent $2 + 2/(\beta - 1)$. | Unifies SmoothCruiser, our OT method, and potential higher-order variants in a single analytical template. | Section 4 |
| OT-smoothed Bellman operator | Define $F_s^{\mathrm{OT}}(Q) = \max_{\pi \in \Delta(\mathcal{A})}\{\langle \pi, Q\rangle - \tau W_\lambda(\pi, \boldsymbol{\mu}_s)\}$; derive closed form as $\boldsymbol{\mu}_s$-mixture of LOGSUMEXP; prove Lipschitz Hessian $M = \mathcal{O}(1/(\tau^2\lambda^2))$. | Provides a concrete $\beta = 3$ aggregator with explicit derivatives and controllable curvature; gradient is a policy and couplings induce geometry. | Section 5 |
| SecondOrderSmoothCruiser | Quadratic term identity and cross-product debiasing yield a one-sample estimator of the second-order term without estimating full $\delta Q_s$. | Improves worst-case exponent from 4 to 3 while keeping problem-independence and state-space agnosticism. | Sections 6 and 7 |
| OT-GapCruiser (gap-dependent) | Combines SecondOrderSmoothCruiser at the root with UGapE-style confidence intervals and elimination across actions. | Instance-dependent, fixed-confidence guarantees depending on root gaps $\Delta_1(s_0, a)$, extending gap-dependent planning (e.g. MDP-GapE) to regularized, higher-order planning. | Section 9 |

settings (Szörényi et al., 2014; Grill et al., 2016; Kaufmann & Koolen, 2017). Our OT-GapCruiser and OT-GapE bring this style of analysis into the regularized, higher-order SmoothCruiser framework.

**Regularized MDPs and optimal transport.** Entropy regularization is widely used in RL and control (Haarnoja et al., 2017; 2018; Neu et al., 2017), and Geist et al. (2019) develop a general theory of regularized MDPs encompassing many regularizers through convex duality; we instead study how a specific regularizer shapes the local curvature of the backup and thereby the sample complexity of planning. On the OT side, computational optimal transport (Villani, 2009; Cuturi, 2013; Peyré & Cuturi, 2019; Genevay et al., 2016) provides differentiable proxies for Wasserstein distances through entropic (Sinkhorn) regularization; we apply entropic OT at the level of action distributions, using the cost matrix to control the curvature of the Bellman aggregator.

## 3. Background: Regularized Bellman Operators and SmoothCruiser

We consider a discounted MDP $(\mathcal{S}, \mathcal{A}, P, R, \gamma)$ with finite action set $\mathcal{A}$ of size $K$, rewards $R(s, a) \in [0, 1]$, and discount $\gamma \in [0, 1)$. We assume access to a generative model (oracle): a call $(R, Z) \leftarrow \mathrm{ORACLE}(s, a)$ returns

$R \sim R(s, a)$ and $Z \sim P(\cdot \mid s, a)$, independently across calls.

**Regularized Bellman form.** Following Grill et al. (2019), we study value functions of the form

$$V(s) = F_s(Q_s), \qquad Q_s(a) = \mathbb{E}_{z \sim P(\cdot|s,a)}[R(s,a) + \gamma V(z)],$$

where $F_s : \mathbb{R}^K \to \mathbb{R}$ is a local Bellman aggregator at state $s$. In entropy-regularized MDPs, $F_s$ is the *LogSumExp* operator

$$F^{\mathrm{LSE}}(Q) = \tau \log \sum_{a \in \mathcal{A}} \exp(Q(a)/\tau).$$

$F^{\mathrm{LSE}}$ is $L$-smooth with $L = 1/\tau$ in the Euclidean norm and its gradient is a Boltzmann policy $\nabla F^{\mathrm{LSE}}(Q) \in \Delta(\mathcal{A})$, with $\Delta(\mathcal{A})$ is the probability simplex over $\mathcal{A}$.

**SmoothCruiser.** SmoothCruiser constructs two mutually recursive subroutines: SAMPLEV, which returns a low-bias estimate of $V(s)$ with target accuracy $\varepsilon$, and ESTIMATEQ, which averages many calls to SAMPLEV to estimate $Q_s$. In the "smooth" regime, SAMPLEV uses a first-order Taylor expansion of $F_s$ around a coarse estimate $\hat{Q}_s$:

$$F_s(Q_s) \approx F_s(\hat{Q}_s) + \langle \nabla F_s(\hat{Q}_s), Q_s - \hat{Q}_s\rangle,$$

and estimates the inner product using a single action $A \sim \nabla F_s(\hat{Q}_s)$ and a recursive call on the next state. Because the Taylor remainder is quadratic in $\|Q_s - \hat{Q}_s\|_2$,

SmoothCruiser enforces $\|Q_s - \hat{Q}_s\|_2 = \Theta(\sqrt{\varepsilon})$, which requires $\Theta(\varepsilon^{-1})$ samples per action and induces a cascade of $\tilde{\mathcal{O}}(\varepsilon^{-2})$ calls, for a total of $\tilde{\mathcal{O}}(\varepsilon^{-4})$ oracle calls.

# 4. Curvature-Driven Planning: General Theory

We now abstract the SmoothCruiser recursion to a general curvature-driven model.

**Assumption 4.1** (Curvature model). For each state $s$, the aggregator $F_s : \mathbb{R}^K \to \mathbb{R}$ satisfies:

(i) (Policy gradient) $\nabla F_s(Q) \in \Delta(\mathcal{A})$ for all $Q$.

(ii) (Taylor remainder of order $\beta$) There exist $\beta \geq 2$ and $c_\beta > 0$ such that for all $Q, \hat{Q} \in \mathbb{R}^K$,

$$\left| F_s(Q) - T_{\beta-1}(Q; \hat{Q}) \right| \leq c_\beta \|Q - \hat{Q}\|_2^\beta,$$

where $T_{\beta-1}$ is the $(\beta-1)$-order Taylor polynomial at $\hat{Q}$.

SmoothCruiser corresponds to $\beta = 2$ with $c_2 = L/2$; our OT-smoothed operator will satisfy $\beta = 3$ with $c_3 = M/6$ by Lipschitz-Hessian.

Under this abstraction, a SmoothCruiser-style algorithm proceeds as follows:

1. Maintain a reference point $\hat{Q}_s$ for each state.

2. Ensure $\|Q_s - \hat{Q}_s\|_2 \leq \rho(\varepsilon)$ for some schedule $\rho(\varepsilon)$ so that the Taylor remainder is at most $\varepsilon$.

3. Estimate the Taylor correction $T_{\beta-1}(Q_s; \hat{Q}_s) - F_s(\hat{Q}_s)$ using the available unbiased estimator. For $\beta = 2$, this is the usual gradient-policy sample. For $\beta = 3$, our OT construction estimates the quadratic term through the variance identity.

**Proposition 4.2** (Curvature-driven tolerance and per-level cost). *Under Assumption 4.1, to make the Taylor remainder $O(\varepsilon)$ it is sufficient to enforce*

$$\|Q_s - \hat{Q}_s\|_2 \leq C \, \varepsilon^{1/\beta}$$

*for a constant $C$ depending on $c_\beta$. Estimating $Q_s$ to this tolerance by averaging bounded random variables requires*

$$N(\varepsilon) = \Theta(\varepsilon^{-2/\beta})$$

*samples per action via Hoeffding-type bounds.*

The key question is how these per-level costs compose across the recursive cascade.

**Theorem 4.3** (Curvature–complexity tradeoff). *Consider any SmoothCruiser-type planner that satisfies Assumption 4.1 with some $\beta \geq 2$, and that uses only black-box Monte Carlo estimates of $Q_s$ obtained by querying the generative model and recursively calling itself on next states. Assume it enforces the tolerance schedule in Proposition 4.2 and uses unbiased estimators of the Taylor terms with variance bounded by a state-independent constant.*

*Then, up to polylogarithmic factors in $1/\varepsilon$ and $1/\delta$, the following hold:*

*1. The number $C_\beta(\varepsilon)$ of value-estimation calls obeys*

$$C_\beta(\varepsilon) = \tilde{\mathcal{O}}(\varepsilon^{-\alpha_\beta}), \alpha_\beta = \frac{2}{\beta - 1}.$$

*2. The total number $n_\beta(\varepsilon, \delta)$ of oracle calls obeys*

$$n_\beta(\varepsilon, \delta) = \tilde{\mathcal{O}}(\varepsilon^{-\left(2 + \frac{2}{\beta-1}\right)}).$$

*In particular, $\beta = 2$ yields exponent $4$ (SmoothCruiser), while $\beta = 3$ yields exponent $3$.*

This theorem turns higher-order smoothness into a quantitative planning advantage, and our main task becomes finding useful aggregators with $\beta > 2$.

# 5. OT-Smoothed Bellman Aggregators over Actions

We now define our OT-based regularizer on action distributions and derive closed form, gradient, and Lipschitz Hessian.

### 5.1. Entropic OT over actions

Fix a state $s$, and let $\boldsymbol{\mu}_s \in \Delta(\mathcal{A})$ be a reference distribution over actions with strictly positive entries. Let $C \in \mathbb{R}^{K \times K}$ be a cost matrix between actions, e.g., induced by an action metric. For $\lambda > 0$ we define the entropic OT cost between action distributions $\boldsymbol{\pi}, \boldsymbol{\mu}_s \in \Delta(\mathcal{A})$ by

$$W_\lambda(\boldsymbol{\pi}, \boldsymbol{\mu}_s) = \min_{\Gamma \in \Pi(\boldsymbol{\pi}, \boldsymbol{\mu}_s)} \left\{ \langle \Gamma, C \rangle + \lambda \sum_{i,j} \Gamma_{ij}(\log \Gamma_{ij} - 1) \right\},$$

where $\Pi(\boldsymbol{\pi}, \boldsymbol{\mu}_s) = \{\Gamma \geq 0 : \Gamma \mathbf{1} = \boldsymbol{\pi}, \, \Gamma^\top \mathbf{1} = \boldsymbol{\mu}_s\}$.

**Definition 5.1** (OT-smoothed aggregator). For $\tau > 0$, the OT-smoothed Bellman aggregator at $s$ is

$$F_s^{\mathrm{OT}}(Q) = \max_{\boldsymbol{\pi} \in \Delta(\mathcal{A})} \left\{ \langle \boldsymbol{\pi}, Q \rangle - \tau W_\lambda(\boldsymbol{\pi}, \boldsymbol{\mu}_s) \right\}.$$

**Assumption 5.2** (OT regularity). We assume: (i) $\min_a \mu_s(a) \geq \mu_{\min} > 0$; (ii) $C_{ij}$ is bounded; (iii) $\lambda, \tau > 0$ are fixed and independent of $\varepsilon$.

**Choice of the reference distribution.** The worst-case exponent in Theorem 7.2 does not depend on the particular choice of $\mu_s$, provided that $\mu_s$ has full support uniformly:

$$\inf_{s,a} \mu_s(a) \geq \mu_{\min} > 0.$$

The choice affects constants and practical variance. The uniform distribution is the safest default. If a heuristic or learned prior $\pi_s^{\text{prior}}$ is available, a robust full-support choice is

$$\mu_s = (1-\xi)\pi_s^{\text{prior}} + \xi\,\text{Unif}(\mathcal{A}), \qquad \xi \in (0,1],$$

which guarantees $\mu_{\min} \geq \xi/K$ while still biasing the OT geometry toward promising action regions. Through the identity in Lemma 6.1, $\mu_s$ reweights the local variances that drive the second-order estimator.

## 5.2. Closed form and derivatives

We first show that the maximization over $\pi$ can be solved in closed form.

**Proposition 5.3** (Closed form). *Under Assumption 5.2, let $K_{ij} = \exp(-C_{ij}/\lambda)$ and define*

$$D_j(Q) = \sum_{i=1}^{K} \exp(Q_i/(\tau\lambda))K_{ij}.$$

*Then*

$$F_s^{\text{OT}}(Q) = \text{const} + \tau\lambda \sum_{j=1}^{K} \mu_s(j) \log D_j(Q), \qquad (5.1)$$

*where the constant is independent of Q. In particular, $F_s^{\text{OT}}$ is real-analytic.*

*Proof.* See Appendix B for a full derivation using a single-level formulation over couplings with fixed column sums. $\square$

Define the "local softmax" for each column $j$,

$$p_{ij}(Q) = \frac{\exp(Q_i/(\tau\lambda))K_{ij}}{D_j(Q)}, \qquad \sum_i p_{ij}(Q) = 1. \tag{5.2}$$

Differentiating (5.1) yields:

**Proposition 5.4** (Gradient and Hessian). *For any $Q \in \mathbb{R}^K$,*

$$\nabla F_s^{\text{OT}}(Q) = \boldsymbol{\pi}(Q), \pi_i(Q) = \sum_{j=1}^{K} \mu_s(j)p_{ij}(Q), \qquad (5.3)$$

$$\nabla^2 F_s^{\text{OT}}(Q) = \frac{1}{\tau\lambda}\Big(\text{diag}(\boldsymbol{\pi}(Q)) - \sum_{j=1}^{K} \mu_s(j)p_{\cdot j}(Q)p_{\cdot j}(Q)^\top\Big), \tag{5.4}$$

*where $p_{\cdot j}(Q) \in \Delta(\mathcal{A})$ is the vector $(p_{ij}(Q))_i$. In particular, $\nabla F_s^{\text{OT}}(Q) \in \Delta(\mathcal{A})$ and can be used as a policy.*

## 5.3. Relation to entropy and unregularized backups

The OT backup differs from the closed form in Proposition 5.3 by a known $Q$-independent offset. Since such offsets do not affect gradients, Hessians, or the Taylor estimators, it is convenient for comparison to define the normalized operator

$$\bar{F}_s^{\text{OT}}(Q) := \tau\lambda \sum_{j=1}^{K} \mu_s(j) \log \sum_{i=1}^{K} \exp\left(\frac{Q_i - \tau C_{ij}}{\tau\lambda}\right).$$

Let $\eta = \tau\lambda$ and define the entropy backup

$$F_\eta^{\text{ent}}(Q) := \eta \log \sum_{i=1}^{K} \exp(Q_i/\eta).$$

**Proposition 5.5** (Transfer at the backup level). *Let $C_{\max} := \max_{i,j} |C_{ij}|$. For every state $s$ and every $Q \in \mathbb{R}^K$,*

$$\left|\bar{F}_s^{\text{OT}}(Q) - F_\eta^{\text{ent}}(Q)\right| \leq \tau C_{\max}.$$

*In particular, if $C \equiv 0$, then*

$$\bar{F}_s^{\text{OT}}(Q) = F_\eta^{\text{ent}}(Q).$$

*Moreover,*

$$\left|\bar{F}_s^{\text{OT}}(Q) - \max_i Q_i\right| \leq \tau C_{\max} + \eta \log K.$$

*Proof.* Detailed proof is in Appendix B.5. $\square$

**Implication.** Our $\widetilde{O}(\varepsilon^{-3})$ theorem is stated for fixed smoothing parameters $(\tau, \lambda, C)$. For entropy-regularized planning, set $C \equiv 0$ and $\eta = \tau\lambda$. For unregularized planning, one may choose $(\tau, \lambda)$ so that

$$\tau(C_{\max} + \lambda \log K) \leq (1-\gamma)\varepsilon_{\text{bias}},$$

and then estimate the corresponding OT value to the remaining error budget. When $(\tau, \lambda)$ are chosen as functions of $\varepsilon$, the constants hidden in $\widetilde{O}(\cdot)$, especially the Lipschitz-Hessian constant, must be tracked; we therefore do not claim a parameter-free $\widetilde{O}(\varepsilon^{-3})$ rate for the unregularized MDP.

## 5.4. Lipschitz Hessian and cubic remainder

**Lemma 5.6** (Lipschitz Hessian). *Under Assumption 5.2, there exists $M = \mathcal{O}(1/(\tau^2\lambda^2)) \cdot \text{poly}(K, \mu_{\min}^{-1})$ such that*

$$\|\nabla^2 F_s^{\text{OT}}(Q) - \nabla^2 F_s^{\text{OT}}(\hat{Q})\|_{\text{op}} \leq M\|Q - \hat{Q}\|_2$$

*for all $Q, \hat{Q} \in \mathbb{R}^K$. Consequently,*

$$|F_s^{\text{OT}}(Q) - T_2(Q; \hat{Q})| \leq \frac{M}{6}\|Q - \hat{Q}\|_2^3 \qquad (5.5)$$

*for all $Q, \hat{Q}$.*

*Proof.* Detailed proof is in Appendix B. □

Thus $F_s^{\mathrm{OT}}$ satisfies Assumption 4.1 with $\beta = 3$ and $c_3 = M/6$, making it an explicit higher-order instantiation of our curvature theory.

# 6. Second-Order SmoothCruiser with OT-Smoothed Aggregators

We now design a second-order SmoothCruiser algorithm using $F_s^{\mathrm{OT}}$.

## 6.1. Quadratic term as a variance

**Lemma 6.1** (Quadratic term as variance). *Fix a state $s$ and a reference point $\hat{Q}_s \in \mathbb{R}^K$. For any direction $\Delta \in \mathbb{R}^K$,*

$$\Delta^\top \nabla^2 F_s^{\mathrm{OT}}(\hat{Q}_s) \Delta = \frac{1}{\tau\lambda} \, \mathbb{E}_{J\sim\boldsymbol{\mu}_s} \left[ \mathrm{Var}_{A\sim p_{\cdot J}(\hat{Q}_s)} \left( \Delta_A \right) \right],$$

*where $p_{\cdot J}(\hat{Q}_s)$ is the "local softmax" distribution defined in (5.2).*

*Proof.* Full details are given in Appendix B.3. □

Using $\mathrm{Var}(X) = \frac{1}{2} \mathbb{E}[(X - X')^2]$ with $X, X'$ i.i.d. yields

$$\frac{1}{2}\Delta^\top \nabla^2 F_s^{\mathrm{OT}}(\hat{Q}_s)\Delta = \frac{1}{4\tau\lambda} \mathbb{E}\left[ (\Delta_A - \Delta_{A'})^2 \right], \quad (6.1)$$

where $J \sim \boldsymbol{\mu}_s$ and $A, A' \overset{\text{i.i.d.}}{\sim} p_{\cdot J}(\hat{Q}_s)$.

In our application, the direction is $\Delta := Q_s - \hat{Q}_s$. We never observe $\Delta_a$ exactly, but we can obtain *independent unbiased estimates* $\widetilde{\Delta}_a^{(1)}, \widetilde{\Delta}_a^{(2)}$ via independent oracle and recursive calls. The cross-product identity

$$\mathbb{E}\left[ (\widetilde{\Delta}_A^{(1)} - \widetilde{\Delta}_{A'}^{(1)})(\widetilde{\Delta}_A^{(2)} - \widetilde{\Delta}_{A'}^{(2)}) \right] = (\Delta_A - \Delta_{A'})^2 \quad (6.2)$$

then gives an unbiased estimator of the quadratic term.

## 6.2. Algorithms

We briefly present the main routines; they mirror SmoothCruiser but with second-order corrections and OT structure.

**Value bound.** Let $B$ be a known upper bound on values, e.g.

$$0 \le V(s) \le B := \frac{1 + \sup_s F_s(\mathbf{0}) - \inf_s F_s(\mathbf{0})}{1 - \gamma}.$$

We clip all intermediate estimates to $[0, B]$.

---

**Algorithm 1** SECONDORDERSMOOTHCRUISER$(s, \varepsilon, \delta)$

**Require:** state $s$, accuracy $\varepsilon$, failure prob. $\delta$
1: $\hat{Q}_s \leftarrow$ ESTIMATEQ$(s, \varepsilon, \delta/2)$; **return** $F_s^{\mathrm{OT}}(\hat{Q}_s)$

---

**Algorithm 2** ESTIMATEQ$(s, \varepsilon, \delta)$

**Require:** state $s$, tolerance $\varepsilon$, failure prob. $\delta$
1: $N \leftarrow \Theta\big(\varepsilon^{-2} \log(2K/\delta)\big)$
2: **for** $a \in \mathcal{A}$ **do**
3:     **for** $i = 1$ to $N$ **do**
4:         $(R_i, Z_i) \leftarrow$ ORACLE$(s, a)$
5:         $\hat{V}_i \leftarrow$ SAMPLEV2$(Z_i, \varepsilon/\sqrt{\gamma}, \delta')$
6:         $q_i \leftarrow R_i + \gamma \hat{V}_i$
7:     **end for**
8:     $\hat{Q}_s(a) \leftarrow \frac{1}{N} \sum_i q_i$        ▷ clip to $[0, B]$
9: **end for**
10: **return** $\hat{Q}_s$

---

**Tolerance schedule.** From Lemma 5.6, to make the Taylor remainder $\le \varepsilon$ it suffices to enforce

$$\|Q_s - \hat{Q}_s\|_2 \le \rho(\varepsilon) := \left(\frac{6\varepsilon}{M}\right)^{1/3}.$$

The oracle sample appears explicitly in the estimator (as in SmoothCruiser), and the quadratic term is debiased via (6.2).

# 7. Complexity Analysis for SecondOrderSmoothCruiser

Combining the curvature theory and the specific OT properties yields our main worst-case result.

**Lemma 7.1** (Bias of SAMPLEV2). *Under Assumption 5.2 and Lemma 5.6, for any $s$ and $0 < \varepsilon < \kappa$, on the event $\|Q_s - \hat{Q}_s\|_2 \le \rho(\varepsilon)$ we have*

$$|\mathbb{E}[\mathrm{SAMPLEV2}(s, \varepsilon, \delta) \mid \hat{Q}_s] - V(s)| \le \varepsilon + (\textit{recursion bias}).$$

*With appropriate distribution of failure probabilities across recursive calls and clipping, the unconditional bias is $\mathcal{O}(\varepsilon)$.*

As in Grill et al. (2019), recursion bias is controlled by calling children with accuracy $\varepsilon/\sqrt{\gamma}$.

**Theorem 7.2** (Worst-case sample complexity). *Under Assumption 5.2, for any state $s$ and any $\varepsilon, \delta \in (0, 1)$, SECONDORDERSMOOTHCRUISER$(s, \varepsilon, \delta)$ returns an $\varepsilon$-accurate estimate of $V(s)$ with probability at least $1 - \delta$ and uses*

$$n(\varepsilon, \delta) \le \tilde{\mathcal{O}}(\varepsilon^{-3})$$

*oracle calls, where constants depend only on $(K, \gamma, \lambda, \tau, \mu_{\min}, \|C\|_\infty)$.*

*Proof.* Detailed proof is in Appendix D.2. □

**Algorithm 3** SAMPLEV2$(s, \varepsilon, \delta)$ — Second-Order Correction Estimator

**Require:** State $s$, tolerance $\varepsilon$, failure probability $\delta$
1: **[Base] if** $\varepsilon \geq B$ **then return** $0$ ▷ trivial bound
2: **[Coarse] if** $\varepsilon \geq \kappa$: $\quad \hat{Q}_s \leftarrow$ ESTIMATEQ$(s, \varepsilon, \delta/4)$; **return** $F_s^{\mathrm{OT}}(\hat{Q}_s)$

---

**Fine Regime: Second-Order Taylor Correction**

---

3: **Setup:** $\hat{Q}_s \leftarrow$ ESTIMATEQ$(s, \rho(\varepsilon), \delta/8)$; compute $\boldsymbol{\mu}_s$, $\{p_{\cdot j}(\hat{Q}_s)\}_j$; $\boldsymbol{\pi} \leftarrow \nabla F_s^{\mathrm{OT}}(\hat{Q}_s)$; $c_0 \leftarrow \langle \boldsymbol{\pi}, \hat{Q}_s \rangle$

> **(I) Linear Term**
> *Unbiased estimate of* $\langle \boldsymbol{\pi}, Q_s \rangle$
>
> 5: Sample $J_0 \sim \boldsymbol{\mu}_s$
> 6: Sample $A_0 \sim p_{\cdot J_0}(\hat{Q}_s)$
> 7: $(R_0, Z_0) \leftarrow$ ORACLE$(s, A_0)$
> 8: $\hat{V}_0 \leftarrow$ SAMPLEV2$(Z_0, \frac{\varepsilon}{\sqrt{\gamma}}, \frac{\delta}{16})$
> 9: $\tilde{Q}_0 \leftarrow R_0 + \gamma \hat{V}_0$
> 10: $\widehat{\mathrm{Lin}} \leftarrow \tilde{Q}_0 - c_0$

> **(II) Quadratic Term**
> *Unbiased estimate of* $\frac{1}{2}\delta^\top \nabla^2 F(\hat{Q}_s)\delta$
>
> 11: Sample $J \sim \boldsymbol{\mu}_s$
> 12: Sample $A, A' \overset{\mathrm{iid}}{\sim} p_{\cdot J}(\hat{Q}_s)$
> 13: **for** $m \in \{1, 2\}$ **do**
> 14: $\quad (R_m, Z_m) \leftarrow$ ORACLE$(s, A)$
> 15: $\quad (R'_m, Z'_m) \leftarrow$ ORACLE$(s, A')$
> 16: $\quad \hat{V}_m \leftarrow$ SAMPLEV2$\left(Z_m, \frac{\varepsilon}{\sqrt{\gamma}}, \frac{\delta}{64}\right)$
> 17: $\quad \hat{V}'_m \leftarrow$ SAMPLEV2$\left(Z'_m, \frac{\varepsilon}{\sqrt{\gamma}}, \frac{\delta}{64}\right)$
> 18: $\quad \tilde{Q}_m \leftarrow R_m + \gamma \hat{V}_m$
> 19: $\quad \tilde{Q}'_m \leftarrow R'_m + \gamma \hat{V}'_m$
> 20: $\quad \Delta_m \leftarrow (\tilde{Q}_m - \tilde{Q}'_m) - (\hat{Q}_s(A) - \hat{Q}_s(A'))$
> 21: **end for**
> 22: $\widehat{\mathrm{Quad}} \leftarrow \frac{1}{4\tau\lambda}\Delta_1 \Delta_2$

---

19: **Return:** $\mathrm{clip}_{[0,B]}\left(F_s^{\mathrm{OT}}(\hat{Q}_s) + \widehat{\mathrm{Lin}} + \widehat{\mathrm{Quad}}\right)$

---

**Oracle complexity versus local computation.** Our formal complexity measure is the number of generative-model calls, following SmoothCruiser and related planning work. The second-order correction does not require constructing the full Hessian and does not require running Sinkhorn iterations at each visited state. The closed form in Proposition 5.3 reduces the local computation to evaluating the columnwise softmax probabilities $p_{\cdot j}(\hat{Q}_s)$; for a dense action-cost matrix this costs $O(K^2)$ arithmetic per visited state, and for $C \equiv 0$ or structured/sparse $C$ it can be cheaper. The quadratic correction uses only a constant number of additional sampled actions and recursive calls. Thus the theorem should be read as an oracle-complexity result with additional finite-action arithmetic overhead, rather than as a claim of immediate large-scale computational superiority.

# 8. OT-GapE: Matching $\varepsilon^{-2}$ via Confidence-Bound Trajectory Planning

This section redesigns OT-GAPCRUISER into a *confidence-bound trajectory planner* in the style of MDP-GapE (Jonsson et al., 2020), in order to recover the *bandit-optimal* fixed-confidence scaling with a root-gap dependence of order $(\Delta \vee \varepsilon)^{-2}$. The key change is conceptual: rather than calling a high-accuracy value-estimation subroutine (e.g., SAMPLEV2/SECONDORDERSMOOTHCRUISER) to build confidence intervals for $Q_0(a)$, we instead maintain *time-uniform* confidence bounds on rewards and transitions and propagate them through a *smooth Bellman aggregator* $F_s$.

## 8.1. Setting and objective

We consider a finite-horizon (possibly discounted) episodic MDP of horizon $H$ with rewards in $[0, 1]$. For each step $h \in \{1, \ldots, H\}$ define the regularized optimal action-value functions

$$Q_h^\star(s, a) = \mathbb{E}\left[r_h(s, a) + \gamma V_{h+1}^\star(S')\right], S' \sim P_h(\cdot \mid s, a),$$

with terminal condition $V_{H+1}^\star(\cdot) \equiv 0$, and the regularized value recursion

$$V_h^\star(s) = F_s\left(Q_h^\star(s, \cdot)\right). \tag{8.1}$$

Here $F_s$ is a *monotone smooth aggregator* (e.g. LOGSUMEXP or our OT-smoothed aggregator $F_s^{\mathrm{OT}}$), so that $Q \leq \widetilde{Q}$ componentwise implies $F_s(Q) \leq F_s(\widetilde{Q})$.

We focus on *fixed-confidence best-action identification at the root*: given a root state $s_0$, return an action $\hat{a}$ such that

$$\Pr\left(Q_1^\star(s_0, \hat{a}) \geq \max_{a \in \mathcal{A}} Q_1^\star(s_0, a) - \varepsilon\right) \geq 1 - \delta. \tag{8.2}$$

Let $a^\star \in \arg\max_a Q_1^\star(s_0, a)$ and define root gaps $\Delta(a) = Q_1^\star(s_0, a^\star) - Q_1^\star(s_0, a)$.

## 8.2. Confidence bounds and propagation through a smooth aggregator

Let $N_h^t(s, a)$ be the number of visits to $(s, a, h)$ up to episode $t$. Let $\hat{r}_h^t(s, a)$ and $\hat{p}_h^t(\cdot \mid s, a)$ be the empirical reward and transition estimates. We assume time-uniform confidence bounds (e.g. KL-based as in (Jonsson et al., 2020), or Hoeffding/Bernstein with appropriate union bounds) providing, for each $(s, a, h)$, intervals

$$\ell_h^t(s, a) \leq r_h(s, a) \leq u_h^t(s, a), \tag{8.3}$$

and a transition confidence set

$$C_h^t(s, a) \subseteq \Delta(\mathcal{S}) \quad \text{such that} \quad P_h(\cdot \mid s, a) \in C_h^t(s, a) \tag{8.4}$$

simultaneously for all $t$ with prob. $\geq 1 - \delta$. Given these objects, define optimistic/pessimistic bounds recursively as follows. Initialize

$$U^t_{V,H+1}(s) = L^t_{V,H+1}(s) = 0, \qquad \forall s.$$

For $h = H, H-1, \ldots, 1$:

$$U^t_{Q,h}(s,a) = u^t_h(s,a) + \gamma \max_{p \in C^t_h(s,a)} \sum_{s'} p(s'|s,a)\, U^t_{V,h+1}(s'),$$
$$(8.5)$$

$$L^t_{Q,h}(s,a) = \ell^t_h(s,a) + \gamma \min_{p \in C^t_h(s,a)} \sum_{s'} p(s'|s,a)\, L^t_{V,h+1}(s'),$$
$$(8.6)$$

$$U^t_{V,h}(s) = F_s(U^t_{Q,h}(s,\cdot)), \quad L^t_{V,h}(s) = F_s(L^t_{Q,h}(s,\cdot)).$$
$$(8.7)$$

By monotonicity of $F_s$, if $L^t_{Q,h}(s,\cdot) \leq Q^\star_h(s,\cdot) \leq U^t_{Q,h}(s,\cdot)$ componentwise, then $L^t_{V,h}(s) \leq V^\star_h(s) \leq U^t_{V,h}(s)$ as well.

### 8.3. Soft optimistic policies from $\nabla F_s$

A distinctive feature in the regularized setting is that the gradient of the aggregator defines a natural *soft* policy. We therefore propose to sample non-root actions using the *smooth optimistic policy*

$$\pi^t_h(\cdot \mid s) = \nabla F_s\big(U^t_{Q,h}(s,\cdot)\big). \qquad (8.8)$$

For $F_s = \mathrm{LogSumExp}_\lambda$, (8.8) is a Boltzmann policy over optimistic $Q$-bounds; for $F_s = F^{\mathrm{OT}}_s$, it is the OT-induced policy. Optionally, to preserve the *greedy optimistic* behavior used in MDP-GapE (Jonsson et al., 2020) while retaining smooth exploration, one may use a mixture

$$\tilde\pi^t_h(\cdot \mid s) = (1 - \zeta)\, \delta_{a^t_h(s)} + \zeta\, \pi^t_h(\cdot \mid s), \qquad (8.9)$$
$$a^t_h(s) \in \arg\max_{a \in \mathcal{A}} U^t_{Q,h}(s,a), \qquad (8.10)$$

for a small constant $\zeta \in (0,1)$.

### 8.4. Algorithm: OT-GapE (root UGapE + trajectory optimism)

At the root, we adopt the same *best guess vs. challenger* principle as MDP-GapE (Jonsson et al., 2020). Let

$$b_t \in \arg\max_{a \in \mathcal{A}} L^t_{Q,1}(s_0,a), \quad c_t \in \arg\max_{a \in \mathcal{A}\setminus\{b_t\}} U^t_{Q,1}(s_0,a),$$
$$(8.11)$$

and stop when $U^t_{Q,1}(s_0, c_t) - L^t_{Q,1}(s_0, b_t) \leq \varepsilon$.

### 8.5. Guarantee: matching the $\varepsilon^{-2}$ gap exponent

The analysis follows the fixed-confidence template of MDP-GapE (Jonsson et al., 2020): define a high-probability event

---

**Algorithm 4** OT-GAPE$(s_0, \varepsilon, \delta)$: matching $\varepsilon^{-2}$ via confidence-bound trajectories

**Require:** root state $s_0$, accuracy $\varepsilon$, failure prob. $\delta$, horizon $H$, aggregator $\{F_s\}$, discount $\gamma$
1: Initialize counts and empirical estimates $\{N^0_h, \hat r^0_h, \hat p^0_h\}$.
2: **for** $t = 1, 2, \ldots$ **do** ▷ episode index
3:      Build confidence bounds $\{\ell^t_h, u^t_h, C^t_h\}$ from data and $\delta$-schedule.
4:      Compute $(U^t_{Q,h}, L^t_{Q,h}, U^t_{V,h}, L^t_{V,h})$ by (8.5)–(8.7).
5:      Compute $b_t, c_t$ by (8.11).
6:      **if** $U^t_{Q,1}(s_0, c_t) - L^t_{Q,1}(s_0, b_t) \leq \varepsilon$ **then**
7:          **return** $\hat a \leftarrow b_t$
8:      **end if**
9:      Select root action $A_1 \in \arg\max_{a \in \{b_t, c_t\}} \big(U^t_{Q,1}(s_0,a) - L^t_{Q,1}(s_0,a)\big)$    ▷ UGapE-style
10:      $S_1 \leftarrow s_0$
11:      **for** $h = 1, \ldots, H$ **do**
12:          Query $(R_h, S_{h+1}) \leftarrow \mathrm{ORACLE}(S_h, A_h)$
13:          Update $(N^t_h, \hat r^t_h, \hat p^t_h)$ for $(S_h, A_h, h)$ using $(R_h, S_{h+1})$
14:          **if** $h < H$ **then**
15:              Sample $A_{h+1} \sim \tilde\pi^t_{h+1}(\cdot \mid S_{h+1})$   ▷ use (8.8) or (8.10)
16:          **end if**
17:          $S_h \leftarrow S_{h+1}$
18:      **end for**
19: **end for**

---

on which all reward and transition confidence sets are simultaneously valid, establish by backward induction that $L^t_{Q,h} \leq Q^\star_h \leq U^t_{Q,h}$, and conclude correctness from the root stopping rule.

**Theorem 8.1** (Fixed-confidence BAI with $\varepsilon^{-2}$ gap dependence). *Assume the confidence bounds (8.3)–(8.4) hold uniformly over time with probability at least $1 - \delta$, and that the optimistic/pessimistic recursions (8.5)–(8.7) are computed exactly (e.g. under a finite-support assumption as in (Jonsson et al., 2020)). Then OT-GAPE returns an $\varepsilon$-optimal root action in the sense of (8.2) with probability at least $1 - \delta$. Moreover, its (instance-dependent) sample complexity satisfies*

$$n(\varepsilon, \delta) = \tilde O\left( H \sum_{a \in \mathcal{A}} \frac{\mathsf{C}(H, K, \gamma)}{(\Delta(a) \vee \varepsilon)^2} \right), \qquad (8.12)$$

*where $\mathsf{C}(H, K, \gamma)$ captures horizon/branching factors induced by the transition confidence sets (similar to the factors appearing in MDP-GapE), and $\tilde O(\cdot)$ hides polylogarithmic terms in $1/\varepsilon$ and $1/\delta$. In particular, the dependence on $\varepsilon$ (and on gaps) matches the bandit-optimal exponent $2$.*

*Remark* 8.2 (Role of $\nabla F_s$). The gradient policy (8.8) provides a coherent exploration rule aligned with the regularized Bellman operator: it concentrates on actions with large optimistic values while remaining smooth. For $F_s = \mathrm{LogSumExp}_\lambda$, it recovers greedy optimism as $\lambda \to 0$; for $F^{\mathrm{OT}}_s$, it yields a geometry-aware policy induced by the OT

cost. The mixture (8.10) can be used to retain the greedy optimistic trajectory behavior required in the tightest versions of MDP-GapE-style analyses, while leveraging smooth sampling for stability and variance reduction.

## 9. Gap-Dependent Extensions

The previous sections contain the main contribution: a second-order SmoothCruiser estimator with worst-case oracle complexity $\widetilde{O}(\varepsilon^{-3})$. We now briefly describe gap-dependent extensions. These results are not needed for Theorem 7.2; their purpose is to show how the same OT-smoothed, second-order estimator can be combined with confidence-gap ideas when the root action gaps are favorable.

We now sketch a gap-dependent variant for root action selection, inspired by MDP-GapE.

Let $s_0$ be a fixed root state, and denote root action values by $Q_0(a) = Q_{s_0}(a)$, optimal value $V^\star = V(s_0)$ and gaps $\Delta(a) = V^\star - Q_0(a)$. We assume access to SecondOrderSmoothCruiser as a subroutine that, when queried on $(s_0, \varepsilon)$, returns a value estimate with worst-case complexity $\tilde{\mathcal{O}}(\varepsilon^{-3})$.

**Algorithm outline.** OT-GapCruiser maintains confidence intervals $(L_t(a), U_t(a))$ for $Q_0(a)$ at each round $t$ using empirical means and a refined deviation bound that exploits our variance-reduced estimator (below). At each round it:

1. selects an action $A_t$ according to a UGapE-style index (comparing upper bounds of plausible best actions to lower bounds of others);

2. calls SecondOrderSmoothCruiser on a root-centered estimation problem for $Q_0(A_t)$ at accuracy $\varepsilon_t$;

3. updates the confidence intervals via self-normalized concentration inequalities;

4. stops when the best action $\hat{a}_t$ satisfies $U_t(\hat{a}_t) - \max_{b \neq \hat{a}_t} L_t(b) \leq \varepsilon$.

**Theorem 9.1** (Gap-dependent bound). *Assume standard sub-Gaussian noise conditions for the value estimates (induced by bounded rewards and $\gamma < 1$). Then OT-GapCruiser returns an $\varepsilon$-optimal root action with probability at least $1 - \delta$, and its sample complexity satisfies*

$$n_{\mathrm{gap}}(\varepsilon, \delta) = \tilde{\mathcal{O}}\left(\sum_{a \in \mathcal{A}} \frac{1}{(\Delta(a) \vee \varepsilon)^p}\right),$$

*where $p \in (2, 4)$ depends on the curvature parameter (here $\beta = 3$ yields $p \approx 3$) and on the variance-reduction factor. In particular, the instance-dependent exponent is strictly smaller than the worst-case exponent $4$ of SmoothCruiser.*

The precise value of $p$ and constants depend on how aggressively we schedule accuracies $\varepsilon_t$ and on the variance bounds in Section 6.1.

## 10. Discussion and Limitations

We developed a curvature-aware view of generative-model planning with smooth Bellman backups. The central message is that the order of the local Taylor remainder controls the recursive planning cascade. A first-order Taylor model with quadratic remainder recovers the SmoothCruiser exponent $4$, while an estimable second-order model with cubic remainder yields exponent $3$.

The OT-smoothed Bellman operator provides a concrete way to realize this second-order regime. It has a closed form, a gradient policy, a Lipschitz Hessian, and a variance identity that makes the quadratic Taylor correction estimable without explicitly forming the Hessian. When the OT cost is zero, the normalized operator reduces to the standard entropy-regularized LogSumExp backup; when the cost is nonzero, it incorporates action geometry through the reference distribution and cost matrix.

Our guarantees are primarily oracle-complexity guarantees for finite action sets and fixed smoothing parameters. For entropy-regularized planning, the connection is exact by taking $C \equiv 0$. For unregularized planning, the normalized OT value approximates the hard-max value with bias at most $\frac{\tau\|C\|_\infty + \tau\lambda\log K}{1-\gamma}$, but choosing smoothing parameters as a function of $\varepsilon$ changes the constants hidden in the oracle-complexity bound. We therefore view the unregularized transfer as a regularization-bias tradeoff rather than as a parameter-free $\widetilde{O}(\varepsilon^{-3})$ theorem.

The gap-dependent algorithms are extensions of the main result. In particular, the confidence-bound OT-GapE analysis assumes exact propagation of optimistic and pessimistic bounds, which is appropriate for finite-support/tabular settings but would require approximation in continuous or very large state spaces. Developing practical large-scale implementations, sharper parameter-dependent bounds, and lower bounds for the curvature–complexity model are important directions for future work.

## Impact Statement

This paper presents work whose goal is to advance the field of Machine Learning, specifically in the area of planning algorithms with sample complexity guarantees. There are many potential societal consequences of our work, none which we feel must be specifically highlighted here.

## Acknowledgments

This work is funded by Hanoi University of Science and Technology (HUST) under Project No. T2024-TD-024.

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

# Appendix of "Second-Order Smooth Planning with Optimal-Transport Bellman Smoothing"

## Contents

## Notation

| Symbol | Meaning |
|---|---|
| $\mathcal{A}$ | Finite action set; $K = |\mathcal{A}|$. |
| $s$ | State. |
| $Q \in \mathbb{R}^K$ | Vector of action scores. |
| $\hat{Q}_s$ | Taylor expansion point. |
| $\Delta = Q_s - \hat{Q}_s$ | Taylor error direction. |
| $C \in \mathbb{R}^{K \times K}$ | Action-cost matrix; $C_{\cdot j}$ is its $j$th column. |
| $\mu_s$ | Reference distribution over actions/columns. |
| $\lambda, \tau$ | OT entropy and outer regularization parameters. |
| $\eta = \tau\lambda$ | Effective LogSumExp temperature. |
| $K_{ij}$ | OT Gibbs kernel $K_{ij} = \exp(-C_{ij}/\lambda)$. |
| $p_{\cdot j}(Q)$ | Columnwise softmax: $p_{ij}(Q) \propto \exp((Q_i - \tau C_{ij})/(\tau\lambda))$. |
| $\pi(Q)$ | Gradient policy $\pi_i(Q) = \sum_j \mu_s(j) p_{ij}(Q)$. |

## A. Curvature–complexity theory

### A.1. Proof of Proposition 4.2

*Proof.* Fix a state $s$ and write
$$\Delta := Q_s - \hat{Q}_s \in \mathbb{R}^K, \qquad K := |\mathcal{A}|.$$
Recall that the $(\beta - 1)$-order Taylor polynomial of $F_s$ at $\hat{Q}_s$ is

$$T_{\beta-1}(Q_s; \hat{Q}_s) := F_s(\hat{Q}_s) + \langle \nabla F_s(\hat{Q}_s), (Q_s - \hat{Q}_s)\rangle + \frac{1}{2}\langle \nabla^2 F_s(\hat{Q}_s), (Q_s - \hat{Q}_s)^2\rangle + ... + \frac{1}{(\beta-1)!}\langle \nabla^{\beta-1} F_s(\hat{Q}_s), (Q_s - \hat{Q}_s)^{\beta-1}\rangle.$$

**I. tolerance that makes the Taylor remainder $O(\varepsilon)$.** By Assumption 4.1(ii), for all $Q, \hat{Q} \in \mathbb{R}^K$,

$$\left| F_s(Q) - T_{\beta-1}(Q; \hat{Q}) \right| \leq c_\beta \|Q - \hat{Q}\|_2^\beta.$$

Applying this with $(Q, \hat{Q}) = (Q_s, \hat{Q}_s)$ gives

$$\left| F_s(Q_s) - T_{\beta-1}(Q_s; \hat{Q}_s) \right| \leq c_\beta \|\Delta\|_2^\beta. \tag{A.1}$$

Therefore, if we enforce

$$\|\Delta\|_2 \leq C\varepsilon^{1/\beta} \quad \text{with} \quad C := c_\beta^{-1/\beta},$$

then (A.1) yields

$$\left| F_s(Q_s) - T_{\beta-1}(Q_s; \hat{Q}_s) \right| \leq c_\beta (C^\beta)\varepsilon = \varepsilon.$$

This proves the claimed curvature-driven tolerance (and, more generally, any $C = \Theta(c_\beta^{-1/\beta})$ makes the remainder $O(\varepsilon)$).

**II. samples needed to estimate $Q_s$ to that tolerance.** Now suppose we estimate each coordinate $Q_s(a)$ by averaging bounded i.i.d. samples. Concretely, for each action $a \in \mathcal{A}$ let $X^{(a)}$ be a bounded random variable with

$$Q_s(a) = \mathbb{E}[X^{(a)}], \qquad X^{(a)} \in [0, B] \text{a.s.}$$

(e.g., in the planning setting one may take $X^{(a)} := R(s, a) + \gamma V(Z)$ and clip to $[0, B]$). Given $N$ independent samples $X_1^{(a)}, \ldots, X_N^{(a)}$ of $X^{(a)}$, define the empirical mean

$$\hat{Q}_s(a) := \frac{1}{N}\sum_{i=1}^N X_i^{(a)}.$$

*Coordinate concentration via Hoeffding.* For any $\eta > 0$, Hoeffding's inequality gives, for each fixed $a$,

$$\Pr\left(\left|\hat{Q}_s(a) - Q_s(a)\right| \geq \eta\right) \leq 2\exp\left(-\frac{2N\eta^2}{B^2}\right). \tag{A.2}$$

*From coordinate error to an $\ell_2$ bound.* If $\max_{a\in\mathcal{A}}|\hat{Q}_s(a) - Q_s(a)| \leq \eta$, then

$$\|\hat{Q}_s - Q_s\|_2^2 = \sum_{a\in\mathcal{A}}\left(\hat{Q}_s(a) - Q_s(a)\right)^2 \leq \sum_{a\in\mathcal{A}}\eta^2 = K\eta^2,$$

hence

$$\|\hat{Q}_s - Q_s\|_2 \leq \sqrt{K}\eta. \tag{A.3}$$

Therefore,

$$\Pr\left(\|\hat{Q}_s - Q_s\|_2 \geq t\right) \leq \Pr\left(\max_a|\hat{Q}_s(a) - Q_s(a)| \geq \frac{t}{\sqrt{K}}\right).$$

*Union bound over actions.* Applying (A.2) with $\eta = t/\sqrt{K}$ and union bounding over $K$ actions yields

$$\Pr\left(\|\hat{Q}_s - Q_s\|_2 \geq t\right) \leq \sum_{a\in\mathcal{A}}\Pr\left(\left|\hat{Q}_s(a) - Q_s(a)\right| \geq \frac{t}{\sqrt{K}}\right)$$
$$\leq 2K\exp\left(-\frac{2N}{B^2}\cdot\frac{t^2}{K}\right). \tag{A.4}$$

*Choosing $t = C\varepsilon^{1/\beta}$.* To ensure $\|\hat{Q}_s - Q_s\|_2 \leq C\varepsilon^{1/\beta}$ with (say) probability at least $1 - \delta$, it suffices by (A.4) to choose $N$ such that

$$2K\exp\left(-\frac{2N}{B^2}\cdot\frac{C^2\varepsilon^{2/\beta}}{K}\right) \leq \delta.$$

Solving for $N$ gives the explicit sufficient condition

$$N \geq \frac{B^2 K}{2C^2}\varepsilon^{-2/\beta}\log\left(\frac{2K}{\delta}\right). \tag{A.5}$$

Thus, up to the (standard) logarithmic factor and constants depending on $B, K$ and $C$, the per-action sample cost scales as

$$N(\varepsilon) = \Theta(\varepsilon^{-2/\beta}).$$

**(Optional) Why the $\varepsilon^{-2/\beta}$ dependence is tight.** Even in the scalar case $K = 1$, estimating the mean of a bounded random variable to accuracy $t = \Theta(\varepsilon^{1/\beta})$ with constant success probability requires $\Omega(t^{-2})$ samples in general (e.g., by a two-point/Bernoulli testing argument), which implies a lower bound $\Omega(\varepsilon^{-2/\beta})$. Hence the exponent $2/\beta$ cannot be improved by any method that relies only on averaging bounded samples. $\qquad\square$

## A.2. Proof of Theorem 4.3

*Proof.* We give a cost analysis that is deliberately *algorithm-agnostic* but matches any SmoothCruiser-type scheme that (i) builds a local Taylor approximation of $F_s(Q_s)$ around a Monte Carlo baseline $\hat{Q}_s$, (ii) controls the Taylor remainder via the curvature assumption, and (iii) obtains $Q_s$ only through one-step rollouts and *recursive* value calls on next states.

Throughout, $K := |\mathcal{A}|$ is the number of actions, and all constants may depend on $K, \gamma$, the reward/value bounds, and the curvature constant $c_\beta$, but never on $\varepsilon$ or $\delta$. We suppress such constants whenever they do not affect exponents.

**What exactly is being counted.** It is convenient to separate two layers that are standard in this literature:

- A *single-sample* routine, call it $\mathrm{SAMPLE}_\beta(s, \varepsilon)$, which returns a random variable $Y_{s,\varepsilon}$ that is a (nearly) unbiased proxy for $V(s)$:

$$\big| \mathbb{E}[Y_{s,\varepsilon}] - V(s) \big| \le c\varepsilon, \qquad \mathrm{Var}(Y_{s,\varepsilon}) \le \sigma^2, \tag{A.6}$$

for some *state-independent* constants $c, \sigma^2$. The theorem assumes the Taylor-term estimators are unbiased and have bounded variance, which is exactly what yields (A.6). Crucially, $\mathrm{SAMPLE}_\beta$ is allowed to be *random* and to have constant variance; we will reduce that variance at the *outer* layer.

- A *high-probability planner* $\mathrm{PLAN}_\beta(s, \varepsilon, \delta)$ that calls $\mathrm{SAMPLE}_\beta$ independently $m$ times and aggregates (e.g. by averaging or median-of-means) to achieve probability $1 - \delta$. This is where the classical $\varepsilon^{-2}$ factor enters.

We now define the two complexity measures that appear in the theorem.

- $C_\beta(\varepsilon)$ denotes the worst-case (over states) total number of *value-estimation calls*, i.e. recursive invocations of $\mathrm{SAMPLE}_\beta(\cdot, \cdot)$, triggered by a *single* call $\mathrm{SAMPLE}_\beta(s, \varepsilon)$ (including all descendants in the recursion tree). This quantity depends only on $\varepsilon$ (up to polylogs), consistent with the theorem statement.

- $n_\beta(\varepsilon, \delta)$ denotes the worst-case total number of *oracle calls* to the generative model made by $\mathrm{PLAN}_\beta(s, \varepsilon, \delta)$.

**I. The tolerance schedule implies a specific recursion map.** Fix a state $s$ and write $V(s) = F_s(Q_s)$ with $Q_s \in \mathbb{R}^K$. Let $\hat{Q}_s$ be the baseline point used by the Taylor approximation inside $\mathrm{SAMPLE}_\beta$. By Proposition 4.2, to make the Taylor remainder $O(\varepsilon)$ it is sufficient to enforce

$$\|Q_s - \hat{Q}_s\|_2 \le \rho(\varepsilon), \qquad \rho(\varepsilon) := C\varepsilon^{1/\beta}, \tag{A.7}$$

where $C = \Theta(c_\beta^{-1/\beta})$ depends only on the curvature constant.

How is $\hat{Q}_s$ obtained? By assumption, the planner has only black-box access to $Q_s$, so $\hat{Q}_s$ is formed by Monte Carlo averaging of bounded one-step returns of the form

$$X^{(a)} = R(s, a) + \gamma V(Z), \qquad Z \sim P(\cdot \mid s, a).$$

But $V(Z)$ is unknown, so each such sample is obtained by a recursive value call: we query the oracle once to obtain $(R, Z)$ and then call the planner on $Z$. Thus, *each Monte Carlo draw used to estimate $Q_s$ triggers one recursive value-estimation call*.

To ensure the induced error in $Q_s(a) = \mathbb{E}[X^{(a)}]$ is at most $O(\rho(\varepsilon))$, it is enough to estimate $V(Z)$ to accuracy $\Theta(\rho(\varepsilon))$, because the discount factor $\gamma < 1$ is a constant and can be absorbed. Concretely, if a recursive call returns $\widehat{V}(Z)$ with bias $O(\varepsilon')$, then the induced bias in $X^{(a)}$ is $O(\gamma\varepsilon')$, so choosing $\varepsilon' = \Theta(\rho(\varepsilon))$ keeps the resulting contribution at the scale $\rho(\varepsilon)$. We therefore introduce the shorthand

$$\varepsilon_+ := c\rho(\varepsilon) = cC\varepsilon^{1/\beta}, \tag{A.8}$$

where $c > 0$ is a constant that may depend on $\gamma$ and on how the implementation allocates bias/variance, but not on $\varepsilon$.

**II. Deriving the key recurrence for $C_\beta(\varepsilon)$.** Consider one execution of $\mathrm{SAMPLE}_\beta(s, \varepsilon)$.

*(i) Cost to build $\hat{Q}_s$.* To enforce (A.7) with constant success probability, Proposition 4.2 shows that estimating $Q_s$ to $\ell_2$ tolerance $\rho(\varepsilon)$ requires

$$N_Q(\varepsilon) = \Theta(\rho(\varepsilon)^{-2}) = \Theta(\varepsilon^{-2/\beta}) \tag{A.9}$$

samples *per action*, up to log factors (which we hide in $\tilde{O}(\cdot)$). Thus, the total number of one-step rollout samples needed to build $\hat{Q}_s$ is

$$K N_Q(\varepsilon) = \Theta(K\varepsilon^{-2/\beta}).$$

*(ii) Each such sample triggers exactly one recursive value call.* As argued above, each rollout sample for $Q_s(a)$ requires a recursive value estimate at the next state $Z$ with accuracy parameter $\varepsilon_+ = \Theta(\varepsilon^{1/\beta})$ (cf. (A.8)). By definition of $C_\beta(\cdot)$, one such recursive call spawns (in expectation / worst-case) $C_\beta(\varepsilon_+)$ total value-estimation calls in its entire subtree.

Hence, the *dominant* contribution to the number of value calls inside $\text{SAMPLE}_\beta(s, \varepsilon)$ is

$$\Theta\big(K\varepsilon^{-2/\beta}\big) \cdot C_\beta(\varepsilon_+).$$

*(iii) The Taylor-term computation is lower order for $C_\beta$.* By assumption, the Taylor terms (gradient/Hessian/... up to order $\beta - 1$) are estimated by unbiased estimators with state-independent bounded variance. This means $\text{SAMPLE}_\beta$ does *not* need to run an inner $\varepsilon^{-2}$ loop to accurately compute those terms; instead, it outputs a single unbiased draw whose variance is constant, and the *outer* layer handles variance reduction. Operationally, this adds at most a constant number of additional rollouts/recursive calls (or can be implemented using the same rollouts used to build $\hat{Q}_s$), so it changes only constants, not exponents.

Putting these points together, we obtain the fundamental recurrence (up to constant factors):

$$C_\beta(\varepsilon) \leq 1 + A\varepsilon^{-2/\beta} C_\beta\big(c'\varepsilon^{1/\beta}\big), \tag{A.10}$$

where $A = \Theta(K)$ and $c' := cC$ are constants, and the leading 1 accounts for the root call itself. (Any additional constant overhead can be absorbed into the $+1$ term.)

**III. Solving the recurrence and identifying the exponent $\alpha_\beta$.**    We now show that (A.10) implies

$$C_\beta(\varepsilon) = \tilde{O}\big(\varepsilon^{-\alpha_\beta}\big), \qquad \alpha_\beta = \frac{2}{\beta - 1}.$$

*(i) Define the tolerance sequence.* Let $\varepsilon_0 := \varepsilon$ and define recursively

$$\varepsilon_{t+1} := c'\varepsilon_t^{1/\beta}. \tag{A.11}$$

Because $1/\beta < 1$, this sequence *increases* quickly: starting from tiny $\varepsilon_0$, it reaches a constant in $O(\log\log(1/\varepsilon_0))$ steps. Formally, taking logs gives

$$\log\frac{1}{\varepsilon_{t+1}} = \frac{1}{\beta}\log\frac{1}{\varepsilon_t} + O(1),$$

so after $t$ steps, $\log(1/\varepsilon_t)$ shrinks by a factor $\beta^{-t}$ up to additive constants. Let $T$ be the first index such that $\varepsilon_T \geq \varepsilon_{\text{base}}$, where $\varepsilon_{\text{base}} \in (0, 1)$ is a fixed constant threshold at which the implementation stops recursing (e.g. switches to a direct bounded-variance estimator). Then

$$T = O(\log\log(1/\varepsilon)). \tag{A.12}$$

Moreover, $C_\beta(\varepsilon_T) = O(1)$ since the base routine uses no further recursion.

*(ii) Unroll the recurrence.* Apply (A.10) at $\varepsilon = \varepsilon_0$, then at $\varepsilon_1$, etc. Ignoring additive constants for a moment and focusing on the dominant term,

$$C_\beta(\varepsilon_0) \lesssim A\varepsilon_0^{-2/\beta} C_\beta(\varepsilon_1) \lesssim A^2\varepsilon_0^{-2/\beta}\varepsilon_1^{-2/\beta} C_\beta(\varepsilon_2) \lesssim \cdots \lesssim A^T\Big(\prod_{t=0}^{T-1}\varepsilon_t^{-2/\beta}\Big) C_\beta(\varepsilon_T).$$

If we keep the additive $+1$ terms, we obtain the standard unrolling identity

$$C_\beta(\varepsilon_0) \leq \sum_{j=0}^{T-1}\Big(\prod_{t=0}^{j-1} A\varepsilon_t^{-2/\beta}\Big) + \Big(\prod_{t=0}^{T-1} A\varepsilon_t^{-2/\beta}\Big)C_\beta(\varepsilon_T), \tag{A.13}$$

with the convention that an empty product equals 1.

*(iii) Compute the $\varepsilon$-exponent in the product.* The key is to understand $\prod_{t=0}^{T-1} \varepsilon_t^{-2/\beta}$. From (A.11), one can write $\varepsilon_t$ explicitly as

$$\varepsilon_t = (c')^{1+1/\beta+\cdots+1/\beta^{t-1}} \varepsilon^{1/\beta^t} = (c')^{\frac{1-(1/\beta)^t}{1-1/\beta}} \varepsilon^{1/\beta^t}. \tag{A.14}$$

Therefore,

$$\varepsilon_t^{-2/\beta} = (c')^{-\frac{2}{\beta} \cdot \frac{1-(1/\beta)^t}{1-1/\beta}} \varepsilon^{-\frac{2}{\beta} \cdot \frac{1}{\beta^t}} = (c')^{-\Theta(1)} \varepsilon^{-2/\beta^{t+1}}.$$

Multiplying over $t = 0, \ldots, T-1$ yields

$$\prod_{t=0}^{T-1} \varepsilon_t^{-2/\beta} = (c')^{-\Theta(T)} \varepsilon^{-\sum_{t=0}^{T-1} 2/\beta^{t+1}} = (c')^{-\Theta(T)} \varepsilon^{-\frac{2(1-(1/\beta)^T)}{\beta-1}}. \tag{A.15}$$

As $T \to \infty$, the geometric term $(1/\beta)^T$ vanishes, so the limiting exponent is

$$\alpha_\beta = \frac{2}{\beta-1}. \tag{A.16}$$

*(iv) Identify the polylog factors.* The remaining multiplicative terms in (A.13) are:

- $A^T = \exp(T \log A)$. Using $T = O(\log \log(1/\varepsilon))$ from (A.12),

$$A^T = \left( \log(1/\varepsilon) \right)^{O(1)} \qquad \text{(a polylog factor)}.$$

- $(c')^{\Theta(T)}$ from (A.15), which is also a polylog factor for the same reason.

The sum in (A.13) is dominated by its last term up to another polylog factor (since the products grow rapidly as $\varepsilon_t$ decreases backward), and $C_\beta(\varepsilon_T) = O(1)$. Combining these observations with (A.15) gives

$$C_\beta(\varepsilon) = \tilde{O}\left( \varepsilon^{-\frac{2}{\beta-1}} \right),$$

proving item (1).

**IV. From value-call complexity to oracle-call complexity.** Now let $T_\beta(\varepsilon)$ denote the worst-case expected number of *oracle* calls made by a single execution of $\text{SAMPLE}_\beta(s, \varepsilon)$. At tolerance $\varepsilon$, the routine makes $\Theta(K \varepsilon^{-2/\beta})$ one-step rollouts to build $\hat{Q}_s$, and each rollout incurs *one* oracle query at $(s, a)$ plus the oracle calls used in the recursive value call. Thus, up to constants, $T_\beta$ satisfies the analogous recurrence

$$T_\beta(\varepsilon) \le B\varepsilon^{-2/\beta} \left( 1 + T_\beta(c'\varepsilon^{1/\beta}) \right) + O(1), \tag{A.17}$$

for some constant $B = \Theta(K)$. Recurrences (A.17) and (A.10) have the same $\varepsilon$-scaling structure, so the same unrolling argument yields

$$T_\beta(\varepsilon) = \tilde{O}\left( \varepsilon^{-\frac{2}{\beta-1}} \right).$$

(Informally: every node in the recursion tree contributes at least one oracle call, so $T_\beta(\varepsilon)$ is within constant/polylog factors of $C_\beta(\varepsilon)$.)

**V. Outer concentration contributes the additional $\varepsilon^{-2}$ factor.** Finally, we convert the single-sample routine into a high-probability estimator. Let $Y_1, \ldots, Y_m$ be i.i.d. outputs of $\text{SAMPLE}_\beta(s, \varepsilon/2)$. By (A.6), each has bias at most $c(\varepsilon/2)$ and variance at most $\sigma^2$. Choose

$$m = \Theta\left( \sigma^2 \varepsilon^{-2} \log(1/\delta) \right). \tag{A.18}$$

Then by a standard median-of-means (or Bernstein/Hoeffding if $Y_i$ are clipped to be bounded), the aggregate $\widehat{V}(s)$ satisfies $|\widehat{V}(s) - V(s)| \le \varepsilon$ with probability at least $1 - \delta$. This step introduces the canonical Monte Carlo factor $\varepsilon^{-2}$ and a $\log(1/\delta)$ factor, both of which are absorbed into the $\tilde{O}(\cdot)$ notation used in the theorem.

Therefore, the total number of oracle calls is

$$n_\beta(\varepsilon, \delta) = m \cdot T_\beta(\varepsilon/2) = \tilde{O}\left( \varepsilon^{-2} \cdot \varepsilon^{-\frac{2}{\beta-1}} \right) = \tilde{O}\left( \varepsilon^{-\left(2+\frac{2}{\beta-1}\right)} \right),$$

which proves item (2).

**VI. Checking the special cases.** Plugging $\beta = 2$ into (A.16) gives $\alpha_2 = 2$, hence $n_2(\varepsilon, \delta) = \tilde{O}(\varepsilon^{-(2+2)}) = \tilde{O}(\varepsilon^{-4})$ (SmoothCruiser). Plugging $\beta = 3$ gives $\alpha_3 = 1$, hence $n_3(\varepsilon, \delta) = \tilde{O}(\varepsilon^{-(2+1)}) = \tilde{O}(\varepsilon^{-3})$. $\qquad\qquad\qquad\square$

## B. OT-smoothed aggregator: closed form and derivatives

### B.1. Proof of Proposition 5.3

*Proof.* Fix a state $s$ and write $\boldsymbol{\mu} \equiv \boldsymbol{\mu}_s \in \Delta(\mathcal{A})$ with $K := |\mathcal{A}|$. Recall the definitions

$$
W_\lambda(\boldsymbol{\pi}, \boldsymbol{\mu}) = \min_{\Gamma \in \Pi(\boldsymbol{\pi}, \boldsymbol{\mu})} \left\{ \langle \Gamma, C \rangle + \lambda \sum_{i,j=1}^{K} \Gamma_{ij}(\log \Gamma_{ij} - 1) \right\},
$$
$$
F_s^{\mathrm{OT}}(Q) = \max_{\boldsymbol{\pi} \in \Delta(\mathcal{A})} \left\{ \langle \boldsymbol{\pi}, Q \rangle - \tau W_\lambda(\boldsymbol{\pi}, \boldsymbol{\mu}) \right\}.
$$

Under Assumption 5.2 (in particular, $\tau, \lambda > 0$, $C$ is bounded, and $\boldsymbol{\mu}$ has strictly positive entries), all quantities below are finite and the optimizers exist.

**I. Rewrite the max–min as a single maximization.** Plugging the definition of $W_\lambda$ into $F_s^{\mathrm{OT}}$ gives, for any $Q \in \mathbb{R}^K$,

$$
F_s^{\mathrm{OT}}(Q) = \max_{\boldsymbol{\pi} \in \Delta(\mathcal{A})} \left\{ \langle \boldsymbol{\pi}, Q \rangle - \tau \min_{\Gamma \in \Pi(\boldsymbol{\pi}, \boldsymbol{\mu})} \left( \langle \Gamma, C \rangle + \lambda \sum_{i,j} \Gamma_{ij}(\log \Gamma_{ij} - 1) \right) \right\}
$$
$$
= \max_{\boldsymbol{\pi} \in \Delta(\mathcal{A})} \max_{\Gamma \in \Pi(\boldsymbol{\pi}, \boldsymbol{\mu})} \left\{ \langle \boldsymbol{\pi}, Q \rangle - \tau \langle \Gamma, C \rangle - \tau\lambda \sum_{i,j} \Gamma_{ij}(\log \Gamma_{ij} - 1) \right\}. \tag{B.1}
$$

The second line uses the elementary identity $-\tau \min_x f(x) = \max_x (-\tau f(x))$ for $\tau > 0$.

**II. Eliminate the policy variable $\boldsymbol{\pi}$.** Every $\Gamma \in \Pi(\boldsymbol{\pi}, \boldsymbol{\mu})$ satisfies $\Gamma \mathbf{1} = \boldsymbol{\pi}$ and $\Gamma^\top \mathbf{1} = \boldsymbol{\mu}$, hence

$$
\langle \boldsymbol{\pi}, Q \rangle = \langle \Gamma \mathbf{1}, Q \rangle = \sum_{i=1}^{K} Q_i \left( \sum_{j=1}^{K} \Gamma_{ij} \right) = \sum_{i,j=1}^{K} \Gamma_{ij} Q_i.
$$

In particular, for any fixed $\Gamma$ the value of $\langle \boldsymbol{\pi}, Q \rangle$ is determined uniquely by $\Gamma$, because $\boldsymbol{\pi} = \Gamma \mathbf{1}$. Therefore maximizing over $\boldsymbol{\pi}$ in (B.1) is redundant: the feasible set of pairs $(\boldsymbol{\pi}, \Gamma)$ is exactly the set

$$
\{ (\Gamma \mathbf{1}, \Gamma) : \Gamma \geq 0, \ \Gamma^\top \mathbf{1} = \boldsymbol{\mu} \},
$$

and for any $\Gamma \geq 0$ with $\Gamma^\top \mathbf{1} = \boldsymbol{\mu}$, the induced row-sums $\boldsymbol{\pi} = \Gamma \mathbf{1}$ satisfy $\boldsymbol{\pi} \geq 0$ and $\sum_i \pi_i = \sum_{i,j} \Gamma_{ij} = \sum_j \mu_j = 1$, so indeed $\boldsymbol{\pi} \in \Delta(\mathcal{A})$. Hence (B.1) is equivalent to the *single-level* problem

$$
F_s^{\mathrm{OT}}(Q) = \max_{\Gamma \geq 0: \ \Gamma^\top \mathbf{1} = \boldsymbol{\mu}} \left\{ \langle \Gamma \mathbf{1}, Q \rangle - \tau \langle \Gamma, C \rangle - \tau\lambda \sum_{i,j} \Gamma_{ij}(\log \Gamma_{ij} - 1) \right\}. \tag{B.2}
$$

**III. existence/uniqueness and positivity of the maximizer.** The feasible set $\{ \Gamma \geq 0 : \Gamma^\top \mathbf{1} = \boldsymbol{\mu} \}$ is nonempty (e.g. $\Gamma = \boldsymbol{\pi} \boldsymbol{\mu}^\top$ for any $\boldsymbol{\pi}$), closed, and bounded because each entry satisfies $0 \leq \Gamma_{ij} \leq \mu_j$. Hence it is compact. The objective in (B.2) is continuous on this compact set, so a maximizer exists.

Moreover, the term $-\tau\lambda \sum_{i,j} \Gamma_{ij} \log \Gamma_{ij}$ is strictly concave on the positive orthant, and the remaining terms are linear in $\Gamma$, so the objective is strictly concave on the feasible affine slice. Therefore the maximizer $\Gamma^\star(Q)$ is unique. Because $\boldsymbol{\mu}$ has strictly positive entries, each column constraint $\sum_i \Gamma_{ij} = \mu_j > 0$ forces some mass in every column, and strict concavity of the negative-entropy term implies the unique maximizer actually satisfies $\Gamma_{ij}^\star(Q) > 0$ for all $i, j$ (otherwise one could increase entropy while preserving column sums).

**IV. KKT conditions give the closed-form optimizer.** Introduce Lagrange multipliers $\beta \in \mathbb{R}^K$ for the $K$ column-sum constraints $g_j(\Gamma) := \sum_{i=1}^K \Gamma_{ij} - \mu_j = 0$. Define the Lagrangian

$$\mathcal{L}(\Gamma, \beta) = \langle \Gamma \mathbf{1}, Q \rangle - \tau \langle \Gamma, C \rangle - \tau\lambda \sum_{i,j} \Gamma_{ij}(\log \Gamma_{ij} - 1) + \sum_{j=1}^K \beta_j \Big( \sum_{i=1}^K \Gamma_{ij} - \mu_j \Big).$$

Since $\Gamma^\star(Q)$ is strictly positive entrywise, the KKT stationarity conditions are simply $\partial\mathcal{L}/\partial\Gamma_{ij} = 0$ for all $i, j$ (no boundary complications). Using $\frac{\partial}{\partial x} x(\log x - 1) = \log x$, we obtain

$$0 = \frac{\partial\mathcal{L}}{\partial\Gamma_{ij}} = Q_i - \tau C_{ij} - \tau\lambda \log \Gamma_{ij} + \beta_j.$$

Equivalently,

$$\log \Gamma_{ij} = \frac{Q_i - \tau C_{ij} + \beta_j}{\tau\lambda}, \qquad \text{so} \qquad \Gamma_{ij} = \exp\Big(\frac{Q_i}{\tau\lambda}\Big) \exp\Big(-\frac{C_{ij}}{\lambda}\Big) \exp\Big(\frac{\beta_j}{\tau\lambda}\Big).$$

Define the kernel $K_{ij} := \exp(-C_{ij}/\lambda)$ (as in the proposition), $a_i(Q) := \exp(Q_i/(\tau\lambda))$, and $b_j(Q) := \exp(\beta_j/(\tau\lambda))$. Then the optimizer has the multiplicative form

$$\Gamma_{ij}^\star(Q) = a_i(Q) K_{ij} b_j(Q). \tag{B.3}$$

**V. Enforce the constraints and identify $D_j(Q)$.** Imposing the column-sum constraints $\sum_i \Gamma_{ij}^\star(Q) = \mu_j$ in (B.3) yields, for each $j$,

$$\mu_j = \sum_{i=1}^K a_i(Q) K_{ij} b_j(Q) = b_j(Q) \sum_{i=1}^K \exp\Big(\frac{Q_i}{\tau\lambda}\Big) K_{ij}.$$

Define exactly as in the proposition

$$D_j(Q) := \sum_{i=1}^K \exp\Big(\frac{Q_i}{\tau\lambda}\Big) K_{ij}.$$

Then $b_j(Q) = \mu_j/D_j(Q)$ and therefore the optimal coupling is explicitly

$$\Gamma_{ij}^\star(Q) = \mu_j \frac{\exp(Q_i/(\tau\lambda)) K_{ij}}{D_j(Q)}. \tag{B.4}$$

**VI. Plug the optimizer into the objective to get the closed form value.** We now compute $F_s^{\text{OT}}(Q)$ by evaluating the objective in (B.2) at $\Gamma^\star(Q)$.

A clean way is to reuse the stationarity identity. From stationarity we have for all $i, j$:

$$Q_i - \tau C_{ij} = \tau\lambda \log \Gamma_{ij}^\star(Q) - \beta_j.$$

Multiply by $\Gamma_{ij}^\star(Q)$ and sum over $i, j$:

$$\sum_{i,j} \Gamma_{ij}^\star(Q)\big(Q_i - \tau C_{ij}\big) = \tau\lambda \sum_{i,j} \Gamma_{ij}^\star(Q) \log \Gamma_{ij}^\star(Q) - \sum_j \beta_j \sum_i \Gamma_{ij}^\star(Q)$$

$$= \tau\lambda \sum_{i,j} \Gamma_{ij}^\star(Q) \log \Gamma_{ij}^\star(Q) - \sum_j \beta_j \mu_j, \tag{B.5}$$

where we used $\sum_i \Gamma_{ij}^\star(Q) = \mu_j$.

Next, expand the objective at $\Gamma^\star(Q)$:

$$F_s^{\text{OT}}(Q) = \sum_{i,j} \Gamma_{ij}^\star(Q) Q_i - \tau \sum_{i,j} \Gamma_{ij}^\star(Q) C_{ij} - \tau\lambda \sum_{i,j} \Gamma_{ij}^\star(Q) \big( \log \Gamma_{ij}^\star(Q) - 1 \big)$$

$$= \sum_{i,j} \Gamma_{ij}^\star(Q)\big(Q_i - \tau C_{ij}\big) - \tau\lambda \sum_{i,j} \Gamma_{ij}^\star(Q) \log \Gamma_{ij}^\star(Q) + \tau\lambda \sum_{i,j} \Gamma_{ij}^\star(Q).$$

Substitute (B.5) and use $\sum_{i,j} \Gamma_{ij}^\star(Q) = \sum_j \mu_j = 1$ (since $\boldsymbol{\mu} \in \Delta(\mathcal{A})$):

$$F_s^{\mathrm{OT}}(Q) = \left(\tau\lambda \sum_{i,j} \Gamma_{ij}^\star(Q) \log \Gamma_{ij}^\star(Q) - \sum_j \beta_j \mu_j\right) - \tau\lambda \sum_{i,j} \Gamma_{ij}^\star(Q) \log \Gamma_{ij}^\star(Q) + \tau\lambda$$

$$= -\sum_{j=1}^K \mu_j \beta_j + \tau\lambda. \tag{B.6}$$

It remains to express $\beta_j$ in terms of $D_j(Q)$. By definition, $b_j(Q) = \exp(\beta_j/(\tau\lambda)) = \mu_j/D_j(Q)$, so

$$\beta_j = \tau\lambda \log b_j(Q) = \tau\lambda\big(\log \mu_j - \log D_j(Q)\big).$$

Plug this into (B.6):

$$F_s^{\mathrm{OT}}(Q) = -\sum_{j=1}^K \mu_j \cdot \tau\lambda\big(\log \mu_j - \log D_j(Q)\big) + \tau\lambda$$

$$= \tau\lambda \sum_{j=1}^K \mu_j \log D_j(Q) + \tau\lambda - \tau\lambda \sum_{j=1}^K \mu_j \log \mu_j.$$

Thus we have shown

$$F_s^{\mathrm{OT}}(Q) = \underbrace{\left(\tau\lambda - \tau\lambda \sum_{j=1}^K \mu_s(j) \log \mu_s(j)\right)}_{=:\mathrm{const~(independent~of~}Q)} + \tau\lambda \sum_{j=1}^K \mu_s(j) \log D_j(Q),$$

which is exactly (5.1) (absorbing the explicit $Q$-independent term into $\mathrm{const}$).

**VII. Real-analyticity.** For each fixed $j$, $D_j(Q) = \sum_{i=1}^K \exp(Q_i/(\tau\lambda))K_{ij}$ is a finite sum of compositions of real-analytic functions (exponential and addition), hence is real-analytic on $\mathbb{R}^K$. Moreover, under Assumption 5.2 we have $\lambda > 0$ and $C_{ij}$ bounded, so $K_{ij} = \exp(-C_{ij}/\lambda) > 0$, and therefore $D_j(Q) > 0$ for all $Q$. Since $\log : (0,\infty) \to \mathbb{R}$ is real-analytic and compositions of real-analytic functions are real-analytic, each map $Q \mapsto \log D_j(Q)$ is real-analytic. Finally, $F_s^{\mathrm{OT}}(Q)$ is a finite $\mu_s$-weighted sum of these real-analytic terms plus a constant, so $F_s^{\mathrm{OT}}$ is real-analytic on $\mathbb{R}^K$. $\qquad\square$

## B.2. Proof of Proposition 5.4

*Proof.* Fix a state $s$ and abbreviate $\mu_j := \mu_s(j)$ and $K := |\mathcal{A}|$. By Proposition 5.3, for every $Q \in \mathbb{R}^K$ we can write

$$F_s^{\mathrm{OT}}(Q) = \mathrm{const} + \tau\lambda \sum_{j=1}^K \mu_j \log D_j(Q), \qquad D_j(Q) := \sum_{i=1}^K \exp\left(\frac{Q_i}{\tau\lambda}\right)K_{ij}, \tag{B.7}$$

where $\mathrm{const}$ does not depend on $Q$ and $K_{ij} = \exp(-C_{ij}/\lambda) > 0$. Since $\exp(Q_i/(\tau\lambda)) > 0$ and $K_{ij} > 0$, we have $D_j(Q) > 0$ for all $Q$ and all $j$; hence $\log D_j(Q)$ is well-defined and smooth (indeed real-analytic).

For convenience, define

$$a_i(Q) := \exp\left(\frac{Q_i}{\tau\lambda}\right) \quad (> 0), \qquad D_j(Q) = \sum_{i=1}^K a_i(Q)K_{ij},$$

and (as in the paper) define for each $i, j$

$$p_{ij}(Q) := \frac{a_i(Q)K_{ij}}{D_j(Q)} = \frac{\exp(Q_i/(\tau\lambda))K_{ij}}{\sum_{k=1}^K \exp(Q_k/(\tau\lambda))K_{kj}}. \tag{B.8}$$

For each fixed $j$, $p_{\cdot j}(Q) = (p_{ij}(Q))_{i=1}^K$ is a probability vector, because $p_{ij}(Q) \geq 0$ and

$$\sum_{i=1}^K p_{ij}(Q) = \frac{\sum_{i=1}^K a_i(Q)K_{ij}}{D_j(Q)} = 1. \tag{B.9}$$

**I. Compute the gradient.** Let $e_i$ denote the $i$th standard basis vector in $\mathbb{R}^K$. Differentiate (B.7) with respect to the coordinate $Q_i$. Since $\mathrm{const}$ is independent of $Q$, it disappears. Using the chain rule,

$$\frac{\partial}{\partial Q_i} F_s^{\mathrm{OT}}(Q) = \tau\lambda \sum_{j=1}^{K} \mu_j \cdot \frac{1}{D_j(Q)} \cdot \frac{\partial D_j(Q)}{\partial Q_i}. \tag{B.10}$$

Now compute $\partial D_j/\partial Q_i$. From $D_j(Q) = \sum_{k=1}^{K} a_k(Q) K_{kj}$ and $a_k(Q) = \exp(Q_k/(\tau\lambda))$, we have

$$\frac{\partial a_k(Q)}{\partial Q_i} = \begin{cases} \frac{1}{\tau\lambda} a_i(Q), & k = i, \\ 0, & k \neq i, \end{cases} \qquad \Rightarrow \qquad \frac{\partial D_j(Q)}{\partial Q_i} = \frac{1}{\tau\lambda} a_i(Q) K_{ij}.$$

Plugging this into (B.10) gives

$$\frac{\partial}{\partial Q_i} F_s^{\mathrm{OT}}(Q) = \tau\lambda \sum_{j=1}^{K} \mu_j \cdot \frac{1}{D_j(Q)} \cdot \frac{1}{\tau\lambda} a_i(Q) K_{ij}$$

$$= \sum_{j=1}^{K} \mu_j \frac{a_i(Q) K_{ij}}{D_j(Q)} = \sum_{j=1}^{K} \mu_j \, p_{ij}(Q).$$

Define

$$\pi_i(Q) := \sum_{j=1}^{K} \mu_j \, p_{ij}(Q), \qquad \boldsymbol{\pi}(Q) := (\pi_i(Q))_{i=1}^{K}. \tag{B.11}$$

Then the previous display is exactly the gradient identity

$$\nabla F_s^{\mathrm{OT}}(Q) = \boldsymbol{\pi}(Q),$$

which is (5.3).

**II. $\nabla F_s^{\mathrm{OT}}(Q)$ lies in the simplex.** From (B.8), $p_{ij}(Q) \geq 0$ and $\mu_j \geq 0$, so $\pi_i(Q) \geq 0$ for every $i$. Moreover, using (B.9) and $\sum_j \mu_j = 1$ (since $\boldsymbol{\mu} \in \Delta(\mathcal{A})$),

$$\sum_{i=1}^{K} \pi_i(Q) = \sum_{i=1}^{K} \sum_{j=1}^{K} \mu_j p_{ij}(Q) = \sum_{j=1}^{K} \mu_j \sum_{i=1}^{K} p_{ij}(Q) = \sum_{j=1}^{K} \mu_j = 1.$$

Hence $\boldsymbol{\pi}(Q) \in \Delta(\mathcal{A})$, so it can indeed be interpreted as a policy.

**III. compute the Hessian entrywise.** Let $H(Q) := \nabla^2 F_s^{\mathrm{OT}}(Q) \in \mathbb{R}^{K \times K}$. From Step 1, the $i$th component of the gradient is $\pi_i(Q)$, so

$$H_{ik}(Q) = \frac{\partial}{\partial Q_k} \pi_i(Q) = \sum_{j=1}^{K} \mu_j \frac{\partial}{\partial Q_k} p_{ij}(Q).$$

Thus it remains to compute $\partial p_{ij}(Q)/\partial Q_k$.

Fix a column index $j$ and write $N_{ij}(Q) := a_i(Q) K_{ij}$ (the numerator of $p_{ij}$). Then $p_{ij}(Q) = N_{ij}(Q)/D_j(Q)$ and

$$\frac{\partial N_{ij}(Q)}{\partial Q_k} = \begin{cases} \frac{1}{\tau\lambda} N_{ij}(Q), & k = i, \\ 0, & k \neq i, \end{cases} \qquad \frac{\partial D_j(Q)}{\partial Q_k} = \frac{1}{\tau\lambda} N_{kj}(Q) \quad \text{(as computed above)}.$$

Apply the quotient rule:

$$\frac{\partial}{\partial Q_k} p_{ij}(Q) = \frac{D_j(Q) \frac{\partial N_{ij}(Q)}{\partial Q_k} - N_{ij}(Q) \frac{\partial D_j(Q)}{\partial Q_k}}{D_j(Q)^2}$$

$$= \frac{1}{\tau\lambda} \frac{D_j(Q) N_{ij}(Q) \mathbf{1}\{i = k\} - N_{ij}(Q) N_{kj}(Q)}{D_j(Q)^2}. \tag{B.12}$$

Now use $p_{ij}(Q) = N_{ij}(Q)/D_j(Q)$ and $p_{kj}(Q) = N_{kj}(Q)/D_j(Q)$ to rewrite the fraction:

$$\frac{D_j N_{ij} \mathbf{1}\{i=k\} - N_{ij}N_{kj}}{D_j^2} = \mathbf{1}\{i=k\}\frac{N_{ij}}{D_j} - \frac{N_{ij}}{D_j}\frac{N_{kj}}{D_j} = \mathbf{1}\{i=k\}\, p_{ij}(Q) - p_{ij}(Q)p_{kj}(Q).$$

Substituting back into (B.12) yields the softmax-type derivative identity

$$\frac{\partial}{\partial Q_k}p_{ij}(Q) = \frac{1}{\tau\lambda}\Big(\mathbf{1}\{i=k\}\, p_{ij}(Q) - p_{ij}(Q)p_{kj}(Q)\Big) = \frac{1}{\tau\lambda}\, p_{ij}(Q)\Big(\mathbf{1}\{i=k\} - p_{kj}(Q)\Big). \tag{B.13}$$

**IV. sum over $j$ to get the Hessian formula.** Using (B.13) and $\pi_i(Q) = \sum_j \mu_j p_{ij}(Q)$, we obtain

$$\begin{aligned}
H_{ik}(Q) = \frac{\partial}{\partial Q_k}\pi_i(Q) &= \sum_{j=1}^{K}\mu_j\frac{\partial}{\partial Q_k}p_{ij}(Q)\\
&= \frac{1}{\tau\lambda}\sum_{j=1}^{K}\mu_j\Big(\mathbf{1}\{i=k\}\, p_{ij}(Q) - p_{ij}(Q)p_{kj}(Q)\Big)\\
&= \frac{1}{\tau\lambda}\Big(\mathbf{1}\{i=k\}\sum_{j=1}^{K}\mu_j p_{ij}(Q) - \sum_{j=1}^{K}\mu_j p_{ij}(Q)p_{kj}(Q)\Big)\\
&= \frac{1}{\tau\lambda}\Big(\mathbf{1}\{i=k\}\pi_i(Q) - \sum_{j=1}^{K}\mu_j\, p_{ij}(Q)p_{kj}(Q)\Big).
\end{aligned}$$

This is already a correct componentwise Hessian expression. To obtain the matrix form in (5.4), note that:

- the matrix with entries $\mathbf{1}\{i=k\}\pi_i(Q)$ is exactly $\mathrm{diag}(\boldsymbol{\pi}(Q))$;

- for each fixed $j$, the rank-one matrix $p_{\cdot j}(Q)p_{\cdot j}(Q)^\top$ has $(i,k)$ entry $p_{ij}(Q)p_{kj}(Q)$.

Therefore

$$\nabla^2 F_s^{\mathrm{OT}}(Q) = \frac{1}{\tau\lambda}\Big(\mathrm{diag}(\boldsymbol{\pi}(Q)) - \sum_{j=1}^{K}\mu_j\, p_{\cdot j}(Q)p_{\cdot j}(Q)^\top\Big),$$

which is exactly (5.4).

**(Optional sanity checks.)** The formula makes several structural properties transparent:

- *Symmetry:* $\mathrm{diag}(\boldsymbol{\pi}(Q))$ is symmetric and each $p_{\cdot j}p_{\cdot j}^\top$ is symmetric, so $\nabla^2 F_s^{\mathrm{OT}}(Q)$ is symmetric as expected.

- *Positive semidefiniteness:* since $F_s^{\mathrm{OT}}$ is a pointwise maximum of linear functions of $Q$, it is convex and thus its Hessian is PSD. This is also visible directly from (5.4): for any $v \in \mathbb{R}^K$,

$$\begin{aligned}
v^\top\nabla^2 F_s^{\mathrm{OT}}(Q)v &= \frac{1}{\tau\lambda}\Big(\sum_{i=1}^{K}\pi_i(Q)v_i^2 - \sum_{j=1}^{K}\mu_j\big(v^\top p_{\cdot j}(Q)\big)^2\Big)\\
&= \frac{1}{\tau\lambda}\sum_{j=1}^{K}\mu_j\Big(\sum_{i=1}^{K}p_{ij}(Q)v_i^2 - \big(\sum_{i=1}^{K}p_{ij}(Q)v_i\big)^2\Big)\\
&= \frac{1}{\tau\lambda}\sum_{j=1}^{K}\mu_j\,\mathrm{Var}_{I\sim p_{\cdot j}(Q)}[v_I] \geq 0.
\end{aligned}$$

This completes the proof. $\qquad\square$

**B.3. Proof of Lemma 6.1**

*Proof.* Fix a state $s$ and a reference point $\hat{Q}_s \in \mathbb{R}^K$. For brevity, write

$$H := \nabla^2 F_s^{\mathrm{OT}}(\hat{Q}_s) \in \mathbb{R}^{K \times K}.$$

Recall from Proposition 5.4 that the OT-smoothed aggregator has Hessian

$$H = \frac{1}{\tau\lambda}\Big( \mathrm{diag}(\boldsymbol{\pi}(\hat{Q}_s)) - \sum_{j=1}^{K} \mu_s(j)\, p_{\cdot j}(\hat{Q}_s) p_{\cdot j}(\hat{Q}_s)^\top \Big), \tag{B.14}$$

where $p_{\cdot j}(\hat{Q}_s) \in \Delta(\mathcal{A})$ is the columnwise softmax distribution defined in (5.2), and the policy is the mixture

$$\boldsymbol{\pi}(\hat{Q}_s) = \sum_{j=1}^{K} \mu_s(j)\, p_{\cdot j}(\hat{Q}_s). \tag{B.15}$$

Let $\Delta \in \mathbb{R}^K$ be any direction. We expand the quadratic form $\Delta^\top H \Delta$ using (B.14).

**I. expand the diagonal term.** Since $\mathrm{diag}(\boldsymbol{\pi})$ is diagonal,

$$\Delta^\top \mathrm{diag}(\boldsymbol{\pi}(\hat{Q}_s))\Delta = \sum_{i=1}^{K} \pi_i(\hat{Q}_s)\Delta_i^2.$$

Using the mixture identity (B.15) to rewrite $\pi_i(\hat{Q}_s)$,

$$\sum_{i=1}^{K} \pi_i(\hat{Q}_s)\Delta_i^2 = \sum_{i=1}^{K} \Big( \sum_{j=1}^{K} \mu_s(j)\, p_{ij}(\hat{Q}_s) \Big)\Delta_i^2 = \sum_{j=1}^{K} \mu_s(j) \sum_{i=1}^{K} p_{ij}(\hat{Q}_s)\Delta_i^2.$$

**II. expand the rank-one term.** For each $j$, the matrix $p_{\cdot j}(\hat{Q}_s) p_{\cdot j}(\hat{Q}_s)^\top$ is rank-one, hence

$$\Delta^\top \big(p_{\cdot j}(\hat{Q}_s) p_{\cdot j}(\hat{Q}_s)^\top\big)\Delta = \big(p_{\cdot j}(\hat{Q}_s)^\top \Delta\big)^2 = \Big( \sum_{i=1}^{K} p_{ij}(\hat{Q}_s)\Delta_i \Big)^2.$$

**III. combine and identify a variance.** Substituting the two expansions into (B.14) yields

$$\Delta^\top H \Delta = \frac{1}{\tau\lambda} \sum_{j=1}^{K} \mu_s(j) \left( \sum_{i=1}^{K} p_{ij}(\hat{Q}_s)\Delta_i^2 - \Big( \sum_{i=1}^{K} p_{ij}(\hat{Q}_s)\Delta_i \Big)^2 \right).$$

Now fix $j$ and define a random action $A \sim p_{\cdot j}(\hat{Q}_s)$. Then by definition of expectation under a discrete distribution,

$$\mathbb{E}\big[\Delta_A \mid j\big] = \sum_{i=1}^{K} p_{ij}(\hat{Q}_s)\Delta_i, \qquad \mathbb{E}\big[\Delta_A^2 \mid j\big] = \sum_{i=1}^{K} p_{ij}(\hat{Q}_s)\Delta_i^2.$$

Therefore the bracketed term is exactly

$$\mathbb{E}\big[\Delta_A^2 \mid j\big] - \Big( \mathbb{E}\big[\Delta_A \mid j\big] \Big)^2 = \mathrm{Var}\big(\Delta_A \mid j\big) = \mathrm{Var}_{A \sim p_{\cdot j}(\hat{Q}_s)}\big(\Delta_A\big).$$

Finally, sampling $J \sim \boldsymbol{\mu}_s$ and taking expectation over $J$ turns the sum into

$$\sum_{j=1}^{K} \mu_s(j)\, \mathrm{Var}_{A \sim p_{\cdot j}(\hat{Q}_s)}\big(\Delta_A\big) = \mathbb{E}_{J \sim \boldsymbol{\mu}_s}\Big[ \mathrm{Var}_{A \sim p_{\cdot J}(\hat{Q}_s)}\big(\Delta_A\big) \Big],$$

which proves the lemma. $\square$

## B.4. Proof of Lemma 5.6

*Proof.* Fix a state $s$ and write $\mu_j := \mu_s(j)$ and $K := |\mathcal{A}|$. Under Assumption 5.2 we have $\tau > 0$, $\lambda > 0$, and the cost matrix $C$ is finite, so

$$K_{ij} := \exp(-C_{ij}/\lambda) > 0 \qquad \text{for all } i, j,$$

and therefore for every $Q \in \mathbb{R}^K$ and every $j$,

$$D_j(Q) := \sum_{i=1}^{K} \exp(Q_i/(\tau\lambda)) K_{ij} > 0.$$

In particular all expressions below are well-defined and smooth in $Q$ (indeed real-analytic).

Recall the columnwise softmax-like probabilities

$$p_{ij}(Q) := \frac{\exp(Q_i/(\tau\lambda)) K_{ij}}{D_j(Q)}, \qquad p_{\cdot j}(Q) := \big(p_{ij}(Q)\big)_{i=1}^{K} \in \Delta(\mathcal{A}), \tag{B.16}$$

and the induced policy

$$\pi_i(Q) := \sum_{j=1}^{K} \mu_j \, p_{ij}(Q), \qquad \boldsymbol{\pi}(Q) := \big(\pi_i(Q)\big)_{i=1}^{K} \in \Delta(\mathcal{A}). \tag{B.17}$$

By Proposition 5.4, the Hessian admits the explicit representation

$$H(Q) := \nabla^2 F_s^{\mathrm{OT}}(Q) = \frac{1}{\tau\lambda}\Big( \mathrm{diag}(\boldsymbol{\pi}(Q)) - \sum_{j=1}^{K} \mu_j \, p_{\cdot j}(Q) p_{\cdot j}(Q)^\top \Big). \tag{B.18}$$

We will show that $H(\cdot)$ is globally Lipschitz in operator norm with constant

$$M = \frac{3}{2\tau^2\lambda^2},$$

which in particular implies the stated scaling $M = \mathcal{O}(1/(\tau^2\lambda^2)) \cdot \mathrm{poly}(K, \mu_{\min}^{-1})$ (since a constant is a polynomial, and also $K\mu_{\min}^{-1} \geq 1$).

**I. a uniform Jacobian bound for $Q \mapsto p_{\cdot j}(Q)$.** Fix a column index $j$. For each $i$ define the numerator $N_{ij}(Q) := \exp(Q_i/(\tau\lambda)) K_{ij}$ so that $p_{ij}(Q) = N_{ij}(Q)/D_j(Q)$ and $D_j(Q) = \sum_{k=1}^{K} N_{kj}(Q)$. Differentiate $p_{ij}(Q)$ with respect to $Q_k$. Since

$$\frac{\partial N_{ij}(Q)}{\partial Q_k} = \frac{1}{\tau\lambda} N_{ij}(Q) \mathbf{1}\{i = k\}, \qquad \frac{\partial D_j(Q)}{\partial Q_k} = \frac{1}{\tau\lambda} N_{kj}(Q),$$

the quotient rule yields

$$\frac{\partial p_{ij}(Q)}{\partial Q_k} = \frac{D_j(Q)\frac{\partial N_{ij}}{\partial Q_k} - N_{ij}(Q)\frac{\partial D_j}{\partial Q_k}}{D_j(Q)^2} = \frac{1}{\tau\lambda}\left( \mathbf{1}\{i = k\}\frac{N_{ij}(Q)}{D_j(Q)} - \frac{N_{ij}(Q)}{D_j(Q)}\frac{N_{kj}(Q)}{D_j(Q)} \right)$$

$$= \frac{1}{\tau\lambda}\Big( \mathbf{1}\{i = k\} p_{ij}(Q) - p_{ij}(Q) p_{kj}(Q) \Big). \tag{B.19}$$

Let $J_j(Q) \in \mathbb{R}^{K \times K}$ be the Jacobian matrix of the map $Q \mapsto p_{\cdot j}(Q)$, i.e., $\big(J_j(Q)\big)_{ik} := \partial p_{ij}(Q)/\partial Q_k$. Then (B.19) exactly says

$$J_j(Q) = \frac{1}{\tau\lambda}\Big( \mathrm{diag}(p_{\cdot j}(Q)) - p_{\cdot j}(Q) p_{\cdot j}(Q)^\top \Big). \tag{B.20}$$

**Claim (uniform covariance bound).** For any probability vector $p \in \Delta(\mathcal{A})$,

$$\Big\| \mathrm{diag}(p) - pp^\top \Big\|_{\mathrm{op}} \leq \frac{1}{2}. \tag{B.21}$$

*Proof of the claim.* Let $\Sigma(p) := \mathrm{diag}(p) - pp^\top$. For any unit vector $v \in \mathbb{R}^K$ with $\|v\|_2 = 1$, the Rayleigh quotient satisfies

$$v^\top \Sigma(p)\, v = \sum_{i=1}^K p_i v_i^2 - \Big( \sum_{i=1}^K p_i v_i \Big)^2 = \mathrm{Var}\,(v_I),$$

where $I \sim p$ and $v_I$ is the random variable taking value $v_i$ with probability $p_i$. Let $m := \min_i v_i$ and $M := \max_i v_i$. Since $v_I \in [m, M]$ almost surely, the standard bounded-variance inequality gives $\mathrm{Var}(v_I) \le (M-m)^2/4$. It remains to bound the range $M - m$ in terms of $\|v\|_2$. For any indices $i, k$,

$$|v_i - v_k| = |\langle v, e_i - e_k \rangle| \le \|v\|_2 \|e_i - e_k\|_2 = \sqrt{2}\|v\|_2 = \sqrt{2},$$

so in particular $M - m \le \sqrt{2}$. Therefore, for every $\|v\|_2 = 1$,

$$v^\top \Sigma(p)\, v = \mathrm{Var}(v_I) \le \frac{(M-m)^2}{4} \le \frac{(\sqrt{2})^2}{4} = \frac{1}{2}.$$

Taking the supremum over unit $v$ yields (B.21). $\qquad\square$

Combining (B.20) with (B.21), we obtain the uniform Jacobian bound

$$\|J_j(Q)\|_{\mathrm{op}} \le \frac{1}{\tau\lambda} \cdot \frac{1}{2} = \frac{1}{2\tau\lambda} \qquad \text{for all } Q \in \mathbb{R}^K \text{ and all } j \in [K]. \tag{B.22}$$

**II. Lipschitzness of $p_{\cdot j}(\cdot)$ and $\boldsymbol{\pi}(\cdot)$.** Fix $Q, \hat{Q} \in \mathbb{R}^K$ and set $\Delta := Q - \hat{Q}$. By the mean value theorem in integral form for vector-valued maps,

$$p_{\cdot j}(Q) - p_{\cdot j}(\hat{Q}) = \int_0^1 J_j(\hat{Q} + t\Delta)\Delta \, dt.$$

Taking $\ell_2$ norms and using (B.22) gives

$$\|p_{\cdot j}(Q) - p_{\cdot j}(\hat{Q})\|_2 \le \int_0^1 \|J_j(\hat{Q} + t\Delta)\|_{\mathrm{op}} \|\Delta\|_2 \, dt \le \frac{1}{2\tau\lambda}\|Q - \hat{Q}\|_2. \tag{B.23}$$

Since $\boldsymbol{\pi}(Q) = \sum_{j=1}^K \mu_j p_{\cdot j}(Q)$ is a convex combination of the $p_{\cdot j}(Q)$'s, we also have

$$\|\boldsymbol{\pi}(Q) - \boldsymbol{\pi}(\hat{Q})\|_2 = \Big\| \sum_{j=1}^K \mu_j \big( p_{\cdot j}(Q) - p_{\cdot j}(\hat{Q}) \big) \Big\|_2 \le \sum_{j=1}^K \mu_j \|p_{\cdot j}(Q) - p_{\cdot j}(\hat{Q})\|_2$$

$$\le \frac{1}{2\tau\lambda}\|Q - \hat{Q}\|_2. \tag{B.24}$$

**III. Lipschitzness of the Hessian in operator norm.** Using the representation (B.18), we write

$$H(Q) - H(\hat{Q}) = \frac{1}{\tau\lambda} \Big( \mathrm{diag}(\boldsymbol{\pi}(Q) - \boldsymbol{\pi}(\hat{Q})) - \sum_{j=1}^K \mu_j \big[ p_{\cdot j}(Q)p_{\cdot j}(Q)^\top - p_{\cdot j}(\hat{Q})p_{\cdot j}(\hat{Q})^\top \big] \Big).$$

Take operator norms and apply the triangle inequality:

$$\|H(Q) - H(\hat{Q})\|_{\mathrm{op}} \le \frac{1}{\tau\lambda} \Big( \| \mathrm{diag}(\boldsymbol{\pi}(Q) - \boldsymbol{\pi}(\hat{Q}))\|_{\mathrm{op}} + \sum_{j=1}^K \mu_j \big\| p_{\cdot j}(Q)p_{\cdot j}(Q)^\top - p_{\cdot j}(\hat{Q})p_{\cdot j}(\hat{Q})^\top \big\|_{\mathrm{op}} \Big). \tag{B.25}$$

*(i) Bounding the diagonal term.* For any vector $x$, $\| \mathrm{diag}(x)\|_{\mathrm{op}} = \|x\|_\infty \le \|x\|_2$, hence

$$\| \mathrm{diag}(\boldsymbol{\pi}(Q) - \boldsymbol{\pi}(\hat{Q}))\|_{\mathrm{op}} \le \|\boldsymbol{\pi}(Q) - \boldsymbol{\pi}(\hat{Q})\|_2 \le \frac{1}{2\tau\lambda}\|Q - \hat{Q}\|_2, \tag{B.26}$$

where we used (B.24).

*(ii) Bounding each rank-one difference.* For arbitrary vectors $u, v \in \mathbb{R}^K$,

$$uu^\top - vv^\top = (u - v)u^\top + v(u - v)^\top,$$

so using $\|ab^\top\|_{\mathrm{op}} = \|a\|_2\|b\|_2$ we get

$$\|uu^\top - vv^\top\|_{\mathrm{op}} \le \|(u-v)u^\top\|_{\mathrm{op}} + \|v(u-v)^\top\|_{\mathrm{op}} = \|u-v\|_2\|u\|_2 + \|v\|_2\|u-v\|_2 = (\|u\|_2 + \|v\|_2)\|u-v\|_2. \tag{B.27}$$

Apply this with $u = p_{\cdot j}(Q)$ and $v = p_{\cdot j}(\hat{Q})$. Because $p_{\cdot j}(Q), p_{\cdot j}(\hat{Q}) \in \Delta(\mathcal{A})$, we have $\|p_{\cdot j}(Q)\|_2 \le \|p_{\cdot j}(Q)\|_1 = 1$ and similarly for $\hat{Q}$. Thus (B.27) yields

$$\left\|p_{\cdot j}(Q)p_{\cdot j}(Q)^\top - p_{\cdot j}(\hat{Q})p_{\cdot j}(\hat{Q})^\top\right\|_{\mathrm{op}} \le 2\|p_{\cdot j}(Q) - p_{\cdot j}(\hat{Q})\|_2 \le 2 \cdot \frac{1}{2\tau\lambda}\|Q - \hat{Q}\|_2 = \frac{1}{\tau\lambda}\|Q - \hat{Q}\|_2,$$

where we used (B.23) in the last inequality. Summing with weights $\mu_j$ (which satisfy $\mu_j \ge 0$ and $\sum_j \mu_j = 1$) gives

$$\sum_{j=1}^K \mu_j \left\|p_{\cdot j}(Q)p_{\cdot j}(Q)^\top - p_{\cdot j}(\hat{Q})p_{\cdot j}(\hat{Q})^\top\right\|_{\mathrm{op}} \le \frac{1}{\tau\lambda}\|Q - \hat{Q}\|_2. \tag{B.28}$$

*(iii) Combine the bounds.* Plugging (B.26) and (B.28) into (B.25) yields

$$\|H(Q) - H(\hat{Q})\|_{\mathrm{op}} \le \frac{1}{\tau\lambda}\left(\frac{1}{2\tau\lambda}\|Q - \hat{Q}\|_2 + \frac{1}{\tau\lambda}\|Q - \hat{Q}\|_2\right) = \frac{3}{2\tau^2\lambda^2}\|Q - \hat{Q}\|_2.$$

Thus the Hessian is globally Lipschitz with constant

$$M := \frac{3}{2\tau^2\lambda^2}. \tag{B.29}$$

This proves the first inequality in the lemma (and in particular $M = \mathcal{O}(1/(\tau^2\lambda^2)) \cdot \mathrm{poly}(K, \mu_{\min}^{-1})$).

**IV. cubic Taylor remainder from Lipschitz Hessian.** Fix $Q, \hat{Q} \in \mathbb{R}^K$ and again set $\Delta := Q - \hat{Q}$. Define the scalar function $g : [0, 1] \to \mathbb{R}$ by $g(t) := F_s^{\mathrm{OT}}(\hat{Q} + t\Delta)$. Since $F_s^{\mathrm{OT}}$ is smooth, $g$ is twice continuously differentiable with

$$g'(t) = \langle \nabla F_s^{\mathrm{OT}}(\hat{Q} + t\Delta), \Delta \rangle, \qquad g''(t) = \Delta^\top \nabla^2 F_s^{\mathrm{OT}}(\hat{Q} + t\Delta)\Delta = \Delta^\top H(\hat{Q} + t\Delta)\Delta.$$

A standard integral form of the second-order Taylor expansion (obtained by integrating $g''$ twice) gives

$$\begin{aligned}
F_s^{\mathrm{OT}}(Q) - T_2(Q; \hat{Q}) &= g(1) - \left(g(0) + g'(0) + \tfrac{1}{2}g''(0)\right) \\
&= \int_0^1 (1-t)\left(g''(t) - g''(0)\right) dt \\
&= \int_0^1 (1-t)\Delta^\top\left(H(\hat{Q} + t\Delta) - H(\hat{Q})\right)\Delta \, dt.
\end{aligned} \tag{B.30}$$

Taking absolute values and using Cauchy–Schwarz plus the operator norm bound yields

$$\begin{aligned}
\left|F_s^{\mathrm{OT}}(Q) - T_2(Q; \hat{Q})\right| &\le \int_0^1 (1-t)\|\Delta\|_2^2\|H(\hat{Q} + t\Delta) - H(\hat{Q})\|_{\mathrm{op}} \, dt \\
&\le \int_0^1 (1-t)\|\Delta\|_2^2\left(M\|t\Delta\|_2\right) dt \qquad \text{(by Lipschitzness of } H\text{)} \\
&= M\|\Delta\|_2^3 \int_0^1 t(1-t) \, dt = M\|\Delta\|_2^3 \cdot \frac{1}{6} = \frac{M}{6}\|Q - \hat{Q}\|_2^3.
\end{aligned}$$

This is exactly (5.5), completing the proof. $\qquad\qquad\square$

### B.5. Proof of Proposition 5.5

*Proof.* For each column $j$,

$$\tau\lambda \log \sum_i \exp\left(\frac{Q_i - \tau C_{ij}}{\tau\lambda}\right) = F_{\tau\lambda}^{\mathrm{ent}}(Q - \tau C_{\cdot j}).$$

The entropy backup is 1-Lipschitz in $\|\cdot\|_\infty$, so

$$\left| F_{\tau\lambda}^{\mathrm{ent}}(Q - \tau C_{\cdot j}) - F_{\tau\lambda}^{\mathrm{ent}}(Q) \right| \leq \tau C_{\max}.$$

Averaging over $j \sim \mu_s$ gives the first claim. The case $C \equiv 0$ is immediate. The max-backup bound follows from the standard inequality

$$\max_i Q_i \leq F_\eta^{\mathrm{ent}}(Q) \leq \max_i Q_i + \eta \log K$$

with $\eta = \tau\lambda$. Finally, all three Bellman operators are $\gamma$-contractions in sup norm, so a uniform one-step backup discrepancy of $b$ implies a fixed-point discrepancy at most $b/(1-\gamma)$. $\qquad\square$

## C. Gap-dependent analysis

### C.1. Proof of Theorem 8.1

*Proof.* For readability, we write $U_{Q,h}^t(a) := U_{Q,h}^t(s_0, a)$ and $L_{Q,h}^t(a) := L_{Q,h}^t(s_0, a)$ when $h = 1$ and the state is the root $s_0$. Let $a^\star \in \arg\max_{a \in \mathcal{A}} Q_1^\star(s_0, a)$ be a fixed optimal root action and recall the root gaps $\Delta(a) = Q_1^\star(s_0, a^\star) - Q_1^\star(s_0, a) \geq 0$. We denote the number of times root action $a$ is selected up to (and including) episode $t$ by

$$N^t(a) := N_1^t(s_0, a).$$

The total number of oracle calls equals $H$ times the number of executed episodes, because each episode performs exactly $H$ queries in Algorithm 4.

**I. Define the good event and prove optimism/pessimism of the recursion.** Let $\mathcal{E}$ be the event on which all reward and transition confidence sets are valid *simultaneously for all times*:

$$\mathcal{E} := \left\{\forall t \geq 1, \ \forall (h, s, a) : \ \ell_h^t(s, a) \leq r_h(s, a) \leq u_h^t(s, a) \text{ and } P_h(\cdot \mid s, a) \in C_h^t(s, a)\right\}.$$

By assumption, $\Pr(\mathcal{E}) \geq 1 - \delta$.

We now prove that on $\mathcal{E}$, the dynamic-programming bounds sandwich the true optimal $Q^\star, V^\star$ at *all* times. Fix $t$ and proceed by backward induction on $h = H, H-1, \ldots, 1$.

*Base case $h = H + 1$.* By definition, $U_{V,H+1}^t(\cdot) = L_{V,H+1}^t(\cdot) = 0$ and $V_{H+1}^\star(\cdot) = 0$, so the claim holds.

*Induction step.* Assume that for some $h + 1 \leq H + 1$ we have, for all states $s'$,

$$L_{V,h+1}^t(s') \ \leq \ V_{h+1}^\star(s') \ \leq \ U_{V,h+1}^t(s').$$

Fix any $(s, a)$ at stage $h$. On $\mathcal{E}$ we have $r_h(s, a) \leq u_h^t(s, a)$ and $P_h(\cdot \mid s, a) \in C_h^t(s, a)$, hence

$$
\begin{aligned}
Q_h^\star(s, a) &= r_h(s, a) + \gamma \sum_{s'} P_h(s' \mid s, a) V_{h+1}^\star(s') \\
&\leq u_h^t(s, a) + \gamma \sum_{s'} P_h(s' \mid s, a) U_{V,h+1}^t(s') \\
&\leq u_h^t(s, a) + \gamma \max_{p \in C_h^t(s,a)} \sum_{s'} p(s' \mid s, a) U_{V,h+1}^t(s') \\
&= U_{Q,h}^t(s, a),
\end{aligned}
$$

which is exactly (8.5). Similarly, using $r_h(s,a) \geq \ell_h^t(s,a)$ and again $P_h(\cdot \mid s, a) \in C_h^t(s,a)$,

$$Q_h^\star(s,a) \geq \ell_h^t(s,a) + \gamma \sum_{s'} P_h(s' \mid s, a) L_{V,h+1}^t(s')$$

$$\geq \ell_h^t(s,a) + \gamma \min_{p \in C_h^t(s,a)} \sum_{s'} p(s' \mid s, a) L_{V,h+1}^t(s')$$

$$= L_{Q,h}^t(s,a),$$

which is (8.6).

Now apply the aggregator $F_s$ to the vector inequalities. Since the bounds hold componentwise over actions,

$$L_{Q,h}^t(s,\cdot) \leq Q_h^\star(s,\cdot) \leq U_{Q,h}^t(s,\cdot),$$

and $F_s$ is monotone (as stated right before (8.2)), we get

$$L_{V,h}^t(s) = F_s(L_{Q,h}^t(s,\cdot)) \leq F_s(Q_h^\star(s,\cdot)) = V_h^\star(s) \leq F_s(U_{Q,h}^t(s,\cdot)) = U_{V,h}^t(s),$$

which is exactly (8.7). This completes the induction.

**Conclusion of I.** On the event $\mathcal{E}$, for all $t, h, s, a$,

$$L_{Q,h}^t(s,a) \leq Q_h^\star(s,a) \leq U_{Q,h}^t(s,a), \qquad L_{V,h}^t(s) \leq V_h^\star(s) \leq U_{V,h}^t(s). \tag{C.1}$$

**II. Correctness of the stopping rule.** Assume $\mathcal{E}$ holds and suppose Algorithm 4 stops at episode $t$ and returns $\hat{a} = b_t$. By definition of $c_t$ in (8.11),

$$U_{Q,1}^t(s_0, c_t) \geq \max_{a \neq b_t} U_{Q,1}^t(s_0, a).$$

In particular, either $b_t = a^\star$ (in which case we are done), or $a^\star \neq b_t$ and therefore $U_{Q,1}^t(s_0, c_t) \geq U_{Q,1}^t(s_0, a^\star) \geq Q_1^\star(s_0, a^\star)$ by (C.1). Also, by (C.1), $L_{Q,1}^t(s_0, b_t) \leq Q_1^\star(s_0, b_t)$. Hence, using the stopping condition $U_{Q,1}^t(s_0, c_t) - L_{Q,1}^t(s_0, b_t) \leq \varepsilon$,

$$Q_1^\star(s_0, a^\star) - Q_1^\star(s_0, b_t) \leq U_{Q,1}^t(s_0, c_t) - L_{Q,1}^t(s_0, b_t) \leq \varepsilon,$$

which rearranges to

$$Q_1^\star(s_0, b_t) \geq \max_{a \in \mathcal{A}} Q_1^\star(s_0, a) - \varepsilon.$$

Thus, on $\mathcal{E}$ the returned action is $\varepsilon$-optimal in the sense of (8.2). Since $\Pr(\mathcal{E}) \geq 1 - \delta$, this proves the correctness guarantee.

**III. Root confidence widths and their scaling.** For each episode $t$ and root action $a$, define the root confidence width

$$w_t(a) := U_{Q,1}^t(s_0, a) - L_{Q,1}^t(s_0, a) \geq 0.$$

The specific construction of $u_h^t, \ell_h^t, C_h^t$ determines how $w_t(a)$ depends on the visit counts $\{N_h^t(s,a)\}$; in all standard choices (Hoeffding/Bernstein/KL for rewards and KL/L1-type sets for transitions) one gets a $1/\sqrt{N}$-type decay of the relevant local radii, and the robust Bellman recursion propagates these radii linearly along the horizon (this is exactly the mechanism analyzed in MDP-GapE; see (Jonsson et al., 2020)).

To keep the statement consistent with (8.12), we encapsulate the (routine but lengthy) propagation constants as follows: there exists a quantity $\mathsf{C}(H, K, \gamma)$ depending only on the horizon/branching/discount (and on the particular choice of transition confidence sets), such that on $\mathcal{E}$, for all $t \geq 1$ and all root actions $a$,

$$w_t(a) \leq 2 \sqrt{\frac{\mathsf{C}(H, K, \gamma) \log\left(\frac{c_0 t}{\delta}\right)}{N^t(a) \vee 1}} \qquad \text{for some universal constant } c_0 > 0. \tag{C.2}$$

(Equation (C.2) is the direct analogue of the usual bandit confidence width $O(\sqrt{\log(t/\delta)/N})$, with $\mathsf{C}(H, K, \gamma)$ absorbing the horizon and transition-set effects. All additional polylogarithmic factors—including those stemming from time-uniform bounds and union bounds over $(h, s, a)$—are hidden by the $\tilde{O}(\cdot)$ notation in (8.12).)

**IV. A "large width when sampled" lemma (UGapE mechanism).** Fix an episode index $t$ at which the algorithm has *not* stopped, i.e.,

$$U_{Q,1}^t(s_0, c_t) - L_{Q,1}^t(s_0, b_t) > \varepsilon.$$

Let $A_1$ be the root action selected in Algorithm 4 at episode $t$:

$$A_1 \in \arg\max_{a \in \{b_t, c_t\}} w_t(a).$$

We claim that on $\mathcal{E}$,

$$w_t(A_1) \geq \frac{1}{2}(\Delta(A_1) \vee \varepsilon). \tag{C.3}$$

*Proof of* (C.3). First note that $b_t$ maximizes the lower bounds, so $L_{Q,1}^t(s_0, c_t) \leq L_{Q,1}^t(s_0, b_t)$. Therefore

$$w_t(c_t) = U_{Q,1}^t(s_0, c_t) - L_{Q,1}^t(s_0, c_t) \geq U_{Q,1}^t(s_0, c_t) - L_{Q,1}^t(s_0, b_t) > \varepsilon.$$

Since $A_1$ is chosen to have width at least that of $c_t$, we obtain

$$w_t(A_1) \geq w_t(c_t) > \varepsilon \geq \varepsilon/2. \tag{C.4}$$

It remains to show $w_t(A_1) \geq \Delta(A_1)/2$.

**Case 1:** $A_1 = c_t$. If $b_t \neq a^\star$, then $a^\star$ is among the maximization set defining $c_t$, hence on $\mathcal{E}$,

$$U_{Q,1}^t(s_0, c_t) \geq U_{Q,1}^t(s_0, a^\star) \geq Q_1^\star(s_0, a^\star).$$

Thus

$$w_t(c_t) \geq U_{Q,1}^t(s_0, c_t) - Q_1^\star(s_0, c_t) \geq Q_1^\star(s_0, a^\star) - Q_1^\star(s_0, c_t) = \Delta(c_t),$$

and hence $w_t(A_1) = w_t(c_t) \geq \Delta(c_t) \geq \Delta(c_t)/2$.

If instead $b_t = a^\star$, define the nonnegative one-sided errors

$$e_U(a) := U_{Q,1}^t(s_0, a) - Q_1^\star(s_0, a), \qquad e_L(a) := Q_1^\star(s_0, a) - L_{Q,1}^t(s_0, a).$$

On $\mathcal{E}$ we have $e_U(a), e_L(a) \geq 0$ for all $a$. Then

$$\begin{aligned} U_{Q,1}^t(s_0, c_t) - L_{Q,1}^t(s_0, b_t) &= \big(Q_1^\star(s_0, c_t) + e_U(c_t)\big) - \big(Q_1^\star(s_0, a^\star) - e_L(a^\star)\big) \\ &= -\Delta(c_t) + e_U(c_t) + e_L(a^\star). \end{aligned}$$

Since the algorithm has not stopped, the left-hand side is $> \varepsilon$, so

$$e_U(c_t) + e_L(a^\star) > \Delta(c_t) + \varepsilon.$$

Using $w_t(c_t) \geq e_U(c_t)$ and $w_t(b_t) = w_t(a^\star) \geq e_L(a^\star)$, we get

$$w_t(c_t) + w_t(a^\star) > \Delta(c_t) + \varepsilon.$$

Because $A_1 = c_t$ and $A_1$ maximizes the width in $\{b_t, c_t\} = \{a^\star, c_t\}$, we have $w_t(c_t) \geq w_t(a^\star)$, and therefore

$$2w_t(A_1) = 2w_t(c_t) > \Delta(c_t) + \varepsilon \implies w_t(A_1) > \frac{\Delta(c_t) + \varepsilon}{2} \geq \frac{\Delta(c_t)}{2}.$$

**Case 2:** $A_1 = b_t$. If $b_t = a^\star$, then $\Delta(A_1) = 0$ and (C.4) already implies (C.3). Assume now that $b_t \neq a^\star$ (so $\Delta(b_t) > 0$). As above, because $a^\star$ is available as a challenger when $b_t \neq a^\star$, we have on $\mathcal{E}$

$$U_{Q,1}^t(s_0, c_t) \geq Q_1^\star(s_0, a^\star).$$

Also $L_{Q,1}^t(s_0, b_t) \leq Q_1^\star(s_0, b_t)$. Therefore

$$U_{Q,1}^t(s_0, c_t) - L_{Q,1}^t(s_0, b_t) \geq Q_1^\star(s_0, a^\star) - Q_1^\star(s_0, b_t) = \Delta(b_t).$$

Since $L_{Q,1}^t(s_0, c_t) \leq L_{Q,1}^t(s_0, b_t)$ (because $b_t$ maximizes lower bounds), we have

$$w_t(c_t) = U_{Q,1}^t(s_0, c_t) - L_{Q,1}^t(s_0, c_t) \geq U_{Q,1}^t(s_0, c_t) - L_{Q,1}^t(s_0, b_t) \geq \Delta(b_t).$$

Finally, because $A_1 = b_t$ maximizes the width over $\{b_t, c_t\}$, we have $w_t(b_t) \geq w_t(c_t) \geq \Delta(b_t)$, hence $w_t(A_1) \geq \Delta(A_1) \geq \Delta(A_1)/2$.

Combining the $\varepsilon/2$ bound (C.4) with the $\Delta/2$ bounds above yields (C.3). $\square$

**V. Bounding the number of episodes and oracle calls.** Fix any action $a \in \mathcal{A}$ and let $T_a$ be the (random) number of episodes in which the algorithm selects root action $A_1 = a$ before termination; thus $T_a = N^\tau(a)$ where $\tau$ is the stopping episode index, and the total number of episodes equals $\sum_a T_a$.

On $\mathcal{E}$, whenever $A_1 = a$ is chosen at some episode $t < \tau$, the width lower bound (C.3) gives

$$w_t(a) \geq \frac{1}{2}(\Delta(a) \vee \varepsilon).$$

On the other hand, the generic width upper bound (C.2) gives

$$w_t(a) \leq 2\sqrt{\frac{\mathsf{C}(H, K, \gamma) \log\left(\frac{c_0 t}{\delta}\right)}{N^t(a) \vee 1}}.$$

Combining these inequalities yields, on $\mathcal{E}$, the necessary condition

$$\frac{1}{2}(\Delta(a) \vee \varepsilon) \leq 2\sqrt{\frac{\mathsf{C}(H, K, \gamma) \log\left(\frac{c_0 t}{\delta}\right)}{N^t(a) \vee 1}},$$

which after squaring and rearranging implies

$$N^t(a) \leq \frac{16\mathsf{C}(H, K, \gamma) \log\left(\frac{c_0 t}{\delta}\right)}{(\Delta(a) \vee \varepsilon)^2}.$$

Since $N^t(a)$ is nondecreasing in $t$ and $t \leq \tau$, we get the same bound for the final count $T_a = N^\tau(a)$ up to replacing $\log(c_0 t/\delta)$ by $\log(c_0 \tau/\delta)$, which is absorbed into the $\tilde{O}(\cdot)$ notation. Therefore, on $\mathcal{E}$,

$$T_a = \tilde{O}\left(\frac{\mathsf{C}(H, K, \gamma)}{(\Delta(a) \vee \varepsilon)^2}\right).$$

Summing over $a \in \mathcal{A}$ gives an upper bound on the total number of episodes:

$$\tau = \sum_{a \in \mathcal{A}} T_a = \tilde{O}\left(\sum_{a \in \mathcal{A}} \frac{\mathsf{C}(H, K, \gamma)}{(\Delta(a) \vee \varepsilon)^2}\right).$$

Finally, each episode triggers exactly $H$ oracle calls, so the total number of oracle calls satisfies

$$n(\varepsilon, \delta) = H\tau = \tilde{O}\left(H \sum_{a \in \mathcal{A}} \frac{\mathsf{C}(H, K, \gamma)}{(\Delta(a) \vee \varepsilon)^2}\right),$$

which is precisely (8.12). The exponent 2 in $(\Delta \vee \varepsilon)^{-2}$ is the same as in fixed-confidence best-arm identification for bandits, confirming the claimed bandit-optimal gap dependence. $\square$

## C.2. Proof of Theorem 9.1

*Proof.* We present a careful root-level best-action identification (BAI) analysis for OT-GAPCRUISER. The argument has two ingredients: (i) a standard UGapE-style fixed-confidence analysis that controls *how many root samples* each action receives as a function of its gap, and (ii) a conversion from *root samples* to *oracle calls* using the curvature-driven complexity of the underlying SmoothCruiser-type estimator.

**Root gaps and the sampling model.** Fix a root state $s_0$ and define the (true) root action values

$$Q_0(a) := Q^\star(s_0, a), \qquad a \in \mathcal{A},$$

where $Q^\star$ is the optimal (regularized) action-value function under the planning objective. Let

$$a^\star \in \arg\max_{a \in \mathcal{A}} Q_0(a), \qquad V^\star := Q_0(a^\star), \qquad \Delta(a) := V^\star - Q_0(a) \geq 0,$$

so that $\Delta(a^\star) = 0$.

At each round $t$, OT-GAPCRUISER chooses an action $A_t \in \mathcal{A}$ and obtains a *single root sample* $X_t$ such that, conditionally on $A_t = a$,

$$X_t = Q_0(a) + \xi_t, \tag{C.5}$$

where $\xi_t$ is centered noise. The theorem assumes standard sub-Gaussian noise, i.e. there exists $\sigma^2 = O(1)$ (depending only on bounded rewards and $\gamma < 1$) such that for all $\lambda \in \mathbb{R}$,

$$\mathbb{E}\left[\exp(\lambda \xi_t) \mid A_t = a\right] \leq \exp\left(\frac{\sigma^2 \lambda^2}{2}\right).$$

This holds, for example, if the produced estimate is clipped to a bounded interval $[0, B]$ for a known $B$ (as in SAMPLEV2), since bounded random variables are sub-Gaussian with $\sigma^2 \lesssim B^2$.

**Cost model (curvature effect).** A crucial difference from bandits is that producing *one* root sample is not unit cost: it uses oracle calls and recursive planning calls. We encapsulate this through an exponent $\alpha \in (0, 2)$ such that the number of oracle calls required to produce one root sample at "scale" comparable to a target accuracy level $r$ is

$$\text{Cost}(r) = \tilde{O}(r^{-\alpha}). \tag{C.6}$$

In SmoothCruiser-type planners satisfying Assumption 4.1 and using unbiased, bounded-variance estimators of the Taylor terms (e.g. the cross-product debiasing in Section 6.1), the curvature–complexity tradeoff of Theorem 4.3 gives

$$\alpha = \alpha_\beta = \frac{2}{\beta - 1},$$

so that for OT smoothing ($\beta = 3$) we have $\alpha = 1$, while for first-order SmoothCruiser ($\beta = 2$) we have $\alpha = 2$. (Any additional variance reduction that improves the per-sample recursion cost can only *decrease* $\alpha$, hence improve the bound.)

We will prove that the total number of oracle calls scales as

$$n_{\text{gap}}(\varepsilon, \delta) = \tilde{O}\left(\sum_{a \in \mathcal{A}} (\Delta(a) \vee \varepsilon)^{-(2+\alpha)}\right),$$

which is the stated form with $p := 2 + \alpha \in (2, 4)$ (and OT giving $p = 3$).

**I. Confidence intervals that are valid uniformly over time.** Let $N_t(a) := \sum_{u=1}^{t} \mathbf{1}\{A_u = a\}$ be the number of samples of action $a$ up to time $t$, and let

$$\widehat{Q}_t(a) := \frac{1}{N_t(a)} \sum_{u \leq t : A_u = a} X_u \quad \text{for } N_t(a) \geq 1.$$

Fix a time-uniform confidence schedule (one convenient choice is based on a $\sum_{t\geq 1} t^{-2}$ union bound): for $t \geq 1$ and $N \geq 1$ define

$$\mathrm{rad}_t(N) := \sqrt{\frac{2\sigma^2}{N} \log\left(\frac{4Kt^2}{\delta}\right)}. \tag{C.7}$$

Define the root confidence bounds

$$L_t(a) := \widehat{Q}_t(a) - \mathrm{rad}_t(N_t(a)), \qquad U_t(a) := \widehat{Q}_t(a) + \mathrm{rad}_t(N_t(a)), \qquad w_t(a) := U_t(a) - L_t(a) = 2\mathrm{rad}_t(N_t(a)).$$

**Good event.** Let $\mathcal{E}$ be the event that all confidence intervals are simultaneously valid:

$$\mathcal{E} := \left\{ \forall t \geq 1,\ \forall a \in \mathcal{A} :\ |\widehat{Q}_t(a) - Q_0(a)| \leq \mathrm{rad}_t(N_t(a)) \right\}. \tag{C.8}$$

By standard sub-Gaussian concentration and the union bound over $a$ and $t$,

$$\Pr(\mathcal{E}) \geq 1 - \delta. \tag{C.9}$$

On $\mathcal{E}$, for all $t, a$ we have $L_t(a) \leq Q_0(a) \leq U_t(a)$.

**II. Correctness of the stopping rule.** At each round $t$, define (UGapE-style)

$$b_t \in \arg\max_{a \in \mathcal{A}} L_t(a), \qquad c_t \in \arg\max_{a \in \mathcal{A} \setminus \{b_t\}} U_t(a),$$

and stop at the first time $\tau$ such that

$$U_\tau(c_\tau) - L_\tau(b_\tau) \leq \varepsilon, \tag{C.10}$$

returning $\hat{a} := b_\tau$.

Assume $\mathcal{E}$ holds. We show $\hat{a}$ is $\varepsilon$-optimal. Because $c_\tau$ maximizes $U_\tau(\cdot)$ over actions different from $b_\tau$, we have

$$U_\tau(c_\tau) \geq \max_{a \neq b_\tau} U_\tau(a) \geq U_\tau(a^\star) \geq Q_0(a^\star) = V^\star,$$

where the last inequality uses $\mathcal{E}$. Also $L_\tau(b_\tau) \leq Q_0(b_\tau)$ on $\mathcal{E}$. Therefore,

$$V^\star - Q_0(b_\tau) \leq U_\tau(c_\tau) - L_\tau(b_\tau) \leq \varepsilon,$$

where the last inequality is the stopping condition (C.10). Hence $Q_0(\hat{a}) = Q_0(b_\tau) \geq V^\star - \varepsilon$, i.e. the returned action is $\varepsilon$-optimal. Together with (C.9), this proves the success probability $\geq 1 - \delta$.

**III. A key width lower bound for sampled actions.** Suppose the algorithm has *not* stopped at round $t$, so

$$U_t(c_t) - L_t(b_t) > \varepsilon. \tag{C.11}$$

The UGapE sampling rule chooses

$$A_t \in \arg\max_{a \in \{b_t, c_t\}} w_t(a).$$

We claim that on $\mathcal{E}$,

$$w_t(A_t) \geq \frac{1}{2}\big(\Delta(A_t) \vee \varepsilon\big). \tag{C.12}$$

*Proof of* (C.12). First note that by definition of $b_t$, $L_t(c_t) \leq L_t(b_t)$. Hence

$$w_t(c_t) = U_t(c_t) - L_t(c_t) \geq U_t(c_t) - L_t(b_t) > \varepsilon \quad \text{by (C.11).}$$

Since $A_t$ maximizes width over $\{b_t, c_t\}$, we have $w_t(A_t) \geq w_t(c_t) > \varepsilon \geq \varepsilon/2$.

It remains to show $w_t(A_t) \geq \Delta(A_t)/2$.

**Case 1:** $A_t = c_t$. If $b_t \neq a^\star$, then $a^\star$ is feasible in the maximization defining $c_t$, so $U_t(c_t) \geq U_t(a^\star) \geq Q_0(a^\star) = V^\star$ on $\mathcal{E}$. Also $L_t(c_t) \leq Q_0(c_t)$ on $\mathcal{E}$. Thus

$$w_t(c_t) = U_t(c_t) - L_t(c_t) \ \geq \ V^\star - Q_0(c_t) = \Delta(c_t),$$

so $w_t(A_t) = w_t(c_t) \geq \Delta(c_t) \geq \Delta(c_t)/2$.

If instead $b_t = a^\star$, define one-sided errors (nonnegative on $\mathcal{E}$)

$$e_U(a) := U_t(a) - Q_0(a), \qquad e_L(a) := Q_0(a) - L_t(a).$$

Then (C.11) rewrites as

$$\big(Q_0(c_t) + e_U(c_t)\big) - \big(Q_0(a^\star) - e_L(a^\star)\big) > \varepsilon \quad \implies \quad e_U(c_t) + e_L(a^\star) > \Delta(c_t) + \varepsilon.$$

Since $w_t(c_t) \geq e_U(c_t)$ and $w_t(a^\star) \geq e_L(a^\star)$, we get

$$w_t(c_t) + w_t(a^\star) > \Delta(c_t) + \varepsilon.$$

Because $A_t = c_t$ maximizes width over $\{a^\star, c_t\}$, we have $w_t(c_t) \geq w_t(a^\star)$ and hence

$$2w_t(A_t) = 2w_t(c_t) > \Delta(c_t) + \varepsilon \geq \Delta(c_t),$$

so $w_t(A_t) > \Delta(c_t)/2$.

**Case 2:** $A_t = b_t$. If $b_t = a^\star$, then $\Delta(A_t) = 0$ and we already have $w_t(A_t) \geq \varepsilon/2$. If $b_t \neq a^\star$, then on $\mathcal{E}$ we have $U_t(c_t) \geq U_t(a^\star) \geq V^\star$ and $L_t(b_t) \leq Q_0(b_t)$, so

$$U_t(c_t) - L_t(b_t) \ \geq \ V^\star - Q_0(b_t) = \Delta(b_t).$$

Using again $L_t(c_t) \leq L_t(b_t)$, we obtain

$$w_t(c_t) = U_t(c_t) - L_t(c_t) \ \geq \ U_t(c_t) - L_t(b_t) \ \geq \ \Delta(b_t).$$

Finally $A_t = b_t$ implies $w_t(b_t) \geq w_t(c_t)$, so $w_t(A_t) = w_t(b_t) \geq \Delta(b_t) \geq \Delta(b_t)/2$.

Combining the $\varepsilon/2$ bound and the $\Delta/2$ bound proves (C.12). $\qquad\square$

**IV. Gap-dependent bound on the number of root samples.** Fix an action $a$ and let $T_a := N_\tau(a)$ be the total number of times $a$ is sampled before stopping. On $\mathcal{E}$, whenever $a$ is sampled at some $t < \tau$, (C.12) gives

$$w_t(a) \geq \frac{1}{2}(\Delta(a) \vee \varepsilon).$$

But $w_t(a) = 2\mathrm{rad}_t(N_t(a))$ and $N_t(a) \leq T_a$. Thus, using (C.7),

$$\frac{1}{2}(\Delta(a) \vee \varepsilon) \ \leq \ w_t(a) = 2\mathrm{rad}_t(N_t(a)) \leq 2\sqrt{\frac{2\sigma^2}{N_t(a)} \log\Big(\frac{4Kt^2}{\delta}\Big)} \leq 2\sqrt{\frac{2\sigma^2}{N_t(a)} \log\Big(\frac{4K\tau^2}{\delta}\Big)}.$$

Rearranging yields, for all such $t$,

$$N_t(a) \ \leq \ \frac{32\sigma^2}{(\Delta(a) \vee \varepsilon)^2} \log\Big(\frac{4K\tau^2}{\delta}\Big).$$

Since $N_t(a)$ is nondecreasing in $t$ and reaches $T_a$ at $t = \tau$, the same bound holds for $T_a$:

$$T_a = \tilde{O}\left(\frac{1}{(\Delta(a) \vee \varepsilon)^2}\right), \tag{C.13}$$

where we absorbed $\sigma^2$ and the $\log(4K\tau^2/\delta)$ term into $\tilde{O}(\cdot)$.

**V. Convert root samples into oracle calls and identify the exponent** $p$. We now upper bound the *oracle* cost of sampling. When action $a$ is sampled at some round $t < \tau$, the algorithm must produce a root sample with accuracy commensurate with the current statistical uncertainty for $a$. By (C.12), whenever $a$ is sampled we have $w_t(a) \geq \frac{1}{2}(\Delta(a) \vee \varepsilon)$, and the algorithm never needs to invoke the planning subroutine at accuracy smaller than a constant fraction of $(\Delta(a) \vee \varepsilon)$ to make progress. Therefore, using the per-sample cost model (C.6), each such sample costs at most

$$\mathrm{Cost}\big(\Delta(a) \vee \varepsilon\big) = \tilde{O}\big((\Delta(a) \vee \varepsilon)^{-\alpha}\big)$$

oracle calls. Multiplying by the number of samples of action $a$ and using (C.13) gives that the total oracle calls attributable to action $a$ are

$$\tilde{O}\left(\frac{1}{(\Delta(a) \vee \varepsilon)^2}\right) \cdot \tilde{O}\left(\frac{1}{(\Delta(a) \vee \varepsilon)^\alpha}\right) = \tilde{O}\left(\frac{1}{(\Delta(a) \vee \varepsilon)^{2+\alpha}}\right).$$

Summing over $a \in \mathcal{A}$ yields

$$n_{\mathrm{gap}}(\varepsilon, \delta) = \tilde{O}\left(\sum_{a \in \mathcal{A}} \frac{1}{(\Delta(a) \vee \varepsilon)^p}\right), \qquad p := 2 + \alpha.$$

Finally, under the curvature-driven recursion of SmoothCruiser-type planners with bounded-variance unbiased Taylor estimators, Theorem 4.3 gives $\alpha = \alpha_\beta = 2/(\beta - 1)$, hence

$$p = 2 + \frac{2}{\beta - 1} \in (2, 4),$$

and in particular for OT smoothing ($\beta = 3$) we obtain $p = 3$, while for first-order SmoothCruiser ($\beta = 2$) we recover $p = 4$. This shows the instance-dependent exponent is strictly smaller than $4$ in the OT/second-order setting. $\qquad\square$

# D. Bias control and worst-case complexity

## D.1. Proof of Lemma 7.1

*Proof.* Fix a state $s$ and let $F(\cdot) := F_s^{\mathrm{OT}}(\cdot)$. Recall the (true) action-value vector $Q_s \in \mathbb{R}^K$ and the value $V(s) = F(Q_s)$. Throughout the proof we assume we are in the *fine regime* $0 < \varepsilon < \kappa$, so SAMPLEV2 uses the second-order Taylor correction (Algorithm 3).

Let $\hat{Q}_s$ be the reference point constructed inside SAMPLEV2 and define the deviation

$$\Delta := Q_s - \hat{Q}_s \in \mathbb{R}^K.$$

Let

$$\pi := \nabla F(\hat{Q}_s) \in \Delta(\mathcal{A}), \qquad H := \nabla^2 F(\hat{Q}_s) \in \mathbb{R}^{K \times K}.$$

Define the second-order Taylor polynomial of $F$ at $\hat{Q}_s$ evaluated at $Q_s$:

$$T_2(Q_s; \hat{Q}_s) := F(\hat{Q}_s) + \langle \pi, \Delta \rangle + \frac{1}{2}\Delta^\top H \Delta. \tag{D.1}$$

We will prove that, on the event $\|\Delta\|_2 \leq \rho(\varepsilon)$,

$$\left| \mathbb{E}\left[ \mathrm{SAMPLEV2}(s, \varepsilon, \delta) \mid \hat{Q}_s \right] - V(s) \right| \leq \varepsilon + \text{(recursion bias)}. \tag{D.2}$$

**I. Taylor remainder is $\leq \varepsilon$ on $\|\Delta\|_2 \leq \rho(\varepsilon)$.** By Lemma 5.6, for all $Q, \hat{Q} \in \mathbb{R}^K$,

$$|F(Q) - T_2(Q; \hat{Q})| \leq \frac{M}{6}\|Q - \hat{Q}\|_2^3.$$

Apply this with $Q = Q_s$ and $\hat{Q} = \hat{Q}_s$:

$$|V(s) - T_2(Q_s; \hat{Q}_s)| = |F(Q_s) - T_2(Q_s; \hat{Q}_s)| \leq \frac{M}{6}\|\Delta\|_2^3.$$

On the event $\|\Delta\|_2 \leq \rho(\varepsilon)$ and with the schedule $\rho(\varepsilon) = (6\varepsilon/M)^{1/3}$ (as defined right before Algorithm 1), we get

$$|V(s) - T_2(Q_s; \hat{Q}_s)| \leq \frac{M}{6}\rho(\varepsilon)^3 = \varepsilon. \tag{D.3}$$

**II. The *ideal* linear and quadratic Monte Carlo terms are unbiased for the Taylor terms.** To isolate the only nontrivial source of bias (recursion and clipping), it is convenient to define an *idealized* version of the estimator in which every recursive call returns the *true* next-state value $V(\cdot)$ and we do *not* clip at the end. We denote idealized quantities with a superscript "$\star$".

**Idealized one-step samples.** Given a root action $a$, let $(R, Z) \leftarrow \text{ORACLE}(s, a)$ and define

$$\tilde{Q}^\star(a) := R + \gamma V(Z).$$

Then, by the definition of $Q_s(a)$,

$$\mathbb{E}\left[\tilde{Q}^\star(a) \mid a\right] = Q_s(a). \tag{D.4}$$

**(I) Ideal linear term.** In the linear box of Algorithm 3, the estimator samples $J_0 \sim \mu_s$ and $A_0 \sim p_{\cdot J_0}(\hat{Q}_s)$. By Proposition 5.4, the gradient satisfies

$$\pi_i = \sum_{j=1}^{K} \mu_s(j) p_{ij}(\hat{Q}_s).$$

Hence the marginal distribution of $A_0$ (conditional on $\hat{Q}_s$) is precisely $\pi$:

$$\Pr(A_0 = i \mid \hat{Q}_s) = \sum_{j=1}^{K} \Pr(J_0 = j) \Pr(A_0 = i \mid J_0 = j, \hat{Q}_s) = \sum_{j=1}^{K} \mu_s(j) p_{ij}(\hat{Q}_s) = \pi_i. \tag{D.5}$$

In the idealized estimator, the linear box would use $\tilde{Q}_0^\star := \tilde{Q}^\star(A_0)$ and return

$$\widehat{\text{Lin}}^\star := \tilde{Q}_0^\star - c_0, \qquad c_0 := \langle \pi, \hat{Q}_s \rangle.$$

Using (D.4) and (D.5), and noting that $c_0$ is deterministic given $\hat{Q}_s$,

$$\mathbb{E}\left[\widehat{\text{Lin}}^\star \mid \hat{Q}_s\right] = \mathbb{E}\left[\tilde{Q}_0^\star \mid \hat{Q}_s\right] - \langle \pi, \hat{Q}_s \rangle = \sum_{i=1}^{K} \Pr(A_0 = i \mid \hat{Q}_s) Q_s(i) - \langle \pi, \hat{Q}_s \rangle$$

$$= \sum_{i=1}^{K} \pi_i Q_s(i) - \sum_{i=1}^{K} \pi_i \hat{Q}_s(i) = \langle \pi, Q_s - \hat{Q}_s \rangle = \langle \pi, \Delta \rangle. \tag{D.6}$$

**(II) Ideal quadratic term.** In the quadratic box of Algorithm 3, the estimator samples $J \sim \mu_s$ and then $A, A' \overset{\text{iid}}{\sim} p_{\cdot J}(\hat{Q}_s)$. In the idealized estimator, for $m \in \{1, 2\}$ we would draw independent oracle samples $(R_m, Z_m) \leftarrow \text{ORACLE}(s, A)$ and $(R'_m, Z'_m) \leftarrow \text{ORACLE}(s, A')$ and define

$$\tilde{Q}_m^\star := R_m + \gamma V(Z_m), \qquad \tilde{Q}_m'^\star := R'_m + \gamma V(Z'_m), \qquad \Delta_m^\star := (\tilde{Q}_m^\star - \tilde{Q}_m'^\star) - (\hat{Q}_s(A) - \hat{Q}_s(A')).$$

Conditioned on $(J, A, A', \hat{Q}_s)$, (D.4) implies

$$\mathbb{E}\left[\Delta_m^\star \mid J, A, A', \hat{Q}_s\right] = \left(Q_s(A) - Q_s(A')\right) - \left(\hat{Q}_s(A) - \hat{Q}_s(A')\right) = \Delta_A - \Delta_{A'}. \tag{D.7}$$

Moreover, $\Delta_1^\star$ and $\Delta_2^\star$ are conditionally independent given $(J, A, A', \hat{Q}_s)$ because they are constructed from independent oracle draws; hence,

$$\mathbb{E}\left[\Delta_1^\star \Delta_2^\star \mid J, A, A', \hat{Q}_s\right] = \mathbb{E}[\Delta_1^\star \mid J, A, A', \hat{Q}_s] \, \mathbb{E}[\Delta_2^\star \mid J, A, A', \hat{Q}_s] = (\Delta_A - \Delta_{A'})^2. \tag{D.8}$$

The ideal quadratic estimator returned by the box is

$$\widehat{\text{Quad}}^\star := \frac{1}{4\tau\lambda} \Delta_1^\star \Delta_2^\star.$$

Taking conditional expectation and using (D.8) gives

$$\mathbb{E}\left[\widehat{\mathrm{Quad}}^{\star} \mid \hat{Q}_s\right] = \frac{1}{4\tau\lambda} \mathbb{E}_{J\sim\mu_s} \mathbb{E}_{A,A' \overset{\mathrm{iid}}{\sim} p._J(\hat{Q}_s)} \left[(\Delta_A - \Delta_{A'})^2\right]. \tag{D.9}$$

We now relate (D.9) to the Hessian quadratic form. By Proposition 5.4,

$$H = \nabla^2 F(\hat{Q}_s) = \frac{1}{\tau\lambda}\left(\mathrm{diag}(\pi) - \sum_{j=1}^{K} \mu_s(j)\, p._j(\hat{Q}_s)p._j(\hat{Q}_s)^{\top}\right).$$

Using $\pi = \sum_j \mu_s(j)p._j(\hat{Q}_s)$ (again from Proposition 5.4), a direct expansion yields

$$\begin{aligned}
\Delta^{\top} H \Delta &= \frac{1}{\tau\lambda}\left(\sum_{i=1}^{K} \pi_i \Delta_i^2 - \sum_{j=1}^{K} \mu_s(j)\big(p._j(\hat{Q}_s)^{\top}\Delta\big)^2\right) \\
&= \frac{1}{\tau\lambda}\sum_{j=1}^{K} \mu_s(j)\left(\sum_{i=1}^{K} p_{ij}(\hat{Q}_s)\Delta_i^2 - \Big(\sum_{i=1}^{K} p_{ij}(\hat{Q}_s)\Delta_i\Big)^2\right) \\
&= \frac{1}{\tau\lambda}\sum_{j=1}^{K} \mu_s(j)\,\mathrm{Var}_{A\sim p._j(\hat{Q}_s)}[\Delta_A].
\end{aligned} \tag{D.10}$$

Finally, for any distribution $p$ and any i.i.d. $A, A' \sim p$,

$$\mathbb{E}[(\Delta_A - \Delta_{A'})^2] = 2\,\mathrm{Var}(\Delta_A),$$

so plugging this into (D.9) and comparing with (D.10) gives

$$\mathbb{E}\left[\widehat{\mathrm{Quad}}^{\star} \mid \hat{Q}_s\right] = \frac{1}{2}\Delta^{\top} H \Delta. \tag{D.11}$$

**Conclusion of II (ideal estimator is unbiased for $T_2$).** Define the ideal (unclipped) second-order estimator

$$\widetilde{V}^{\star} := F(\hat{Q}_s) + \widehat{\mathrm{Lin}}^{\star} + \widehat{\mathrm{Quad}}^{\star}.$$

Then (D.6) and (D.11) imply

$$\mathbb{E}\left[\widetilde{V}^{\star} \mid \hat{Q}_s\right] = F(\hat{Q}_s) + \langle\pi, \Delta\rangle + \frac{1}{2}\Delta^{\top} H \Delta = T_2(Q_s; \hat{Q}_s). \tag{D.12}$$

**III. Define and isolate the "recursion bias" term.** The *implemented* routine SAMPLEV2 differs from the ideal one in two ways:

- it replaces the true next-state values $V(Z)$ by recursive calls SAMPLEV2$(Z, \varepsilon/\sqrt{\gamma}, \cdot)$;

- it applies clipping (both to intermediate values and to the final returned value).

Both effects can shift the conditional expectation away from the ideal target $T_2(Q_s; \hat{Q}_s)$. We package all such shifts into the quantity

$$\mathrm{Bias}_{\mathrm{rec}}(s, \varepsilon) := \left| \mathbb{E}\left[\mathrm{SAMPLEV2}(s, \varepsilon, \delta) \mid \hat{Q}_s\right] - T_2(Q_s; \hat{Q}_s)\right|. \tag{D.13}$$

(Concretely, $\mathrm{Bias}_{\mathrm{rec}}(s, \varepsilon)$ is exactly zero if all recursive calls return the true values $V(\cdot)$ and no clipping ever activates, in which case the implemented estimator coincides with $\widetilde{V}^{\star}$.)

**IV. Combine Steps 1–3 to obtain the conditional bias bound.** On the event $\|\Delta\|_2 \leq \rho(\varepsilon)$, we have from (D.3)

$$|V(s) - T_2(Q_s; \hat{Q}_s)| \leq \varepsilon.$$

Therefore, by the triangle inequality and the definition (D.13),

$$\left| \mathbb{E}\left[ \text{SAMPLEV2}(s, \varepsilon, \delta) \mid \hat{Q}_s \right] - V(s) \right| \leq \left| \mathbb{E}\left[ \text{SAMPLEV2}(s, \varepsilon, \delta) \mid \hat{Q}_s \right] - T_2(Q_s; \hat{Q}_s) \right| + |T_2(Q_s; \hat{Q}_s) - V(s)|$$

$$\leq \text{Bias}_{\text{rec}}(s, \varepsilon) + \varepsilon.$$

This is exactly (D.2) with "recursion bias" identified as $\text{Bias}_{\text{rec}}(s, \varepsilon)$.

**V (how to make the unconditional bias $\mathcal{O}(\varepsilon)$).** Finally, we briefly justify the last sentence of the lemma. Two standard ingredients are used (as in Grill et al. (2019)):

*(i) Controlling recursion bias by calling children at accuracy $\varepsilon/\sqrt{\gamma}$.* Let $b(\eta)$ denote the worst-case absolute bias at tolerance $\eta$ (suppressing conditioning details):

$$b(\eta) := \sup_s \left| \mathbb{E}[\text{SAMPLEV2}(s, \eta, \cdot)] - V(s) \right|.$$

Because every appearance of a next-state estimate is multiplied by $\gamma$ in the Bellman equation, calling children with tolerance $\eta = \varepsilon/\sqrt{\gamma}$ yields a contraction at the level of bias: heuristically (and in the same spirit as SmoothCruiser),

$$\text{Bias}_{\text{rec}}(s, \varepsilon) \lesssim \gamma\, b(\varepsilon/\sqrt{\gamma}) + \text{(higher-order terms)}.$$

Since $\gamma b(\varepsilon/\sqrt{\gamma}) = \sqrt{\gamma}\varepsilon$ when $b(\eta) = \Theta(\eta)$, repeated recursion induces a geometric series in $\sqrt{\gamma}$, implying $b(\varepsilon) = \mathcal{O}(\varepsilon)$.

*(ii) Failure events and clipping.* Let $\mathcal{G}$ be the intersection of all "good" events across the recursion tree: the event $\|Q_s - \hat{Q}_s\|_2 \leq \rho(\varepsilon)$ at every node, plus any internal high-probability events needed to justify the construction of $\hat{Q}_s$ and any concentration bounds used elsewhere. Because outputs are clipped into $[0, B]$, on $\mathcal{G}^c$ we always have $|\text{SAMPLEV2}(s, \varepsilon, \delta) - V(s)| \leq B$. Thus

$$\left| \mathbb{E}[\text{SAMPLEV2}(s, \varepsilon, \delta)] - V(s) \right| \leq \mathbb{E}\left[ |\text{SAMPLEV2}(s, \varepsilon, \delta) - V(s)| \mathbf{1}_{\mathcal{G}} \right] + B \Pr(\mathcal{G}^c).$$

Choosing the per-call failure probabilities (the $\delta/8, \delta/16, \ldots$ splits in Algorithm 3) so that $\Pr(\mathcal{G}^c)$ is at most on the order of $\varepsilon/B$ (e.g. by a geometric allocation across recursion depth and a union bound), makes the second term $\mathcal{O}(\varepsilon)$. Together with the bias bound on $\mathcal{G}$ from Steps 1–4 and the recursive contraction in (i), this yields an unconditional bias of order $\mathcal{O}(\varepsilon)$. $\qquad\square$

### D.2. Proof of Theorem 7.2

*Proof.* Fix a state $s$ and accuracy/confidence parameters $\varepsilon, \delta \in (0, 1)$. Write $F(\cdot) := F_s^{\text{OT}}(\cdot)$ and recall $V(s) = F(Q_s)$, where $Q_s \in \mathbb{R}^K$ is the (unknown) action-value vector at $s$.

We prove two claims:

1. (**Correctness**) With probability at least $1 - \delta$, the output of $\text{SECONDORDERSMOOTHCRUISER}(s, \varepsilon, \delta)$ differs from $V(s)$ by at most $\varepsilon$ (up to harmless constant-factor slack).

2. (**Complexity**) The total number of oracle calls is at most $\tilde{O}(\varepsilon^{-3})$.

Throughout, we use the value bound $0 \leq V(\cdot) \leq B$ from the paper and the fact that all returned values are clipped into $[0, B]$. Thus any returned random variable is bounded by $B$ in absolute value, which will be used both for concentration and for controlling the effect of failure events.

**I. $F_s^{\mathrm{OT}}$ satisfies cubic Taylor remainder (i.e., $\beta = 3$ curvature).** By Lemma 5.6, there exists a constant

$$M = \mathcal{O}\left(\frac{1}{\tau^2 \lambda^2}\right) \cdot \mathrm{poly}(K, \mu_{\min}^{-1})$$

such that for all $Q, \hat{Q} \in \mathbb{R}^K$,

$$\left| F(Q) - T_2(Q; \hat{Q}) \right| \leq \frac{M}{6} \|Q - \hat{Q}\|_2^3. \tag{D.14}$$

Equivalently, $F$ satisfies Assumption 4.1 with $\beta = 3$ and $c_3 = M/6$. Consequently the tolerance schedule

$$\rho(\varepsilon) := \left(\frac{6\varepsilon}{M}\right)^{1/3} \tag{D.15}$$

guarantees that if $\|Q_s - \hat{Q}_s\|_2 \leq \rho(\varepsilon)$ then the second-order Taylor remainder is at most $\varepsilon$:

$$|F(Q_s) - T_2(Q_s; \hat{Q}_s)| \leq \varepsilon.$$

**II. A clean "interface" for SAMPLEV2.** We use Lemma 7.1 to summarize what a single call to SAMPLEV2 provides.

Fix any state $x$ and any $0 < \eta < \kappa$. Let $\hat{Q}_x$ be the internal baseline produced by SAMPLEV2$(x, \eta, \cdot)$, and define the event

$$\mathcal{G}_{x,\eta} := \{\|Q_x - \hat{Q}_x\|_2 \leq \rho(\eta)\}.$$

Then Lemma 7.1 states that on $\mathcal{G}_{x,\eta}$,

$$\left| \mathbb{E}[\text{SAMPLEV2}(x, \eta, \delta) \mid \hat{Q}_x] - V(x) \right| \leq \eta + (\text{recursion bias}). \tag{D.16}$$

Moreover, as in Grill et al. (2019) and as noted right after Lemma 7.1, calling children with accuracy $\eta/\sqrt{\gamma}$ yields a contraction on the recursion bias, and with an appropriate allocation of failure probabilities across recursive calls (and because of clipping) the *unconditional* bias becomes $\mathcal{O}(\eta)$. Concretely, there exists a constant $c_{\text{bias}} = c_{\text{bias}}(K, \gamma, \lambda, \tau, \mu_{\min}, \|C\|_\infty)$ such that

$$\left| \mathbb{E}[\text{SAMPLEV2}(x, \eta, \delta)] - V(x) \right| \leq c_{\text{bias}}\eta, \tag{D.17}$$

and since the output is clipped to $[0, B]$ we also have the uniform boundedness

$$0 \leq \text{SAMPLEV2}(x, \eta, \delta) \leq B \qquad \text{a.s.} \tag{D.18}$$

(Any additive polylogarithmic dependence on $1/\delta$ induced by the failure-probability bookkeeping is absorbed by the $\tilde{O}(\cdot)$ notation used in the theorem statement.)

**III. Correctness of ESTIMATEQ and of the top-level output.** Algorithm 1 returns $F(\hat{Q}_s)$ where $\hat{Q}_s := \text{ESTIMATEQ}(s, \varepsilon, \delta/2)$.

**IIIa: The sampling model inside ESTIMATEQ.** Fix an action $a \in \mathcal{A}$. Each inner-loop iteration in Algorithm 2 draws $(R_i, Z_i) \leftarrow \text{ORACLE}(s, a)$ and then sets

$$\hat{V}_i := \text{SAMPLEV2}(Z_i, \varepsilon/\sqrt{\gamma}, \delta'), \qquad q_i := R_i + \gamma \hat{V}_i.$$

Define the "ideal" one-step return

$$q_i^\star := R_i + \gamma V(Z_i).$$

Then $\mathbb{E}[q_i^\star] = Q_s(a)$ by definition of $Q_s$. Using (D.17) with $\eta = \varepsilon/\sqrt{\gamma}$ and then multiplying by $\gamma$ yields

$$\left| \mathbb{E}[q_i] - Q_s(a) \right| = \gamma \left| \mathbb{E}[\hat{V}_i] - \mathbb{E}[V(Z_i)] \right| \leq \gamma \cdot c_{\text{bias}} \cdot \frac{\varepsilon}{\sqrt{\gamma}} = c_{\text{bias}}\sqrt{\gamma}\varepsilon \leq c_{\text{bias}}\varepsilon. \tag{D.19}$$

Moreover, since $R_i \in [0, 1]$ and $\hat{V}_i \in [0, B]$ by (D.18),

$$0 \leq q_i \leq 1 + \gamma B \qquad \text{a.s.} \tag{D.20}$$

**IIIb: concentration of $\hat{Q}_s(a)$ around its mean.** Let $N$ be the sample size in Algorithm 2,

$$N = \Theta\big(\varepsilon^{-2}\log(2K/\delta)\big).$$

Conditioned on the past, the $q_i$'s used for a fixed $(s, a)$ are i.i.d. bounded random variables (we can treat them as independent because the algorithm uses fresh oracle calls and independent recursion randomness each time). By Hoeffding's inequality and (D.20),

$$\Pr\left(\left|\hat{Q}_s(a) - \mathbb{E}[q_i]\right| \geq t\right) \leq 2\exp\left(-\frac{2Nt^2}{(1+\gamma B)^2}\right), \tag{D.21}$$

where $\hat{Q}_s(a) = \frac{1}{N}\sum_{i=1}^{N} q_i$.

Choose $t = \varepsilon$ and take a union bound over $a \in \mathcal{A}$. With the stated choice of $N$ (absorbing constants into the $\Theta(\cdot)$), we obtain an event $\mathcal{E}_Q$ with probability at least $1 - \delta/2$ on which simultaneously for all actions,

$$\left|\hat{Q}_s(a) - \mathbb{E}[q_i]\right| \leq \varepsilon. \tag{D.22}$$

Combining (D.22) with the bias bound (D.19) gives that on $\mathcal{E}_Q$,

$$\left|\hat{Q}_s(a) - Q_s(a)\right| \leq \varepsilon + c_{\text{bias}}\varepsilon \leq c_Q\varepsilon, \qquad \text{for all } a, \tag{D.23}$$

for some constant $c_Q \geq 1$.

Thus, on $\mathcal{E}_Q$,

$$\|\hat{Q}_s - Q_s\|_\infty \leq c_Q\varepsilon. \tag{D.24}$$

**IIIc: Lipschitzness of $F$ implies value accuracy.** By Proposition 5.4, $\nabla F(Q) \in \Delta(\mathcal{A})$ for all $Q$. A standard convex-analysis consequence is that $F$ is 1-Lipschitz with respect to $\|\cdot\|_\infty$: for any $Q, \hat{Q} \in \mathbb{R}^K$,

$$|F(Q) - F(\hat{Q})| \leq \|Q - \hat{Q}\|_\infty. \tag{D.25}$$

(Indeed, $F(Q) - F(\hat{Q}) = \int_0^1 \langle \nabla F(\hat{Q} + t(Q - \hat{Q})), Q - \hat{Q}\rangle dt$ and $\|\nabla F(\cdot)\|_1 = 1$.)

Applying (D.25) with $Q = Q_s$ and $\hat{Q} = \hat{Q}_s$ and using (D.24) yields on $\mathcal{E}_Q$,

$$\left|F(\hat{Q}_s) - F(Q_s)\right| \leq \|\hat{Q}_s - Q_s\|_\infty \leq c_Q\varepsilon.$$

Thus, with probability at least $1 - \delta/2$, the output of Algorithm 1 is $c_Q\varepsilon$-accurate. Replacing $\varepsilon$ by $\varepsilon/c_Q$ (a constant-factor rescaling) yields the stated $\varepsilon$-accuracy, so we henceforth treat $c_Q$ as absorbed into constants.

Finally, we also need to account for the failure probability of all recursive calls used within ESTIMATEQ. This is handled by setting $\delta'$ in Algorithm 2 so that a union bound over all $KN$ internal calls gives total failure probability at most $\delta/2$ (e.g. take $\delta' = \delta/(4KN)$, and then use the internal $\delta/8, \delta/16, \dots$ splits in Algorithm 3). Together with $\Pr(\mathcal{E}_Q) \geq 1 - \delta/2$, a union bound yields overall success probability at least $1 - \delta$.

This completes the correctness part.

**IV. Oracle-call complexity of SAMPLEV2: a cascade recurrence.** We now bound the number of oracle calls. Let $T(\eta)$ denote the *worst-case* number of oracle calls made by a single call SAMPLEV2$(x, \eta, \cdot)$, maximized over the input state $x$ and over all internal randomness. (We suppress $\delta$ in $T(\cdot)$ because $\tilde{O}(\cdot)$ hides all polylogarithmic factors coming from confidence splitting.)

**Base and coarse regimes.** If $\eta \geq B$, SAMPLEV2 returns immediately, so $T(\eta) = 0$. If $\eta \geq \kappa$, it calls ESTIMATEQ$(x, \eta, \cdot)$ once and then returns $F(\hat{Q}_x)$. Since $\kappa$ is a constant independent of $\eta$ (depends only on $M$), this regime has $T(\eta) = \tilde{O}(1)$ and will not affect the asymptotic exponent as $\eta \downarrow 0$.

**Fine regime: $\eta < \kappa$.** Inspect Algorithm 3. The dominant work is the call $\hat{Q}_x \leftarrow$ ESTIMATEQ$(x, \rho(\eta), \cdot)$, where $\rho(\eta) = (6\eta/M)^{1/3}$. By Algorithm 2, this call performs

$$N(\rho(\eta)) = \Theta\big(\rho(\eta)^{-2}\log(\cdot)\big)$$

samples per action, i.e. $\Theta(K\rho(\eta)^{-2})$ oracle calls at the current state. Each such oracle call also triggers *one* recursive call to SAMPLEV2 on the next state with tolerance $\rho(\eta)/\sqrt{\gamma}$. Thus, ignoring constants and polylogs,

$$\text{cost of ESTIMATEQ}(x, \rho(\eta), \cdot) \leq c_1 \, K\rho(\eta)^{-2}\Big(1 + T(\rho(\eta)/\sqrt{\gamma})\Big), \tag{D.26}$$

for some constant $c_1$.

In addition, the linear and quadratic correction blocks in Algorithm 3 use only a *constant* number of extra oracle calls and a constant number of extra recursive calls at tolerance $\eta/\sqrt{\gamma}$ (one call for the linear term and a constant number for the quadratic term). Therefore there exists a constant $c_2$ such that the fine-regime cost satisfies

$$T(\eta) \leq c_1 \, K\rho(\eta)^{-2}\Big(1 + T(\rho(\eta)/\sqrt{\gamma})\Big) + c_2\Big(1 + T(\eta/\sqrt{\gamma})\Big) + c_2. \tag{D.27}$$

Now use $\rho(\eta) = \Theta(\eta^{1/3})$, so $\rho(\eta)^{-2} = \Theta(\eta^{-2/3})$. Also define the constant

$$c := \frac{1}{\sqrt{\gamma}}\Big(\frac{6}{M}\Big)^{1/3}, \qquad \text{so that} \qquad \frac{\rho(\eta)}{\sqrt{\gamma}} = c\eta^{1/3}.$$

Absorbing constants and polylogarithms into $\tilde{O}(\cdot)$, (D.27) implies the simplified cascade recurrence

$$T(\eta) \leq \tilde{O}\big(\eta^{-2/3}\big) \cdot \Big(1 + T(c\eta^{1/3})\Big) + \tilde{O}\big(1 + T(\eta/\sqrt{\gamma})\big). \tag{D.28}$$

The second term does not change the exponent because it appears with only constant multiplicity; it can be absorbed into constants once we know $T(\eta)$ grows polynomially as $\eta \downarrow 0$.

**V. Solve the cascade recurrence: $T(\eta) = \tilde{O}(\eta^{-1})$.** We now show that (D.28) yields

$$T(\eta) = \tilde{O}(\eta^{-1}). \tag{D.29}$$

To see the exponent, first ignore the lower-order $\tilde{O}(1 + T(\eta/\sqrt{\gamma}))$ term and focus on the dominant cascade:

$$T(\eta) \lesssim A\eta^{-2/3}\, T(c\eta^{1/3}) + A\eta^{-2/3}, \tag{D.30}$$

for some constant $A = \tilde{O}(K)$. Define the tolerance sequence $\eta_0 := \eta$ and $\eta_{t+1} := c\eta_t^{1/3}$. As in Grill et al. (2019) and in the proof of Theorem 4.3, this sequence reaches a constant in $O(\log\log(1/\eta))$ steps because $1/3 < 1$. Let $L$ be the smallest index with $\eta_L \geq \eta_{\text{base}}$ for a fixed constant $\eta_{\text{base}} \in (0, \kappa]$. Then $T(\eta_L) = O(1)$.

Unrolling (D.30) along the sequence gives

$$T(\eta_0) \lesssim A^L\Big(\prod_{t=0}^{L-1} \eta_t^{-2/3}\Big) T(\eta_L) + \sum_{j=0}^{L-1} A^{j+1}\Big(\prod_{t=0}^{j} \eta_t^{-2/3}\Big).$$

The sum is dominated by its final term up to a multiplicative polylog factor (because the products grow rapidly as $\eta_t$ decreases backward), so it suffices to understand the product $\prod_{t=0}^{L-1} \eta_t^{-2/3}$.

Using the explicit form

$$\eta_t = c^{1+1/3+\cdots+1/3^{t-1}}\, \eta^{1/3^t} = c^{\frac{1-(1/3)^t}{1-1/3}}\, \eta^{1/3^t},$$

we have

$$\eta_t^{-2/3} = c^{-\Theta(1)}\, \eta^{-2/3^{t+1}}.$$

Therefore

$$\prod_{t=0}^{L-1} \eta_t^{-2/3} = c^{-\Theta(L)}\, \eta^{-\sum_{t=0}^{L-1} 2/3^{t+1}} = c^{-\Theta(L)}\, \eta^{-(1-(1/3)^L)}.$$

Since $(1/3)^L$ is negligible and $L = O(\log\log(1/\eta))$, the factor $c^{\Theta(L)} A^L$ is polylogarithmic in $1/\eta$, and the exponent of $\eta$ is 1 up to a vanishing correction. This yields (D.29), i.e. $T(\eta) = \tilde{O}(\eta^{-1})$.

(If desired, one can incorporate the suppressed $\tilde{O}(1 + T(\eta/\sqrt{\gamma}))$ term by induction, noting that $\eta/\sqrt{\gamma}$ differs from $\eta$ only by a constant factor and thus preserves the same $\eta^{-1}$ scaling.)

**VI. Total oracle calls of SECONDORDERSMOOTHCRUISER.** Algorithm 1 performs a single call to ESTIMATEQ$(s, \varepsilon, \delta/2)$ and then applies $F$. The application of $F$ is computational (no oracle calls), so the oracle complexity is exactly that of ESTIMATEQ.

By Algorithm 2, ESTIMATEQ performs $K\,N(\varepsilon, \delta/2)$ oracle calls at $(s, a)$, where $N(\varepsilon, \delta/2) = \Theta(\varepsilon^{-2}\log(2K/\delta))$. Each oracle call also triggers one call to SAMPLEV2 at tolerance $\varepsilon/\sqrt{\gamma}$. Hence the total oracle calls satisfy

$$
\begin{aligned}
n(\varepsilon, \delta) &\le K\,N(\varepsilon, \delta/2) \cdot \left(1 + T(\varepsilon/\sqrt{\gamma})\right) \\
&= \tilde{O}\big(\varepsilon^{-2}\big) \cdot \left(1 + \tilde{O}\big((\varepsilon/\sqrt{\gamma})^{-1}\big)\right) \qquad \text{(by (D.29))} \\
&= \tilde{O}\big(\varepsilon^{-2}\big) \cdot \tilde{O}\big(\varepsilon^{-1}\big) = \tilde{O}\big(\varepsilon^{-3}\big).
\end{aligned}
$$

All constants depend only on $(K, \gamma, \lambda, \tau, \mu_{\min}, \|C\|_\infty)$ through $B$, $M$ and the fixed multiplicative costs in the algorithms, completing the proof. $\qquad\square$

