# OpenReview forum: "Second-Order Smooth Planning with Optimal-Transport Bellman Smoothing"
_ICML.cc/2026/Conference — ICML 2026 spotlight_

### Official Review · Reviewer_KKDc · 2026-02-26

**Soundness:** 3
**Presentation:** 1
**Significance:** 2
**Originality:** 3
**Overall Recommendation:** 4
**Confidence:** 3

**Summary:**

The paper improves existing sample complexity bounds with tighter curvature-complexity theory. They formulated a optimal transport regulariser over action distributions and the resulting Bellman objective remains in close-form. Under such formulation, the complexity bound can be sharpened from existing $O(\epsilon^{-4})$ to $O(\epsilon^{-3})$.

Specifically, the paper introduced the OT-smoothed aggregator as a new type of Bellman objective in Definition 5.1. Then it shows that such objective is has closed form solution just like the KL regulariser case. Under some regularity conditions, the Hessian of the aggregator is has Lipschitz property. The second term in the Hessian of the aggregator is a type of variance. Leverage these properties, the paper introduces SecondOrderSmoothCrusier that satisfy sharp sample complexity bound per given state. Further, they develop a gap-dependent variant of the algorithm and show various of theoretical advantages.

**Compliance With Llm Reviewing Policy:**

Affirmed.

**Final Justification:**

The paper shows clear merits by neatly using curvature theory and optimal transport to reduce the convergence speed of regularised MDP planning. I believe the theory and technique are useful for broader areas too.

**Key Questions For Authors:**

What is the reason to develop so many variants of the algorithm to eventually arrive at Theorem 9.2?

Or perhaps what are the motivations and reasons for each of the algorithm under the optimal transport setups? These should be made clearer for the reader to follow.

A worst case complexity bound is given, but how does the bound chance when the reference policy is picked with extra conditions? Or how to pick a good reference policy to make the algorithm converge fast? I think this can be useful for practice.

**Limitations:**

Yes in the form of future research directions. But I don't see how the theoretical algorithm can be made efficient in practice. Also, as mentioned before, Theorem 7.1 requires discrete settings, but this is not mentioned in the paper.

**Strengths And Weaknesses:**

Strength

The paper introduced a novel second order Bellman objective based on optimal transport instead of entropy. Theoretical results improve the current existing bound and proof techniques can be useful for future research. Multiple variants of the algorithms are proposed and comprehensive theory around the proposed objective is established.

Weakness
The presentation needs to be drastically improved as I can't follow the paper without reading some of the precursor work, such as Grill et.al. (2019). Motivations, broad impacts and why the method is important is missing in the introduction section. Many terminologies and definitions are provided with minimum explainations. For instance, the second paragraph of the introduction already introduced many symbols and $F$ is not defined at all until much later. The way introduction is written seems to assume the reader is very familiar with this line of work, otherwise it's very hard to follow. In section 4, the paper assumes strong knowledge in sampling based planners, which may also not be the case. Also, the motivation of optimal transport regulariser and the intuition behined the proposed equation is lacking. Meanwhile, it's not obvious straightaway what motivates the author to use confidence gap on top of the proposed SecondOrderSmoothCrusier. If all algorithms are written in full, why only an algorithm outline is given for OT-GapCruiser? The confidence gap theory is also not reviewed in the background.

Overall, I suggest the author to be less technical in the introduction section, and aim to explain the broader impact of the work, its motivation and why it is valuable. The background section can be expanded to detail how sampling based planners work and how people typically bound the estimations in the sampling based planners. When definition, proposition and theory are given, explain explictly in plain text their importance and intuition. To make more space, I suggest the proof sketches can be removed and less important technical details can be moved into the appendix.

In Theorem 7.1, it requires the bounds to be computed exactly, which is only possible for discrete settings? Hence, the resulting theory and algorithm is actually only useful for discrete problems? However, this quite crucial point is not addresses anywhere in the paper.

It's very hard to see how to apply any of the theoreticaly results to practice, especially given the computational complexity of the algorithm is very high.

---

> ### Author Rebuttal · Authors · 2026-03-31
>
> Thank you for the careful review. We address your points one by one.
>
> **1) Presentation / accessibility.** Thank you for pointing this out. In the revision, we will substantially rewrite the introduction and background to be more pedagogical: defining regularized Bellman operators early, explaining why entropy regularization yields LogSumExp, and describing how sampling-based planners like SmoothCruiser control their estimators. We will then motivate OT smoothing as the higher-order alternative. We will also add plain-language intuition after each main result, move proof sketches and lower-priority material to the appendix, and include a brief background on confidence-gap / MDP-GapE style analyses.
>
>  **2) Why OT regularization?** Our goal is not to replace entropy by an unrelated objective. The OT-smoothed backup is a structured generalization of entropy smoothing: after removing the $Q$-independent offset, it becomes a $\mu_s$-weighted mixture of LogSumExp terms applied to cost-shifted scores; when $C=0$, it reduces to the standard entropy-regularized LogSumExp backup. The reason for introducing OT is that the action-cost matrix $C$ couples nearby actions, which gives geometry-aware smoothing and, crucially, a Lipschitz Hessian / cubic Taylor remainder. That is what enables the improvement.
>
>  **3) Why so many algorithmic variants?** The main thread of the paper is:
>  curvature theory (Sec. 4) $\rightarrow$ OT-smoothed operator (Sec. 5) $\rightarrow$ SecondOrderSmoothCruiser (Sec. 6) $\rightarrow$ worst-case theorem (Theorem 9.2).
>  This is the core contribution. OT-GapCruiser and OT-GapE are extensions showing that the same OT / second-order ideas can also yield instance-dependent guarantees. The motivation for the gap-based variant is that worst-case bounds do not exploit easy instances where the root action gaps are large; UGapE-style elimination does.
>
> **4) Theorem 7.1.** Section 3 already assumes a finite action set, so the setting of this paper is a discrete action set, not a continuous one..
>
>  **5) Practical efficiency.** Our main contribution is in **oracle/sample complexity**, where the result improves the worst-case dependence from $\widetilde O(\varepsilon^{-4})$ to $\widetilde O(\varepsilon^{-3})$. At the same time, we agree that the current submission is primarily a **theory paper**, not a claim of immediate large-scale practical superiority. The algorithm does **not** explicitly construct the full Hessian, and it does **not** run a generic OT solver at every state: the entropic OT-smoothed Bellman operator has a closed form, and the quadratic correction is estimated through a variance / cross-product identity using only a constant number of extra sampled actions and recursive calls.
>
> **6) Reference policy $\mu_s$.** In the current theory, the worst-case exponent $\widetilde O(\varepsilon^{-3})$ does not change with the choice of $\mu_s$ as long as Assumption 5.2 holds uniformly, i.e., each $\mu_s$ has full support with $\inf_{s,a}\mu_s(a)\ge \mu_{\min}>0$ independent of $\varepsilon$. What changes is the hidden constant. The reference policy $\mu_s$ enters the OT operator through its closed form, the sampling policy, and the Hessian correction. In particular, the Lipschitz-Hessian constant scales as $M = O((\tau^2\lambda^2)^{-1}) \cdot \mathrm{poly}(K, \mu_{\min}^{-1})$, so better-bounded $\mu_s$ improves the constants while the exponent remains 3.
>
>  The practical interpretation is that $\mu_s$ is a reference prior over action neighborhoods. This becomes especially clear from our variance identity (Lemma 6.1):
>
> $\Delta^\top \nabla^2 F_s^{OT}(\hat Q_s) \Delta = \frac{1}{\tau \lambda} E_{J \sim \mu_s}[\text{Var}_{A \sim p_J(\hat Q_s)}(\Delta_A)]$
>
>  Thus $\mu_s$ reweights the local variances that drive the second-order estimator. A better $\mu_s$ can therefore improve practical convergence by emphasizing informative / low-variance action neighborhoods. Also, when (C=0), the OT operator collapses to the standard entropy-regularized LogSumExp backup (up to the explicit $Q$-independent constant), so $\mu_s$ effectively disappears; this shows that $\mu_s$ matters only through its interaction with the action geometry encoded by $C$.
>
> For practice, our recommendation is: use a $\mu_s$ with full support, avoid overly concentrated choices, and align it with any available prior over promising actions or action clusters. Uniform $\mu_s$ is the safest default. If a heuristic or learned prior $\pi_s$ is available, a robust choice is a smoothed prior
>
> $$
> \mu_s = (1 - \eta ) \pi_s + \eta \cdot U(A)
> $$
>
> which preserves full support, keeps $\mu_{\min} \ge \eta / K$, and still biases the OT coupling toward plausible actions.
>
> Thank you again for the constructive feedback, which has helped us improve the paper substantially. If our responses have clearly addressed the raised concerns, we kindly ask you to reconsider your evaluation. We welcome any further questions or comments.

---

> > ### Author Rebuttal · Reviewer_KKDc · 2026-04-02
> >
> > I believe reading through the replies does clarify things. The curvature theory around OT may be useful for other areas too. Are you aware of any other works, maybe in learning theory, that rely on similar theoretical insights to yours? Those should be cited in the revised paper too, but draw the distinctions between your method and theirs. Please update overall presentation for the revised manuscript as suggested. I have improved your score.

---

> > > ### Author Response · Authors · 2026-04-02
> > >
> > > Thank you for this helpful suggestion. We agree that there are important conceptual parallels in optimization and learning theory where stronger smoothness or higher-order smoothness leads to improved complexity rates, and we will cite them in the revised manuscript with clear distinctions.
> > >
> > > In optimization, the closest analogy is the cubic-regularization / higher-order tensor-method literature. The cubic-regularization line of Nesterov and Polyak (2006) uses a second-order local model under a Lipschitz-Hessian assumption, and later work such as Nesterov (2021), Gasnikov et al. (2019), and Bubeck et al. (2019) develops higher-order tensor methods whose complexity improves with the order of available smoothness. This is philosophically close to our curvature-complexity viewpoint: a tighter Taylor remainder leads to a better complexity exponent. The key difference is that those works optimize a static objective, whereas our paper studies planning with a generative model, where the relevant curvature is that of the Bellman aggregator.
> > >
> > > There are also close parallels in learning theory. Akhavan, Pontil, and Tsybakov (2020) explicitly study how higher-order smoothness improves rates in derivative-free optimization and continuous bandits. This is again conceptually related, but the object whose smoothness matters is different: in their setting it is the reward/objective function over the action space, while in our setting it is the regularized Bellman backup itself.
> > >
> > >  On the OT side, Cuturi (2013) introduced entropically regularized OT as a computational smoothing device, and Rigollet and Stromme (2025) analyze the sample complexity of entropic OT. These works are not planning papers, but they are relevant for situating our use of entropic OT as a way to engineer smoother geometry. We will add these citations and make the distinction clearer in the revision: our contribution is not merely that “higher-order smoothness helps,” but that in planning with a generative model, the order of the Bellman Taylor remainder controls the full recursive oracle complexity, and entropic OT provides a concrete Bellman regularizer that moves the problem from the ($\beta=2$) regime to the first nontrivial higher-order regime ($\beta=3$).
> > >
> > > We will also incorporate all presentation improvements as discussed. We are grateful for the Reviewer's feedback, which has meaningfully strengthened the paper.
> > >
> > > Best regards,
> > >
> > > The Authors
> > >
> > > **References**
> > >
> > > * Nesterov, Y., and Polyak, B. T. (2006). *Cubic regularization of Newton method and its global performance*. Mathematical Programming.
> > > * Nesterov, Y. (2021). *Implementable tensor methods in unconstrained convex optimization*. Mathematical Programming.
> > > * Gasnikov, A., Dvurechensky, P., Gorbunov, E., Vorontsova, E., Selikhanovych, D., and Uribe, C. A. (2019). *Optimal Tensor Methods in Smooth Convex and Uniformly Convex Optimization*. COLT / PMLR 99.
> > > * Bubeck, S., Jiang, Q., Lee, Y. T., Li, Y., and Sidford, A. (2019). *Near-optimal method for highly smooth convex optimization*. COLT / PMLR 99.
> > > * Akhavan, A., Pontil, M., and Tsybakov, A. B. (2020). *Exploiting Higher Order Smoothness in Derivative-free Optimization and Continuous Bandits*. NeurIPS 2020.
> > > * Cuturi, M. (2013). *Sinkhorn Distances: Lightspeed Computation of Optimal Transport*. NeurIPS 2013.
> > > * Rigollet, P., and Stromme, A. J. (2025). *On the sample complexity of entropic optimal transport*. Annals of Statistics.

---

### Official Review · Reviewer_FjN6 · 2026-03-10

**Soundness:** 2
**Presentation:** 2
**Significance:** 2
**Originality:** 2
**Overall Recommendation:** 4
**Confidence:** 1

**Summary:**

This paper studies planning with a generative model for entropy-regularized MDPs. By using an entropic optimal-transport regularizer over action distributions together with curvature complexity theory, the authors propose Second-Order SmoothCruiser and OT-GapCruiser, which provide problem-independent and instance-dependent oracle calls, respectively. In particular, Second-Order SmoothCruiser improves the worst-case complexity exponent from $4$ to $3$ when Taylor remainder of order $3$ while prior works only cover Taylor remainder of order $2$.

**Compliance With Llm Reviewing Policy:**

Affirmed.

**Final Justification:**

This paper first gives a general theorem that relates the local Taylor remainder order to sample complexity through curvature-complexity theory, and introduces an OT-smoothed Bellman operator that admits a closed-form expression, an explicit policy gradient, and a Lipschitz Hessian. Based on these results, the paper proposes Second-Order SmoothCruiser, which uses the OT-smoothed aggregator together with a cross-product debiasing trick, and OT-GapCruiser. As noted in my review, I did not understand the value of improving the sample complexity when the Taylor remainder is of order 3. The rebuttal kindly addresses this point.

**Key Questions For Authors:**

Could you elaborate on the computational cost and compare it with prior work? (If I understand correctly, SmoothCruiser avoids computing the Hessian by using an unbiased estimator.)

**Limitations:**

Yes

**Strengths And Weaknesses:**

As the authors clearly present, this paper first gives a general theorem that relates the local Taylor remainder order to sample complexity through curvature-complexity theory, and introduces an OT-smoothed Bellman operator that admits a closed-form expression, an explicit policy gradient, and a Lipschitz Hessian. Based on these results, the paper proposes Second-Order SmoothCruiser, which uses the OT-smoothed aggregator together with a cross-product debiasing trick, and OT-GapCruiser, which combines the second-order estimator with UGapE-style elimination. In particular, Second-Order SmoothCruiser achieves an $\epsilon^3$ sample complexity when the Taylor remainder is of order $3$.

Regarding the novelty of this paper, I may be missing some background, but I am not fully convinced. Although this work improves the sample complexity when the Taylor remainder is of order $3$, and I do believe the paper is valuable, I am not sure whether its level of novelty meets the bar for this conference. If I understand correctly, when the Taylor remainder is of order $2$, this work achieves the same complexity as prior work. If so, why is the case where the Taylor remainder is of order $3$ important?

---

> ### Author Rebuttal · Authors · 2026-03-31
>
> Thank you for the thoughtful review. We appreciate the novelty concern, and we would like to make the “why $\beta=3$ matters” point more explicit.
>
>
> **Regarding the novelty:** The $\beta=2$ part of our theory is **not** intended as the novelty claim; it is the consistency check that the framework correctly recovers SmoothCruiser. The novelty is that the paper identifies the order $\beta$ of the local Bellman Taylor remainder as the quantity controlling the full recursive planning cascade, and then gives a concrete regularized Bellman operator and estimator that provably move the problem into the next regime, $\beta=3$.
>
> This matters because $\beta=3$ is the **first nontrivial value** that strictly changes the worst-case oracle exponent. In our theory, a SmoothCruiser-type planner has total exponent $2 + 2/(\beta-1)$: thus $\beta=2$ gives $4$, whereas $\beta=3$ is the first value that drops the exponent to $3$. This is not a cosmetic gain. It changes the required coarse $Q$-accuracy from $\Theta(\sqrt{\varepsilon})$ to $\Theta(\varepsilon^{1/3})$, reduces the per-action Monte Carlo cost from $\Theta(\varepsilon^{-1})$ to $\Theta(\varepsilon^{-2/3})$, and shortens the recursive cascade from exponent $2$ to exponent $1$, yielding $\widetilde{O}(\varepsilon^{-3})$ instead of $\widetilde{O}(\varepsilon^{-4})$.
>
> What is technically new is that reaching this $\beta=3$ regime requires more than a generic theorem. One needs:
> (i) a Bellman operator whose gradient is still a valid policy;
> (ii) a provable cubic remainder / Lipschitz Hessian; and
> (iii) a way to estimate the second-order term from generative-model samples **without** reconstructing the full $Q$-error vector.
>
> Our OT-smoothed operator provides (i)–(ii), and Lemma 6.1 plus the cross-product debiasing trick provides (iii). In this sense, the contribution is not just to restate SmoothCruiser in more general language, but to show that changing the Bellman regularizer can genuinely move regularized planning into a new statistical regime.
>
> **On computational cost:** as in SmoothCruiser and related generative-model planning work, our formal complexity measure is the number of oracle calls. Under that metric, the improvement is from $\widetilde O(\varepsilon^{-4})$ to $\widetilde O(\varepsilon^{-3})$. Importantly, our algorithm does **not** explicitly form the full Hessian, nor does it solve a generic OT problem iteratively at each state. The entropic OT regularizer admits a closed-form aggregator, and the quadratic term $\frac12 \Delta^\top \nabla^2F(\hat Q)\Delta$ is rewritten as an expectation/variance. Using two independent samples, we obtain an unbiased estimator of this quadratic correction with only a constant number of additional sampled actions / recursive calls. So, in oracle terms, the second-order correction has constant overhead relative to SmoothCruiser; the asymptotic gain comes from the improved tolerance schedule and hence the smaller cascade exponent.
>
>  There is, however, additional local arithmetic overhead. SmoothCruiser evaluates a single LogSumExp-style object, whereas our OT-smoothed backup requires computing the local softmax mixture $p_{\cdot j}(\hat Q)$ over the finite action set, so per visited state the arithmetic is somewhat higher. We will add a paragraph clarifying this distinction between oracle complexity and local computation, and stress that no full Hessian construction or Sinkhorn loop is required in the implementation analyzed here.
>
> Thank you again. If our response has clearly addressed your concerns, we kindly ask you to reconsider your evaluation. We are happy to address any additional concerns.

---

> > ### Author Rebuttal · Reviewer_FjN6 · 2026-04-03
> >
> > Thank you for the clarifications. I have improved my score, but honestly, due to my limited background, I am not fully confident in my judgment and therefore have kept my confidence level unchanged.

---

### Official Review · Reviewer_TFKi · 2026-03-15

**Soundness:** 3
**Presentation:** 3
**Significance:** 3
**Originality:** 4
**Overall Recommendation:** 5
**Confidence:** 2

**Summary:**

The authors consider the planning problem in general MDPs with a generative model. First, they studied a general setting with an oracle that optimizes the Taylor polynomial of regularized values, and established that, in the case of linear approximation to usual entropy-regularized values, they recover the sample complexity of an algorithm from prior work, SmoothCruiser. At the same time, the general theory predicts that it might be possible to achieve faster convergence if one considers a different regularization, where the second-order Taylor polynomial approximation will be feasible to estimate.  The authors demonstrated that the optimal-transport-related regularization exactly satisfies the required properties and achieves improved convergence bounds under a new regularization.

**Compliance With Llm Reviewing Policy:**

Affirmed.

**Final Justification:**

As I mentioned in the rebuttal, I believe that this paper provides a highly non-trivial extension of existing work, and I'm happy to keep my Accept.

**Key Questions For Authors:**

1. What is the difference between OT-regularized, entropy-regularized, and unregularized values? It is well-known that in the entropy case, since the trajectory entropy is bounded by $\log(A)$, the worst-case difference between entropy-regularized and unregularized value is of order $\lambda \cdot \log(A) / (1-\gamma)$. Is there a similar relation for the OT-regularized setting?
2. (connected to the previous one) How to transfer your $1/\varepsilon^3$ guarantees for OT-regularized setting to the solution of entropy-regularized or unregularized MDPs?

**Limitations:**

Yes.

**Strengths And Weaknesses:**

### Strength

- Very clear motivation and introduction of the particular method, tailored for second-order correction. Additionally, I found the approach to switch to OT regularization an elegant and interesting solution.
- The authors analyzed the method in both worst-case and instance-dependent settings.
- The proof techniques are very reasonable and seem to be correct, although I did not verify all the details in Appendix.


### Weaknesses
- It is not clear how the OT-regularized solutions are transferred to unregularized ones, or at least entropy-regularized, which are much more well-studied.




#### Small remark
There is a misprint in Definition 4.1: I believe that the Taylor remainder polynomial should have an index $\beta-1$ ($T_{\beta-1}$), not $T_1$.

---

> ### Author Rebuttal · Authors · 2026-03-31
>
> Thank you for the positive assessment and for the helpful questions. We are glad that the OT regularization idea and the second-order construction came across as interesting.
>
> On the relation between the three objectives: the unregularized value uses the hard max backup; the entropy-regularized value uses the usual LogSumExp backup; and our OT-regularized backup is
> $$
> F_s^{OT}(Q)=\max_{\pi\in\Delta(A)}{\langle \pi,Q\rangle-\tau W_\lambda(\pi,\mu_s)}.
> $$
> Importantly, this is not unrelated to entropy regularization. Proposition 5.3 shows that, after removing the explicit $Q$-independent offset, $(F_s^{OT})$ is a $\mu_s$-weighted mixture of LogSumExp terms applied to cost-shifted scores $(Q_i-\tau C_{ij})$. In the special case $(C\equiv 0)$, this normalized operator reduces exactly to the standard entropy-regularized LogSumExp backup with temperature $(\tau\lambda)$. We will make this connection explicit much earlier in the paper.
>
> This also yields a clean transfer statement. After the same normalization, OT and entropy backups differ by at most $O(\tau |C|\_{\infty})$ uniformly, since each local LogSumExp sees inputs shifted by at most $\tau |C|\_\infty$. Combining this with the standard entropy-vs-max bound gives an OT vs Unregularized gap of order
> $$
> O\big(\tau(|C|\_{\infty} + \lambda \log K)\big)
> $$
> per Bellman backup, and therefore
> $$
> O\big(\tau(|C|\_{\infty} + \lambda \log K)/(1-\gamma)\big)
> $$
> at the value-function level by contraction. We agree this comparison is important, and we will add it explicitly in the revision.
>
> Consequently, the transfer to entropy-regularized planning is direct: with $(C\equiv 0)$, our framework recovers the entropy-regularized objective (up to the explicit removable offset), so the same $(\widetilde O(\varepsilon^{-3}))$ guarantee applies to that target as well. For unregularized planning, one can choose $(\tau,\lambda)$ so that the regularization bias is at most $(O(\varepsilon))$, and then estimate the corresponding OT value to the remaining error budget.
>
> Furthermore, thank you for catching the typo around Assumption 4.1. You are right that the generic curvature assumption should refer to the appropriate Taylor polynomial order; in the general statement this should be the order$-((\beta-1))$ Taylor polynomial (so $T_1$ for ($\beta=2), T_2$ for $(\beta=3)$). We will fix this notation.
>
> We truly appreciate this valuable feedback. We hope our revisions and responses satisfactorily resolve all the issues you raised. We are happy to address any additional concerns.

---

> > ### Author Rebuttal · Reviewer_TFKi · 2026-04-02
> >
> > Thank you for clarifying the connection between OT- and entropy-regularized objectives. Overall, this work seems to be a strong and quite non-trivial extension of Grill et al. (2019) from the point of view of the underlying idea using non-standard regularization, and I'm happy to keep my score.

---

> > > ### Author Response · Authors · 2026-04-02
> > >
> > > We thank the Reviewer once more for the valuable comments and support. The revised manuscript will fully incorporate the feedback from this rebuttal discussion.
> > > Best regards,
> > > The Authors

---

### Official Review · Reviewer_5U7X · 2026-03-18

**Soundness:** 3
**Presentation:** 2
**Significance:** 4
**Originality:** 3
**Overall Recommendation:** 5
**Confidence:** 4

**Summary:**

The paper studies planning with a generative model in a regularized Markov decision process. In this setting, the learner is allowed to query an oracle for samples of rewards and next states at any chosen state--action pair, and the goal is to estimate the optimal value of a root state using as few oracle calls as possible. In the entropy-regularized setting, the usual hard Bellman maximization over actions is replaced by a smooth backup, the LogSumExp operator, which corresponds to optimizing over stochastic action distributions with an entropy penalty. Prior work such as \emph{SmoothCruiser} exploits the smoothness of this entropy-regularized Bellman operator to derive a recursive Monte Carlo planning method with worst-case sample complexity of order $\widetilde O(\varepsilon^{-4})$.

The main idea of the present paper is that this complexity exponent is tied to the local curvature of the Bellman aggregator: the smoother the Bellman backup, the better one can exploit Taylor expansion and recursive Monte Carlo estimation. The authors first formulate a general curvature--complexity relationship, and then propose to replace entropy regularization by an \emph{optimal-transport} regularizer over action distributions. This yields an OT-smoothed Bellman operatorcwhose Hessian is Lipschitz, allowing the use of a second-order approximation rather than the first-order approximation used in SmoothCruiser. The resulting algorithm uses a debiased quadratic correction estimator and achieves a claimed $\widetilde O(\varepsilon^{-3})$ complexity. In short, the paper's punchline is that replacing entropy-based Bellman smoothing by optimal-transport-based Bellman smoothing leads to a better curvature regime and hence to a better worst-case planning complexity exponent.

I think this is a nice observation. The recursive estimation structure is also somewhat reminiscent of recent multilevel Monte Carlo ideas for entropy-regularized MDPs, in particular Meunier, Reisinger, and Zhang, Efficient Learning for Entropy-Regularized Markov Decision Processes via Multilevel Monte Carlo (2025), which the authors should cite and compare to carefully. The technical routes are not identical, but the connection is close enough that the lack of discussion weakens the positioning of the paper.

**Compliance With Llm Reviewing Policy:**

Affirmed.

**Key Questions For Authors:**

The authors should cite and compare to Meunier, Reisinger, and Zhang (2025) on multilevel Monte Carlo for entropy-regularized MDPs. Even if the comparison ultimately favors the present approach in the OT setting, it seems important to explain how the two-level / recursive Monte Carlo viewpoint here differs from or relates to that recent line of work.

The paper should be rewritten substantially for clarity. The introduction should define the main objects much earlier and motivate them in a way accessible to readers who are not already experts in regularized dynamic programming.

**Limitations:**

yes

**Strengths And Weaknesses:**

Strengths:

 The paper contains a genuinely interesting core idea: the local curvature of the regularized Bellman operator controls the recursive planning complexity, and optimal-transport regularization provides a concrete way to improve this curvature relative to the standard entropy-regularized/LogSumExp case.
 The proposed viewpoint is conceptually clean once understood: entropy regularization gives a soft Bellman backup, SmoothCruiser uses a first-order approximation of that backup, and the present paper argues that an OT-based backup admits a better second-order treatment.


Weaknesses:

 My main concern is exposition. The paper is poorly written and hard to follow for a reader who is not already familiar with regularised MDPs, smooth Bellman backups, and the SmoothCruiser line of work. The paper would benefit from a much more pedagogical introduction. In particular, it should explain early on what a regularised Bellman operator is, why entropy regularisation leads to the LogSumExp backup, how SmoothCruiser uses first-order Taylor approximation, and only then introduce the OT-smoothed Bellman operator as a higher-order alternative. At present, these pieces arrive too late and in a way that makes the paper unnecessarily difficult to parse.

The novelty claim also needs to be positioned more carefully relative to adjacent work. In particular, the recursive Monte Carlo flavor is reminiscent of multilevel Monte Carlo methods for entropy-regularized MDPs, especially the recent work of Meunier, Reisinger, and Zhang (2025). Even if the present paper is different in that it changes the Bellman regularizer and leverages higher-order curvature rather than randomized multilevel debiasing, this comparison seems important and should be discussed explicitly.

---

> ### Author Rebuttal · Authors · 2026-03-31
>
> Thank you for the positive assessment and for highlighting the core idea. We are glad that the curvature-driven viewpoint and the OT-based smoothing came across as interesting and potentially impactful.
>
> We agree with your point about Meunier, Reisinger, and Zhang (2025), and we thank you for the pointer. We will cite and discuss this paper explicitly in the revision. We view the two works as complementary rather than competing. Their paper stays with entropy regularization / the soft Bellman operator and improves sampling through multilevel Monte Carlo, fixed-point iteration, and an unbiased randomized approximation of the Bellman operator. By contrast, our paper changes the Bellman regularizer itself: we introduce an OT-smoothed Bellman aggregator, prove a Lipschitz Hessian / cubic remainder, and use a debiased quadratic correction inside a SmoothCruiser-style recursion. In other words, their gain comes from MLMC debiasing under the standard entropy regularizer, whereas ours comes from a new regularizer whose higher-order local curvature changes the recursive complexity exponent. We will add a paragraph in Related Work and in the introduction making this distinction explicit, and also clarify that our current focus is finite-action, root-state planning with an explicit worst-case exponent $~O(ε^{-3})$, rather than full Q-function learning in general Polish spaces.
>
> About the current exposition of the paper, we will restructure the introduction so that the main objects appear in the order you suggest: (1) define the regularized Bellman operator early; (2) explain why entropy regularization gives the LogSumExp backup; (3) explain in one paragraph how SmoothCruiser uses a first-order Taylor approximation and why the quadratic remainder forces the $ O(\epsilon^{-4})$ cascade; (4) only then introduce the OT-smoothed Bellman operator as the higher-order alternative; and (5) add a short roadmap paragraph summarizing the chain “OT regularizer -> Lipschitz Hessian -> quadratic-term estimator -> $ O(\epsilon^{-3})$ complexity.” We also plan to add a compact comparison paragraph/table to make the positioning against SmoothCruiser and MLMC-based entropy-regularized methods easier to parse.
>
>
> Thank you again; these are very helpful suggestions, and we believe they will materially improve the clarity and positioning of the paper. We welcome any further questions or comments.

---

> > ### Author Rebuttal · Reviewer_5U7X · 2026-03-31
> >
> > Thanks for the clarifications. I keep my score as is.

---

> > > ### Author Response · Authors · 2026-04-01
> > >
> > > We sincerely thank the Reviewer's constructive feedback and positive assessment. All suggestions from this discussion will be carefully addressed in the final revision.
> > >
> > > With best regards,
> > >
> > > The Authors

---

### Decision · Program_Chairs · 2026-04-30

**Decision:**

Accept (spotlight)

**Comment:**

The authors introduced a second order Bellman objective based on optimal transport, which is an interesting idea. The reviewers voted unanimously for acceptance. Thus, I recommend acceptance.